# Global coincident bursts of high frequency oscillations across the human cortex coordinate large-scale memory processing

Sathwik Prathapagiri [1] ✉, Jan Cimbalnik[2,3], Jesús S. García-Salinas [1], Marina Galanina[1], Lenka Jurkovicova [2,3,4], Pavel Daniel[3,4], Martin Kojan [3,4], Robert Roman [3,4], Martin Pail[4], Wojciech Fortuna[5], Monika Sluzewska-Niedzwiedz[6], Pawel Tabakow[5], Andrzej Czyzewski [1], Milan Brazdil[3,4] & Michal T. Kucewicz [1,7] ✉

Oscillations in the high gamma and ripple frequency ranges are known to coordinate local hippocampal and neocortical neuronal assemblies during memory encoding and recall. Here, we explored spatiotemporal dynamics and the role of global coordination of these fast oscillatory discharges across the sensory and associational cortical areas in distinct phases of memory processing. Individual bursts of high frequency oscillations were detected in intracranial recordings from epilepsy patients remembering word lists for immediate free recall. We found constant coincident bursting across visual and higher order processing areas, peaking before recall and elevated during encoding of words. This global co-bursting was modulated by memory processing, engaged approximately half of the recorded electrode contact sites, and clustered into a sequence of multiple consecutive bursting events. Our results suggest a general role of global coincident high frequency oscillations in organizing large-scale information processing across the brain necessary especially, but not exclusively, for memory functions.

How widespread brain regions dynamically communicate to generate coherent behavior in response to external or internal stimuli remains difficult to capture. This challenge arises from the high temporal resolution but limited spatial sampling of the current electrophysiological methods. Temporal coordination of neural activities has been proposed as a universal mechanism for integrating information across the brain[1-4]. Oscillations in gamma frequency ranges beyond the classic Berger bands of the EEG spectrum (30–80 Hz)[5], were originally shown to synchronize neuronal firing responses to particular attended stimuli in the cat visual cortex[6,7]. This suggests a role for high-frequency oscillatory activities in binding coherent object representations[8]. Although the exact role in this perceptual binding is debatable[9], oscillations in high gamma and ripple frequency ranges (>60 Hz) have recently been revisited in the context of more general retrieving and integrating information in the brain. Ripple frequency oscillations across the hippocampus and neocortex are associated with encoding and recalling remembered stimuli[10-16]. Several groups have now confirmed that these oscillations are coupled in time within

[1]Brain & Mind Electrophysiology Laboratory, BioTechMed Center, Department of Multimedia Systems, Faculty of Electronics, Telecommunications and Informatics, Gdańsk University of Technology, 80-233 Gdańsk, Poland. [2]Department of Biomedical Engineering & Neurology, St. Anne's University Hospital in Brno, Brno 60200, Czech Republic. [3]Central European Institute of Technology Masaryk University, Brno, Czech Republic. [4]Brno Epilepsy Center, Department of Neurology, St. Anne's University Hospital, member of ERN-EpiCARE, Faculty of Medicine, Masaryk University, Pekarska 53, 602 00 Brno, Czech Republic. [5]Clinical Department of Neurosurgery, Faculty of Medicine, Wroclaw Medical University, Borowska 213, 50-556 Wroclaw, Poland. [6]Clinical Department of Neurology, Faculty of Medicine, Wroclaw Medical University, Borowska 213, 50-556 Wroclaw, Poland. [7]Department of Neurology, Mayo Clinic, 200 1st Street SW, Rochester, MN 55905, USA. ✉e-mail: sathwik.prathapagiri@pg.edu.pl; michal.kucewicz@pg.edu.pl

less than 500 ms across particular functionally related and spatially distributed cortical areas[12,13,17–19]. These findings support the hypothesis that such temporally coordinated high-frequency activities can facilitate integration and binding of information processing across the brain[20]. This would provide an elegant mechanism for global communication through temporal coordination of these fast activities in distributed cortical areas without necessarily requiring precise phase synchronization.

High frequency oscillations (HFOs) reflect coordinated electrophysiological discharges that occur both locally among neuronal assemblies and globally across cortical networks[21]. HFOs encompass a wide range of physiological and epileptiform discharges from gamma, ripple and fast ripple frequencies[21–28]. Originally reported as sharp-wave ripple complexes in the rodent hippocampus and associated with physiological functions[29], HFOs were subsequently found in cortical networks supporting memory and cognitive functions[12–14,18,30,31]. In humans, they were first described as bursts of high gamma power in the visual cortex[32] and as ripple oscillations in the mesial temporal cortex during memory task performance[33]. They were later detected across the cortex in the entire range of gamma, ripple and fast ripple frequencies (60–500 Hz) induced by memory encoding and recall[14]. Although multiple terms and definitions have been established for ripple oscillations and other high-frequency activities[21–23,25], there is a general consensus that they reflect coordinated firing of local neuronal ensembles. These ensembles can be detected and tracked in time and anatomical space.

Growing evidence from multiple groups shows that physiological ripple frequency oscillations in the human cortex are underpinned by coordinated phase-locked spiking of neuronal assemblies[10,12,18,34,35]. This neuronal spiking is not only locked to phases of the oscillation but also reveals specific phase and firing sequences that can be used to decode particular items and predict their correct recall[10,36,37]. The sequences are reactivated following presentation of word stimuli for encoding until their recall. This bursting of spike sequences across multiple neurons occurs at the times of HFO discharges, which in turn are detected around critical events like encoding or recalling stimuli[11,13,14,17]. The sequential firing has so far been studied locally on the micro-scale level of neuronal bursting. Multiple coincident HFOs (co-HFOs) have lately been shown to co-occur at the critical points of scene cuts during movie watching, when a memory trace for a given scene that was just displayed is encoded. This trace is then replayed during subsequent co-HFO events[16], suggesting a possible global macro-scale temporal organization. Altogether, these studies support the role of coordinated bursting of high-frequency oscillations in integrating information for particular memory traces. They can thus be used as neural substrates of engram activities[21], reflecting storage and recalling declarative memories.

In this study, we investigate these coincident HFO discharges in response to four types of events related to the preparation, encoding, distraction and recalling of stimuli in a free recall (FR) and also in a paired-associate learning (PAL) verbal memory tasks to test their possible behavioral functions. Our hypothesis is that global HFO bursting is modulated primarily by memory processing. We explore whether this global cortical coordination[12] engages mainly sensory or higher-order processing areas of the association cortex, as suggested in the previous research[13,16,17,19]. How selective are these globally coordinated oscillations in terms of specific stimuli remembered and recording sites involved? Animal research suggests that encoding a single engram can engage half of the brain areas studied[38]. Finally, what is the temporal organization of distinct coincident HFO discharges? HFO bursting is known to increase before the onset of a behavioral response[10,11,13,16,17,39], but it is unclear how this is related to the actual moment of explicit or implicit recall. Whether this is underpinned on a macro-scale by a single bursting event or a sequence of distinct cortical networks in a train of bursting events has been difficult to assess in the previous paradigms of freely recalling multiple stimuli and complex pictures or movie scenes together. However, it becomes possible with individual word concepts in this study. Our goal was to determine the large-scale structure of coincident HFO discharges underlying the memory processing of common word concepts.

## Results
### Cortical HFO bursts co-occur during memory encoding and recall
We detected a total of 5,266,937 distinct bursts of HFOs from cortical stereo EEG depth electrode contacts implanted in epilepsy patients (Table 1). Each burst had a discrete duration of at least four cycles and peak frequency within a broad 50–600 Hz range[14,15]. We recorded these bursts during a FR task performance and, for some participants, during additional verbal memory tasks such as PAL[40]. All 17 patients with drug-resistant epilepsy (Table 1) performed the FR task at accuracies >15% of words remembered an average from each list (Suppl. Table 1). We used this threshold as the main criterion for inclusion in data analysis. Recall performance averaged 30.7% ± 8.0% (mean ± SD), ranging from 12.8% to 44.4% across participants. Some, but not all, participants completed the PAL and audio versions of the two tasks (Table 1). We detected HFO bursts during four phases of the FR task (Supplementary Fig. S1): (1) countdown digits before the start of each trial, (2) presentation of words for memory encoding, (3) presentation of arithmetic problems during the delay period, and (4) onset of verbalization of the successfully recalled words (Fig. 1a). We semi-automatically marked recall onset using audio signal processing (Supplementary Fig. S2). We confirmed previously reported increases in HFO rates induced by memory encoding and especially by memory recall[11,13,14] (Fig. 1a, b). We observed increased HFO activity in the sensory visual areas of the occipital cortex, both during visual word presentation and during free verbal recall. Notably, participants verbally recalled the remembered words in the absence of any visual stimulation on the computer screen (Fig. 1b, c). This internally driven sensory activation in the occipital cortex may reflect mental imagery of the recalled words.

We quantified each HFO detection as a single point in time[14,41]. This approach allowed us to study not only the rates within particular contact channels but also temporal correlations between channels from the same and different cortical regions. Although we recorded elevated rates on selected electrode contacts during the encoding or recall phases of the task, overall background bursting rates did not differ significantly across the four phases of the task (Kruskal–Wallis test, $H(3) = 2.94$, $p = 0.40$, $N = 12$; Supplementary Fig. S3). The baseline rates in particular frequency ranges were consistent with those reported in previous studies[11,14,15], with about 0.8 bursts detected per second per channel in the high gamma/ripple frequency range (60-250 Hz). Based on the findings of coupled hippocampal-cortical[13] and cortical–cortical[12] HFO detections, we sorted the channels by their pairwise correlation (Pearson's). This analysis revealed a pattern of coincident HFO (co-HFO) detections within a 100-ms window across multiple channels (Fig. 1c), which was most prominent at the moment of recall onset (Fig. 2). To test whether co-HFO abundance tracked behavioral performance, we correlated each participant's total number of co-HFO bursts during recall with their recall accuracy. We found no significant relationship ($r = 0.24$, $p = 0.445$), indicating that overall co-HFO activity was not predictive of memory performance (Suppl. Table 1).

### Coincident high gamma and ripple HFO bursts engage all cortical lobes
In each phase of the FR task, we observed global HFO discharges comprising coincident detections across all five cortical lobes (Fig. 2a). We observed these coincident detections more frequently around stimulus presentation and, especially at the onset of FR (Fig. 2b), on

**Table 1 | Summary of subject demographics and electrode contact data**

| Dataset | Language | Subject ID | Sex | Age | FR runs | PAL runs | Audio FR run | Audio PAL run | Micro contacts | Macro contacts |
|---|---|---|---|---|---|---|---|---|---|---|
| BR | CS | 2 | F | 31 | 1 | 1 | 0 | 0 | 0 | 173 |
| BR | CS | 3 | M | 40 | 1 | 1 | 0 | 0 | 9 | 118 |
| BR | CS | 5 | M | 29 | 1 | 1 | 0 | 0 | 0 | 174 |
| BR | SK | 6 | M | 21 | 1 | 1 | 0 | 0 | 0 | 173 |
| BR | CS | 7 | M | 36 | 2 | 0 | 0 | 0 | 0 | 170 |
| BR | SK | 8 | M | 31 | 1 | 1 | 0 | 0 | 0 | 157 |
| BR | SK | 9 | M | 30 | 1 | 0 | 0 | 0 | 0 | 174 |
| BR | CS | 10 | F | 27 | 2 | 0 | 0 | 0 | 0 | 172 |
| BR | SK | 11 | F | 36 | 2 | 2 | 0 | 0 | 0 | 172 |
| BR | CS | 13 | M | 43 | 1 | 1 | 0 | 0 | 0 | 102 |
| BR | CS | 14 | M | 41 | 1 | 0 | 0 | 0 | 10 | 134 |
| BR | CS | 16 | F | 41 | 2 | 0 | 0 | 0 | 0 | 146 |
| WR | PL | 1 | F | 24 | 2 | 2 | 0 | 0 | 26 | 56 |
| WR | PL | 2 | M | 31 | 3 | 3 | 1 | 1 | 44 | 44 |
| WR | PL | 3 | F | 32 | 2 | 2 | 1 | 0 | 48 | 40 |
| WR | PL | 4 | M | 28 | 2 | 3 | 1 | 1 | 57 | 58 |
| WR | PL | 5 | F | 22 | 3 | 3 | 1 | 1 | 58 | 44 |
| Total counts | CS –8, SK –4, PL –5 | 17 | 10 M | 31.9 | 34 | 24 | 4 | 3 | 282 | 2957 |

Demographic and recording details of subjects included in the study, categorized by language (ISO 639-1 code; *CS* Czech, *SK* Slovak, PL Polish).

channel pairs showing significantly correlated HFO bursting (Pearson's correlation, $r > 0.50$). The probability of detecting co-HFO bursting differed significantly across the four task phases (one-way ANOVA: $F(3,44) = 97.32$, $p = 3.4 \times 10^{-60}$, $N = 12$). Specifically, we found that this probability was significantly higher at the time bins of recall onset compared to the time bins of countdown, distractor and encoding onsets (one-way ANOVA, $F(3,44) = 63.29$, $p = 6.88 \times 10^{-40}$, $N = 12$). Conversely, we found that the probability was significantly lower one second before recall (one-way ANOVA, $F(3,44) = 2.21$, $p = 0.013$, $N = 12$) and increased sharply ~300 ms prior to reaching its peak at onset of vocalization (Fig. 2c). During the encoding phase, the relative rate of co-HFO bursting was the highest during time of word presentation, likely reflecting active memory processing of the word displayed for 1.5 s on the screen. Although countdown digits and arithmetic equations were also presented for at least one second, they elicited significantly lower co-HFO bursting rates than word presentation during the 1-s period following stimulus presentation (one-way ANOVA, $F(2,33) = 12.28$, $p = 5.56 \times 10^{-6}$, $N = 12$). These temporal patterns were consistent across the entire frequency range studied (60–500 Hz), as we confirmed in three independent subranges (Fig. 2d). To account for inter-subject variability inherent in intracranial studies, we reanalyzed the data using a linear mixed-effects model (LMM) with Condition as a fixed effect and Subject as a random intercept. This model tested for task-related effects in each 10-ms time bin, and we indicate significant intervals ($p < 0.05$) by black significance bars in Fig. 2b–d.

We then applied false discovery rate (FDR) correction across time bins for each comparison. We present FDR-corrected trajectories (Supplementary Fig. S4), which provide a time-resolved map of condition-specific differences. These contrast effects aligned closely with those identified in the omnibus model, revealing condition-dependent divergence in co-HFO dynamics.

To assess the influence of the correlation threshold on the observed effects, we performed additional analyses using both more liberal ($r > 0.3$) and more conservative ($r > 0.7$) cutoffs (see Supplementary Fig. S1e). The temporal structure of co-HFO bursting across task phases remained stable across thresholds, consistent with the patterns described above.

We found no significant differences in the average rates of co-HFO bursting in particular cortical lobes across the four task phases (one-way ANOVA: temporal: $F(3, 40) = 0.45$, $p = 0.72$; frontal: $F(3, 40) = 0.02$, $p = 0.997$; parietal: $F(3, 40) = 0.02$, $p = 0.997$; occipital: $F(3, 40) = 0.001$, $p = 0.999$; limbic: $F(3, 40) = 0.45$, $p = 0.72$; Fig. 2e). However, we found a significant effect of the cortical lobes on co-HFO bursting (one-way ANOVA: countdown: $F(4, 50) = 2.90$, $p = 0.031$; encode: $F(4, 50) = 3.29$, $p = 0.018$; distractor: $F(4, 50) = 2.95$, $p = 0.029$; recall: $F(4, 50) = 3.10$, $p = 0.024$). This analysis revealed a consistent pattern across all four task phases, with lower rates in occipital and parietal areas compared with the other three lobes (Fig. 2e). Together, our findings suggest that while co-HFO bursting occurs throughout all task phases but is differentially modulated by memory processing, showing elevated rates during stimulus encoding, and a sharp peak at the time of memory recall.

## Successful memory encoding and cued recall modulate co-HFO bursting

We tested further whether this modulation of coincident HFO bursting is related to memory or other sensory or motor processes. For instance, the peak of co-HFO bursting around the onset of recall word verbalization could potentially be related to preparatory motor activity underlying vocalization. Notably, we observed that the increased probability of coincident bursting preceded the very onset of vocalization by ~300 ms (Fig. 2c), resembling the timing reported of the readiness potential before subjective urge or will to make a movement[42]. In one of the additional tasks, we used the same word pool in a cued recall PAL paradigm, in which patients encoded six pairs of words instead of twelve single words presented sequentially on the screen (Fig. 3a). The same participants also performed audio versions of both the FR and the PAL tasks to ensure that the observed patterns were not specific to visual stimulation. During the PAL recall, we presented only one word from each pair as a cue for recalling its associate. In this paradigm, we observed that the peak of co-HFO bursting shifted earlier, occurring around cue presentation approximately 1 s before the onset of vocalization (Fig. 3c). We tested these time-resolved effects using the previously described linear mixed-effects model (LMM). These findings demonstrated that increased co-HFO bursting

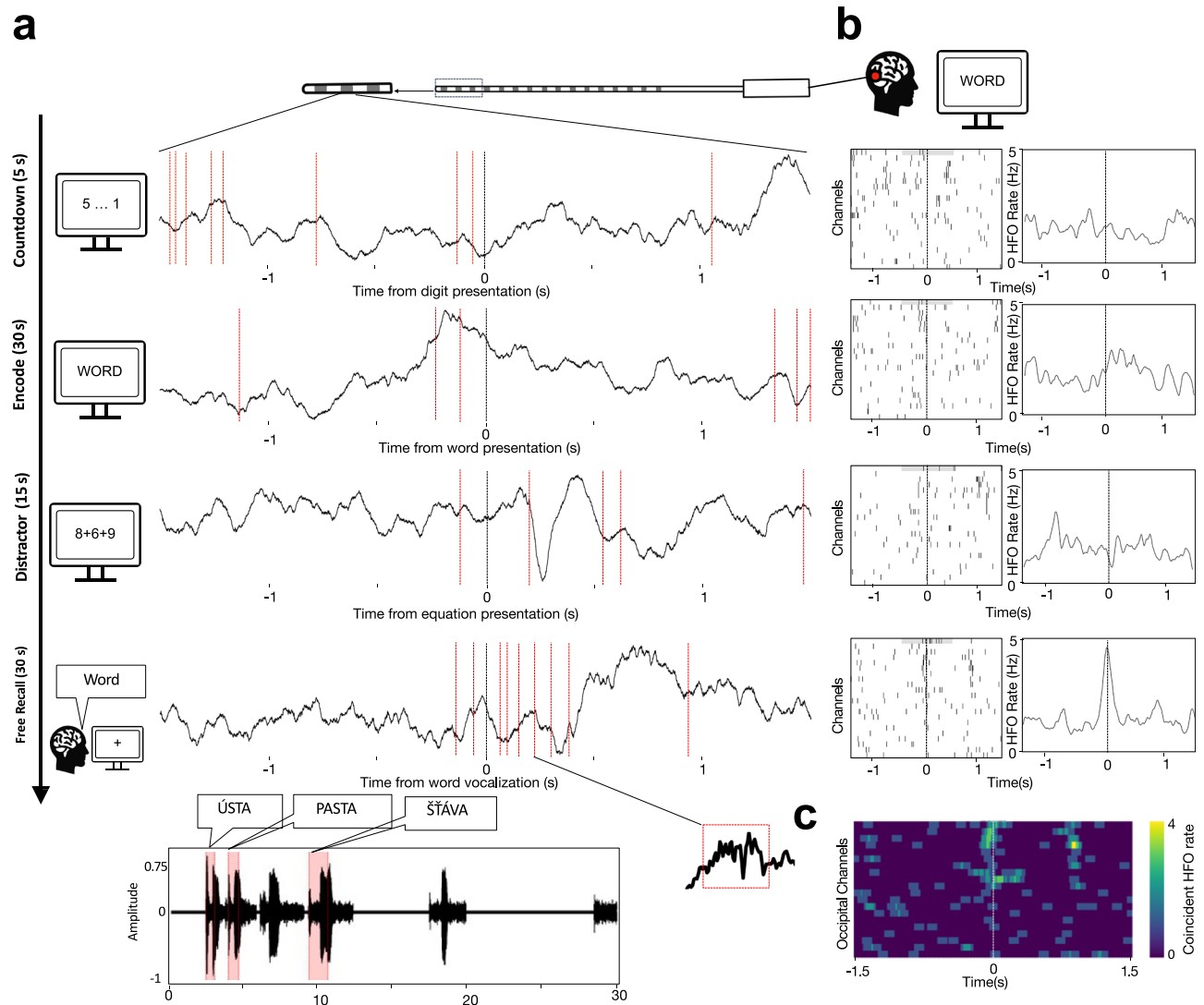

**Fig. 1 | Cortical HFO bursts co-occur during memory encoding and at the time of recall. a** Raw LFP traces from an example macro-electrode contact implanted in the occipital cortex show timepoints when individual HFO bursts were detected (red lines) around the times of presenting a countdown digit, an encoded word, a distractor equation, and of recalling a word. **b** HFOs detected across all channels implanted in the occipital cortex for example traces in (**a**) with a summary of the mean detection rate from the entire session. **c** Matrix of coincident detections in 100-ms time bins from the example recall phase raster in (**b**). Channels are sorted based on the maximum pairwise correlation with other channels to emphasize co-bursting structure. Notice highly coincident bursting among a subgroup of these visual cortical electrodes, even though no visual stimulation is delivered in this phase of the task—patients are freely recalling remembered words with the gaze fixated in the middle of a blank screen.

(1) was related to memory retrieval triggered by the cue, (2) was not linked to preparatory motor activity for word vocalization, and (3) was not specific to FR, although the peak was more prominent in the FR task.

We also compared the co-HFO bursting probabilities during encoding between the words that were subsequently recalled and forgotten. The probability was lower during the inter-stimulus interval, especially after presentation of the previous word and before the moment of presenting words that were later successfully recalled (Fig. 3b). Conversely, it was relatively higher during presentation of the subsequently recalled words on the screen (Fig. 3b). We assessed these differences using the same LMM framework. We observed these subsequent memory effects not only in the grand-average analysis involving all cortical lobes, but also within individual lobes studied in both hemispheres (Supplementary Fig. S5). This pattern suggests that suppression of coincident bursting in preparation for encoding new stimuli, followed by enhanced bursting during word presentation, heralded successful subsequent recall, confirming a role for co-HFOs in memory processing.

## Widespread distribution and low stimulus specificity of the global co-HFOs

Given that the global co-HFO bursting across the cortex is modulated by memory processing, we quantified the proportion of areas engaged in encoding and recalling individual words (Fig. 4a). On average, 45.2% of all implanted contacts were involved in coincident HFO bursting during word recall, and 42.0% during word encoding. We found no significant differences in these proportions between the encoding and recalling phases (Chi-squared test, $\chi^2(1) = 0.70$, $p = 0.999$, $N = 12$). Similarly, all five cortical lobes studied participated in co-HFO bursting in similar proportions (Fig. 4b) with no significant differences (Chi-squared test, $\chi^2(4) = 0.56$, $p = 0.97$, $N = 12$). Thus, both sensory and associational cortical areas were equally involved in the co-HFO discharges.

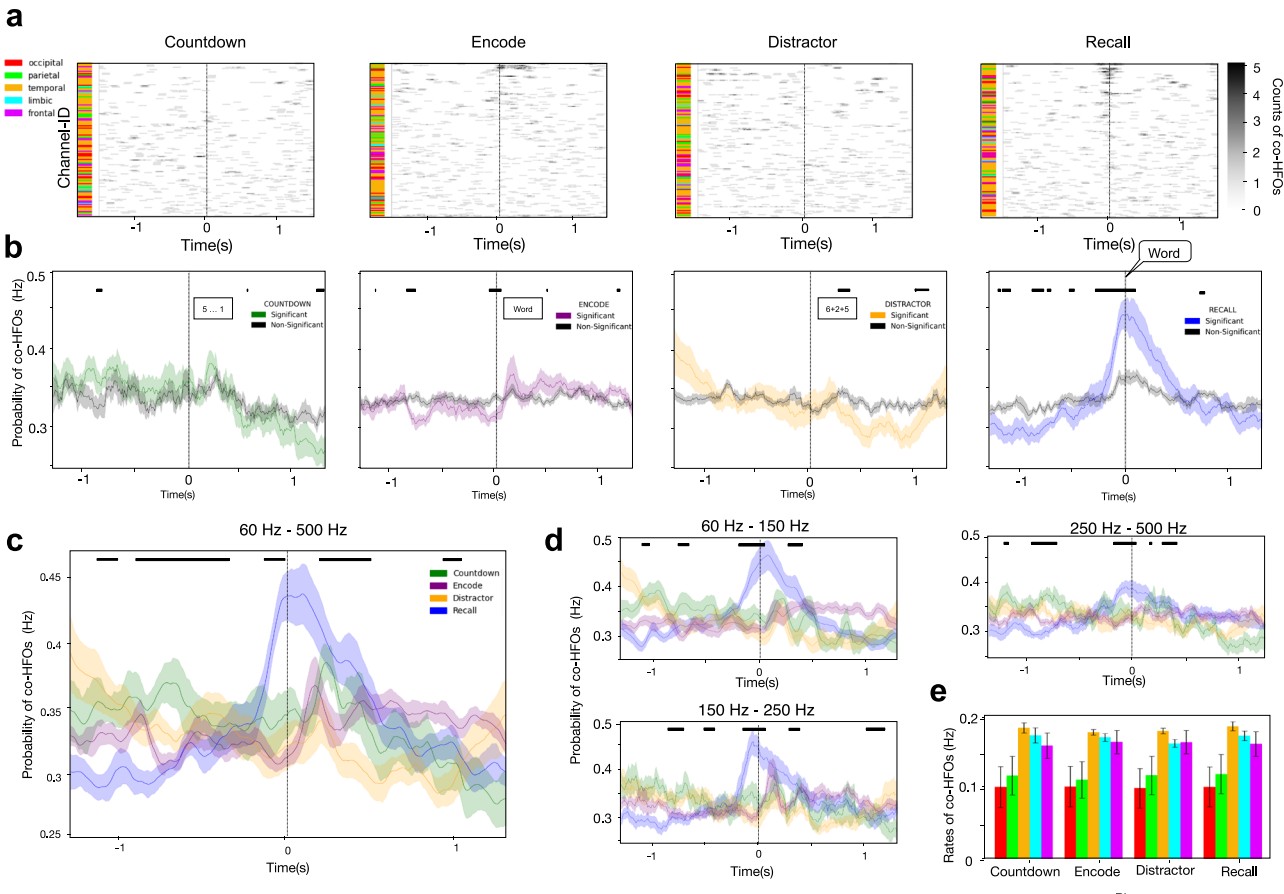

**Fig. 2 | Coincident bursting across sensory and associational cortical areas is modulated by memory processing. a** Matrices of coincident HFO detections from an example patient, plotted as in Fig. 1c, show all channels localized in the five cortical lobes studied in a descending order from the most correlated pairs on top (sorted by maximum pairwise correlations). Channel color indicates anatomical lobe (defined in (**e**)). **b** Probabilities of co-HFO bursting across the correlated channels from all patients reveal times of significantly increased or decreased co-activity relative to the non-correlated channels (shown in gray; asterisks denote a significant difference identified by the linear mixed-effects model (LMM), two-sided $P < 0.05$, FDR-corrected). Shaded regions represent mean ± SEM. **c** Comparison between the four task phases indicates significantly decreased

probability of coincident HFO bursting approximately one second before recall verbalization (blue) with subsequent gradual rise and significantly increased probability a couple of hundred milliseconds before and after the verbalization onset (asterisks denote an LMM-derived significant difference between the four task phases, two-sided $P < 0.05$, FDR-corrected). Shaded regions represent mean ± SEM. **d** A consistent temporal pattern of HFO co-bursting is present in the high gamma, ripple, and fast ripple frequency ranges. Shaded regions represent mean ± SEM. **e** Summary of cortical lobe engagement in HFO co-bursting shows equal contribution in all task phases. Data are presented as mean values ± SEM across $N = 12$ biologically independent participants. No significant differences were found between lobes across phases (One-way ANOVA, $P > 0.05$).

This significantly coordinated bursting was not restricted to local channels within a single cortical lobe; most of the significantly correlated channels were separated by 20–100 mm (Fig. 4c), confirming widespread coordination over distances as large as 200 mm[12]. Interestingly, despite the widespread engagement across cortical lobes in co-HFO bursting, we observed a significantly greater proportion of channels in the occipital (Wilcoxon signed-rank test, $W = 10$, $N = 5$, $p = 0.045$) and parietal (Wilcoxon signed-rank test, $W = 10$, $N = 7$, $p = 0.035$) lobes during the encoding phase compared to the recall phase. Conversely, the frontal (paired $t$-test, $t(9) = 2.89$, $N = 10$, $p = 0.014$) and limbic (Wilcoxon signed-rank test, $W = 14$, $N = 12$, $p = 0.040$) lobes exhibited significantly greater channel involvement during recall (Fig. 4d). This differential activation reflects the expected contributions of more sensory and more associational cortical areas in the two task phases: early processing regions more engaged in memory encoding whereas higher-order cognitive areas more engaged in memory retrieval.

How specific are these globally coordinated cortical networks to particular stimuli? In other words, are the same networks engaged in all instances of memory encoding and recalling, or do they reveal specificity to particular words? We found that the majority of electrode

contacts implanted participated in co-HFO bursting during the encoding or recall of approximately half of the words studied (Fig. 4e). These electrode contacts participated in varying proportions in the encoding and recall of words across the five cortical lobes, with significant differences in four out of the five lobes. The parietal lobe showed the largest difference in the proportion of words between the encoding and recall phases ($t$-test, $t = −26.84$, $N = 12$, $p = 2.05 \times 10^{−151}$, Cohen's $d = 0.63$), followed by the temporal ($t = −40.50$, $N = 12$, $p = 4.4 \times 10^{−208}$, $d = 0.47$), occipital ($t = −10.09$, $N = 12$, $p = 1.06 \times 10^{−23}$, $d = −0.29$), and frontal ($t = −7.07$, $N = 12$, $p = 1.56 \times 10^{−12}$, $d = −0.11$) lobes (Fig. 4f). We observed a considerable overlap between the contacts engaged during encoding and recall of the same words, involving on average more than 10% of all contacts (Supplementary Fig. S6). Despite significant differences between the cortical lobes in both phases, the overall proportions varied within a consistent range of approximately 10–50% of words in the pool for any one contact (Fig. 4g). Hence, at least on the level of macro-contacts used in this study, the global co-HFO bursting patterns were not specific to individual word stimuli but rather showed a general engagement during encoding and recall of the words.

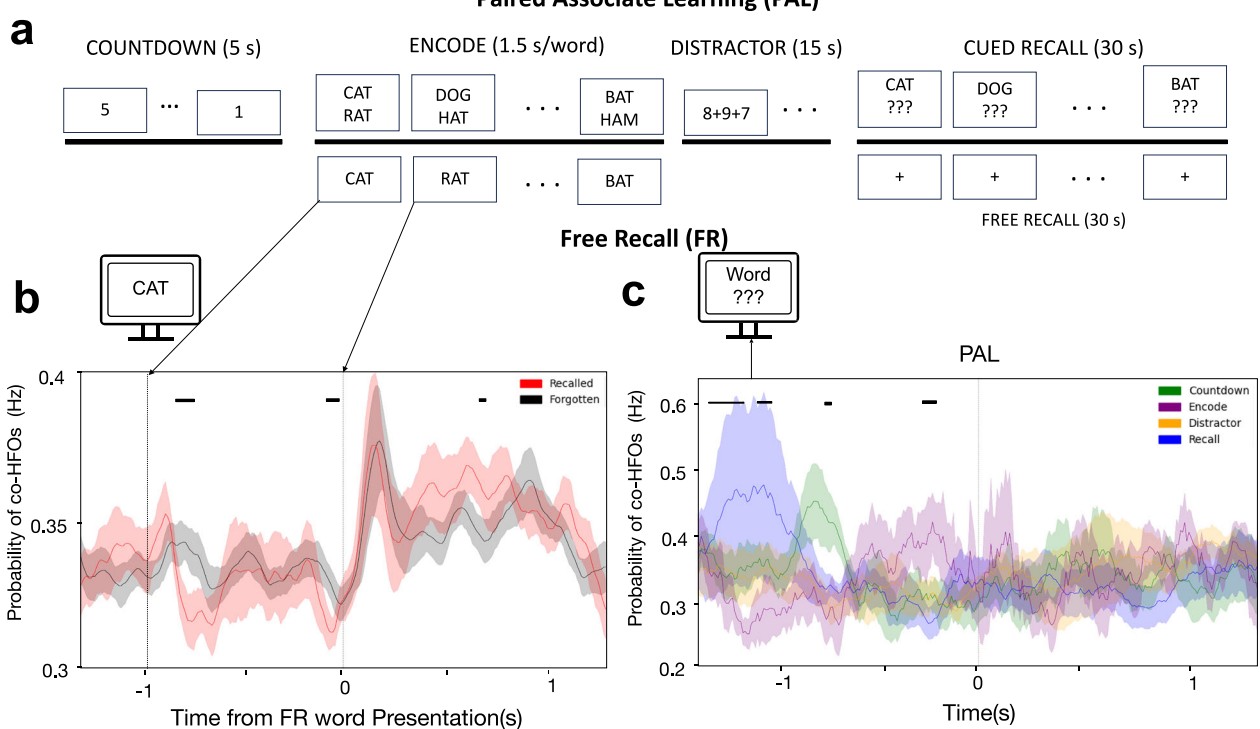

**Fig. 3 | Memory processing modulates coincident HFO bursting during stimulus encoding and cued recall. a** Schematic of a cued-recall version of the task (top), alongside schematics of the free recall (bottom), all using the same pool of words. **b** Probability of co-HFO bursting across the correlated channels from all patients is lower before and higher during presentation of subsequently recalled (red) compared to forgotten (black) words. Black bar indicates significant bins identified by the linear mixed-effects model (LMM) (two-sided $P < 0.05$, FDR-corrected). Shaded regions represent mean ± SEM. Notice the significant difference right before the onset and following the offset of the presented words on the

screen, suggesting that suppression of co-HFO bursting during the inter-stimulus interval is predictive of successful encoding. **c** Analogous probability plots from the cued recall version of the task reveal enhanced co-HFO bursting at the time of the cue presentation and not at the onset of recall vocalization itself. Shaded regions represent mean ± SEM. Notice that the peak of co-HFO bursting occurs much earlier than in the free recall version of the task when the pair-associated words are recalled, separated from any vocalization (black bar indicates bins of significant difference between the task phases identified by the LMM, two-sided $P < 0.05$, FDR-corrected).

## Temporal organization of the global sequential co-HFO bursting

Finally, we asked whether the global co-HFO bursting during encoding or recalling a word is a unitary event or involves multiple related bursting events. We applied hierarchical clustering to HFO burst trains from all contact channels, according to how similar they are across time. This analysis revealed a temporal sequence of multiple bursting events that was not visible when the contact channels were sorted according to the pairwise correlations. The clustering revealed multiple distinct networks of channels bursting in a sequence—one cluster following another in time from the onset to the end of word recall (Fig. 5a). Each cluster comprised channels from multiple cortical lobes and was related to other clusters in a complex tree hierarchy of correlation. Particular clusters showed enhanced co-HFO bursting at distinct moments in time, each preceded by a suppression around one second earlier (Fig. 5a; see also Fig. 2c). This temporal pattern of relatively suppressed co-HFO bursting before recall and induced bursting at the time of recall followed by a long 'tail' of elevated bursting probability fits the profile that we originally observed in the recall phase (Fig. 5a). To confirm that these sequential modules reflected genuine temporal structure rather than spurious correlations, we implemented a surrogate jitter control. We randomly perturbed HFO timestamps within a ±50–100 ms window, preserving overall burst rates while disrupting precise timing. The jittering produced a marked reduction in co-HFO synchrony, measured as the mean pairwise Pearson correlation across channels, indicating that millisecond-scale precision is critical for the observed modular structure (Supplementary Fig. S7).

We further assessed the robustness of clustering using a bootstrap resampling procedure, resampling contact channels independently for each recalled word. The Adjusted Rand Index (ARI) between original and resampled data averaged $0.496 \pm 0.021$ (0 = chance, 1 = identical), suggesting moderate clustering stability across trials.

Encoding trials, although generally exhibiting lower overall co-HFO bursting compared to recall (Fig. 2b), also showed evidence for non-random temporal organization. Surrogate jitter analysis confirmed that these sequences reflect structured timing at the millisecond scale. However, clustering stability during encoding was lower relative to recall (mean ARI = $0.354 \pm 0.026$ vs. $0.496 \pm 0.021$; paired $t(17) = 7.21$, $p < 0.0001$). Synchrony differences after jitter were smaller (paired $t(17) = 4.83$, $p = 0.0002$), indicating weaker temporal coordination.

Forgotten recall trials displayed clustering stability (mean ARI = $0.472 \pm 0.024$) not significantly different from remembered trials (paired $t(17) = 1.32$, $p = 0.20$). Synchrony values are broadly comparable across participants, with no consistent differences.

This successive propagation of co-bursting cortical networks is reminiscent of sequences in place of cell firing reported in rodent studies[43–46]. In the context of our task, one bursting event underlying recall of a word could elicit another event related to recalling associated concepts (Fig. 5b). This would give rise to a sequence of successive co-HFO bursting events. We have recently proposed such sequential discharges of cortical HFO bursting on macro- and micro-scales for tracking electrophysiological engram activities in humans[21].

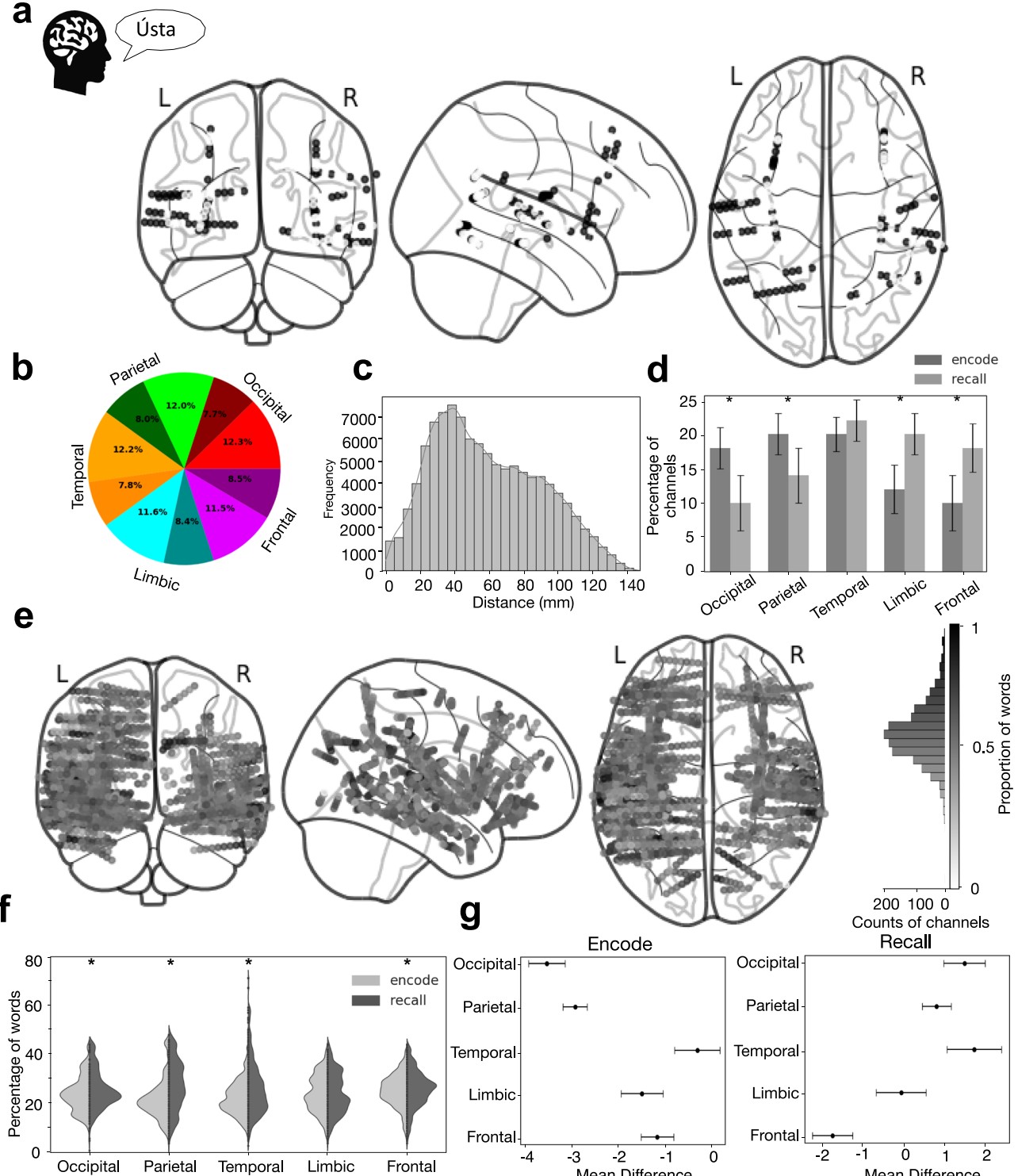

**Fig. 4 | Widely distributed networks of coincident bursting are involved in encoding and recalling words. a** Localization of electrode channels that displayed coincident HFO bursting (black dots) during recall of an example word (same word in Figs. 1 and 2) is distributed throughout the cortex, engaging approximately half of the contacts implanted. **b** Proportion of all channels showing significantly correlated (bright shades) and non-correlated (dark shades) co-HFO bursting averaged across all patients reveals equal contribution of all five lobes (color-coded as in Fig. 2). **c** Anatomical distances between the correlated electrode channels (Pearson's correlation, $r > 0.50$) show the greatest density between 20–100 mm. **d** Correlated channels distribution is significantly higher during encoding in occipital ($P = 0.045$) and parietal ($P = 0.035$) lobes, and higher during recall in limbic ($P = 0.040$) and frontal ($P = 0.014$) lobes (two-sided Wilcoxon signed-rank or paired

$t$-tests; $N = 12$ biologically independent participants). **e** Summary of the percent involvement of any one electrode contact (dot) in co-HFO bursting across the entire pool of words reveals relatively low specificity, with most channels engaged in more than half of the words recalled. **f** Violin plots of co-HFO bursting proportion for encoded (light) vs. recalled (dark) words. Significant phase differences found in Parietal ($P = 2.05 \times 10^{-151}$), Temporal ($P = 4.4 \times 10^{-208}$), Occipital ($P = 1.06 \times 10^{-23}$), and Frontal ($P = 1.56 \times 10^{-12}$) lobes (two-sided independent $t$-tests, $N = 12$). Center white dot: median; thick bar: interquartile range (IQR). **g** Tukey's post-hoc analysis compares the mean proportions from (**f**) across the five cortical lobes in the two task phases. Significant differences (indicated by asterisks, $P < 0.05$) were found between phases in Parietal ($P = 2.05 \times 10^{-151}$), Temporal ($P = 4.4 \times 10^{-208}$), Occipital ($P = 1.06 \times 10^{-23}$), and Frontal ($P = 1.56 \times 10^{-12}$) lobes.

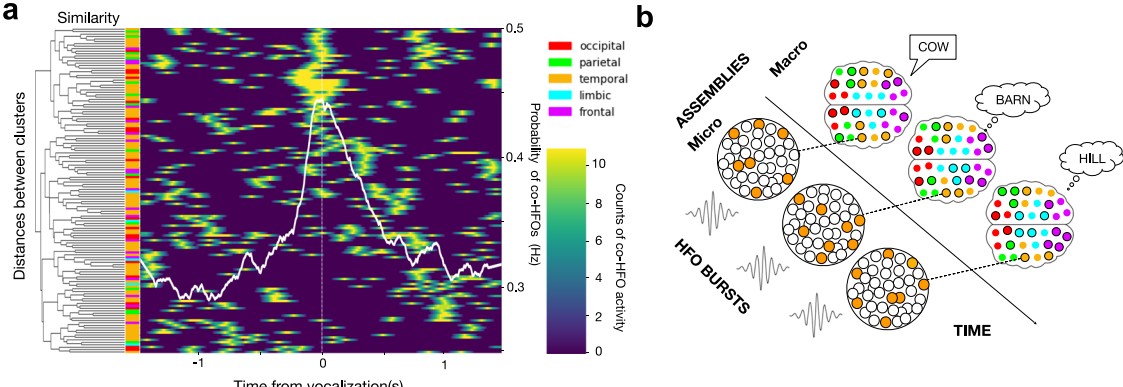

**Fig. 5 | Global cortical bursting is temporally organized into sequences underlying word recall. a** Heatmap of hierarchical clustering of co-HFO bursting during recall of an example word reveals distinct sequential discharges, each involving multiple cortical lobes and aligning with the overlaid grand-average pattern of co-HFO probability from Fig. 2c (white line). Notice that the sequence of discharges along the diagonal of the heatmap is preceded by a parallel diagonal of co-HFO bursting suppression approximately 1 s before each discharge. **b** Schematic of the model illustrates how local neuronal assemblies that generate the global sequences of co-HFO bursting can account for recall of a word followed by related concepts.

## Discussion

In this study, we took a global perspective beyond the more focal framework of hippocampo-cortical ripples to explore wider scales of anatomical, temporal and spectral organization of coincident HFO bursting. Our results show that bursts of high-frequency oscillations co-occur over a large anatomical scale of multiple sensory and associational cortical areas. These coincident bursting happens across all phases of performing a task, but their occurrence is strongly modulated by memory recalling and encoding. This co-HFO bursting is suppressed before and enhanced during presentation of the stimuli that will subsequently be recalled, compared to those that will be forgotten, and peaks approximately 300 ms before the recall vocalization. Around half of the recorded sites from all cortical lobes are engaged in this global coordination of HFO bursting underlying encoding and recalling the remembered stimuli, which was not specific to particular areas or words. Multiple distinct cortical networks were coordinated together in a sequence of bursts when recalling a single word. Instead of one bursting network, our results suggest a sequence of temporally coordinated networks. These findings suggest possible roles of globally coordinated discharges of high-frequency oscillations in organizing communication and integration of information for memory processing. Our study offers a large-scale perspective on the mechanism and functions of coincident HFO bursts, bridging previous investigations of hippocampal and cortical ripple HFOs, global cortical networks, and their role in human memory and cognition.

### Memory processing or general integration of information?

Globally coordinated HFO bursting can be particularly important for memory processing, even though it is ubiquitous throughout all memory and non-memory related task phases. Hippocampal ripple HFOs were shown to co-occur with cortical spindles and slow oscillations to mediate transfer and consolidation of memory traces in humans[47]. More recent studies found that hippocampal–cortical ripple HFOs co-occur during sleep and waking periods and are especially prominent in memory recall[12,18]. Our results confirm the maximum occurrence of coincident HFO bursting around the onset of recall vocalization. We found that the significantly greater occurrence compared to stimuli presentations in the other task phases started 300 ms before the vocalization onset and was significantly suppressed approximately 1 s before the onset (see Fig. 2c). This detailed temporal timeframe is in agreement with the expected dynamics of recalling an item—first a suppression of bursting in response to the previously

recalled item would be followed by a gradual build-up of activity corresponding to recalling a new item. The timing resembles the 'readiness potential' from Libet's original experiments[42], which was more recently also demonstrated at the level of single unit firing[48]. While this study was not designed to address the emergence of conscious volition or its neural substrates[49], the timing of this peak in coincident HFO bursting preceding free recall at least suggests that it is related to memory processing and not to other processes like preparation or planning for vocalization. This was confirmed in the additional cued recall paradigm, which showed a shift of the peak co-HFO bursting to 1000 ms before word vocalization (see Fig. 3a) around the time of cue presentation.

Furthermore, it was not only memory recall but also memory encoding that induced more coincident HFO bursting after stimulus presentation than in the other task phases. There was an increased occurrence starting with a local peak at 150–200 ms after stimulus onset, followed by an elevated rate throughout the time of word presentation on the screen, which corresponds to the expected information processing timeframe that we observed in the same paradigm across the ventral visual stream[50,51]. Like in the recall phase, the increased occurrence was preceded by a suppression in co-HFO bursting with one dip following presentation of the previous word and another right before the onset of a new word presentation. Word stimuli that showed less of this suppression before and less enhancement after presentation were more likely to be subsequently forgotten (see Fig. 3b). Altogether, these results point to an important role of the globally coordinated HFO discharges not only around memory recall but more generally in memory processing.

Previous studies proposed an even more general role for the global HFO bursting in integrating information processing across the brain[18,31], which would provide a solution for the binding problem[1,9,52]. Such a role would apply both to memory, attention, and perceptual functions[2,6,8,53]. In agreement with a more general role, our results show internally induced co-HFO bursting in the occipital cortex during recall in the absence of any external visual stimulation (see Fig. 1). On the other hand, visual stimulation with countdown digit or algebra equation presentations during the preparatory and the delay phases of the task, respectively, induced a smaller increase in co-HFO occurrence (see Fig. 2b), even though both were perceived and engaged subjects' attention in the task. One could argue that these other, less relevant stimuli were not as attended to or required as much cognitive effort as memorizing or recalling items. In other words, perceiving countdown

digits or solving algebra is more automated compared to intentionally focused formation of new memory traces and especially their internally-driven retrieval, which would arguably require more global cortical coordination. It would explain why cued recall induced a smaller rise in the co-HFO bursting than the free recall (see Fig. 3a). Dissociating specific roles in particular cognitive functions would necessitate a more appropriate experimental paradigm.

## Beyond hippocampal sharp-wave ripples, memory consolidation

Still, high-frequency oscillations, especially in the ripple frequency range, have been specifically associated with memory processing, given the link with hippocampal sharp-wave ripple complexes[11,13,14,19,29,30,33,39,54,55]. Bursts of oscillations in gamma and ripple frequencies are ideally suited for inducing synaptic plasticity[56] and transferring information in 'packets' across the cortex[57], congruent with the proposed roles in integrating and storing information in memory. In human studies, identifying and defining a particular electrophysiological activity corresponding to the rodent sharp-wave ripple complexes is challenging due to a lack of clear detection criteria[21,25], including precise anatomical localization, presence of a sharp wave in addition to an oscillation above a particular power threshold and a minimum number of cycles in the unfiltered signal. Hence, based on our previous studies[14,15], we took a more general approach, detecting distinct oscillations across a broad frequency range and all cortical locations with no need for isolating hippocampal sharp-wave ripple complexes. Rodent studies showed that high gamma/epsilon and ripple oscillations in the hippocampus shared common neuronal ensembles and mechanisms[22,58,59], despite clear differences even among ripple detections[60]. Using a similar threshold for detecting oscillations of at least four cycles, we found that the global co-HFO bursting is not limited to a narrow 'ripple' frequency range but involves oscillations across gamma, ripple, and fast ripple ranges. The same pattern of induced coincident bursting before recall onset was present in 60–150 Hz, 150–250 Hz, and even 250–500 Hz, although the peak was gradually fading with increasing frequency of the detected bursts (see Fig. 2c). Interestingly, at these higher frequencies there is an increased likelihood of detecting pathological HFOs, as proposed in the animal models of pathological and physiological HFOs[61] (our analyses were limited to HFO detections, which are temporally related to memory processes in the tasks). The reported patterns of co-HFO bursting were also not any stronger in the limbic cortex, which included the hippocampus, than in the other cortical lobes (see Supplementary Fig. S5). Therefore, it is plausible that the global cortical bursting may have a more general role than the one previously proposed for hippocampal-cortical dialog in human memory encoding, consolidation, and retrieval[13,16,17,19].

We have limited our study to bursts of high-frequency oscillations above the classic Berger bands. Similar discharges, however, can be detected in the lower frequencies. A recent study by Sieber et al. reported hippocampal 'bouts' of theta frequency oscillations preceding a turn in real-world or imaginary navigation in epilepsy patients[62]. The theta 'bouts' were on average four cycles long, they were not related to motor activities when making a turn, and were used to reconstruct spatial locations even in the imaginary navigation. Similar bouts, bursts or, more generally, discharges of approximately four cycles of oscillations were also found in alpha and beta frequencies in that study, which suggests a universal organization of cortical oscillations. How the high and the low frequency activities are related to each other in time and anatomical space[51,63,64] remains to be explored on the level of individual oscillatory discharges, but a common role emerges. Whether it is freely recalling word concepts or imagining spatial locations, these abstract cognitive processes are preceded by the discharges of low and high-frequency oscillations.

## Tracking dynamic engram activities of large-scale neuronal assemblies?

What do these global coincident bursts reflect? We have recently proposed HFOs as electrophysiological biomarkers of engram activities[21]. Individual bursts are generated by neuronal assemblies that coordinate their firing when processing information about a particular item or its features. Coincident bursting across the cortex can integrate this processing in large-scale networks of assemblies processing features of related items. Our results suggest that these large-scale networks of assemblies underlying encoding or recall of any one item are widespread, involve around half of the recording channels, and are not highly specific to any particular words (see Fig. 4). This picture is similar to a map of all cells activated in response to encoding or recalling a single memory trace—an engram of episodic memory in mice, which engaged up to half of all the brain regions studied[38]. To spot differences between such large-scale maps of two separate engrams for episodic memory traces would be possible on the level of single activated cells, but challenging on the level of entire brain regions. That is likely why our macro-contact recordings resulted in responses to a large proportion of word items and relatively high overlap between them. Using micro- or meso-contact recordings and study designs presenting the same words multiple times would be expected to resolve more specific engram contributions. There are probably multiple distinct neuronal assemblies generating HFOs corresponding to different words that would be detected on the same macro-contact or distinct meso- or micro-contacts. A previous study by Vaz and colleagues confirmed that detecting single neuron firing on micro-contacts in parallel to HFO bursts on macro-contacts can be used to decode correct and incorrect recall of specific words from sequences of single cell firing[10]. The same group, more recently, showed that this temporal organization of unit firing carries specific information about the remembered stimuli[37]. Our results suggest an analogous temporal organization of global co-HFO bursting on a macro-scale. In this study, we were limited to relatively low numbers of micro-electrode contacts, which prevented us from detecting micro-scale co-HFO discharges to specific words. Future studies employing combined macro- and micro-scale electrophysiological recordings[24,65] and study designs adjusted to detect the hypothetical engram activities will not only track them with high spatial resolution but also in time to follow dynamically changing neuronal assemblies[66]. Direct electrical stimulation to evoke engram activities[67,68] at macro- and micro-scales offers causal evidence for these hypotheses.

Although our study was limited to macro-contact recordings, we also observed two levels of organization for coincident HFO bursting. On the first level, HFOs generated by local neuronal assemblies were coordinated across the cortex in global coincident bursting, as is now increasingly reported in the most recent studies[16–18,20]. This level can basically be described as a global network of local assemblies. The second level that emerged from our results comprises a larger network of multiple distinct global bursting networks, which are coordinated together in a temporal sequence (see Fig. 5). This macro-scale 'sequence of networks' could be compared to the micro-scale 'synfiring chains'[69], in which firing trains of one neuron or one assembly of neurons corresponding to the first item triggers another train corresponding to the second item, then the third, and so on. In summary, there is a 'micro' level of coordinated sequential firing of neuronal assemblies giving rise to individual HFO bursts, and a 'macro' level of sequentially coincident bursting in a higher order global network organization, as we have recently proposed[21]. Notably, these co-HFO bursting networks did not activate simultaneously but sequentially within recall episodes, indicating a structured temporal organization rather than random or purely synchronous co-activation. This pattern argues against non-specific epileptiform synchronization and suggests temporally coordinated cortical activity. The 'macro' level sequences of the global coincident bursting remain to be further investigated.

## Roles and applications of global co-HFO bursting

There are other outstanding questions about the global HFO bursting that need to be addressed in future studies. First of all, can this wide-spread temporal coordination of local neural activities provide a mechanism to address the binding problem [1,52]? Recent studies suggest so[12,18,31], but testing it would require an appropriate behavioral paradigm probing conscious awareness of the perceived stimuli. Neural activities in the high frequency ranges (above 50 Hz) were not only associated with encoding and recall of human memory[14,70,71] but also with explicit awareness of visual stimuli in monkeys[72] or conscious dream experiences in humans[73]. From the perspective of these studies, coordination of HFOs across the brain in coincident bursts could be linked to global integration of information and thus provide a plausible neural mechanism for concepts like 'ignition' or 'global broadcasting' proposed in the global neuronal workspace hypothesis[74]. At the same time, we acknowledge that such dynamics may also be compatible with other theoretical frameworks, including predictive processing and neurorepresentationalism, recurrent processing theory, and information integration theory; while our study does not adjudicate between these accounts, the present findings may inform multiple perspectives on the neural basis of consciousness[75]. Even though it is speculative at this point, one could nevertheless ask whether the sequences of bursting events correspond to a train of thought or a stream of consciousness. In the case of our paradigm, which is not designed to address these questions, we can only discuss potential future directions for testing the roles of co-HFO discharges in these functions; the sequences reported can, however, be explained more concretely in terms of a model for recalling engram representations of related word concepts from the memorized lists (see Fig. 5). Coincident HFO bursting presents a testable activity for these and other models of cognitive functions, including perceptual binding, conscious awareness, or engram representations of related concepts in the human mind.

Having a concrete electrophysiological activity reflecting abstract processes in the human mind would enable not only tracking across the brain and time but also modulating them with targeted brain stimulation in brain disorders. Memory and cognitive deficits are hall-marks of neurological and psychiatric diseases with limited treatment options. Emerging technologies for chronic brain recording, analytics and stimulation using distributed cloud computation and machine learning tools can now deliver personalized adaptive modulation therapies for restoring movement[76], mood[77], and memory functions[78]. Biomarkers of electrophysiological activities like the HFO bursting promise to guide, monitor and assess new neuromodulation therapies across the anatomical brain regions and time.

## Methods

### Participants and electrode contact localization

Seventeen patients (10 males, mean age $31.94 \pm 1.63$ years) undergoing intracranial stereo EEG seizure monitoring for epilepsy surgery at St. Anne's University Hospital in Brno and Jan Mikulicz-Radecki University Hospital in Wroclaw participated in this study, which was approved by the local Institutional Review Board ethical committees (see Table 1). Participants were native speakers of Czech ($n = 8$), Slovak ($n = 4$), or Polish ($n = 5$), and all provided written informed consent (see Table 1). Electrode implantation was dictated by the clinical priority of treating drug-resistant seizures, leading to a non-uniform distribution of electrode contacts across all cortical areas (see Suppl. Table 2). We excluded data from five patients from further analysis due to technical issues related to the accurate synchronization of the task events with the electrophysiological recordings, which were critical for this study. Following implantation, we co-registered pre-operative high-resolution CT and post-operative MRI scans, normalized to determine electrode contact locations in MNI space and assign anatomical labels according to the Automated Anatomical Labeling atlas (AAL3)[79]. We grouped electrode contacts into distinct cortical lobes: occipital, parietal, temporal, frontal, and mesial temporal (referred to as 'limbic'). The limbic lobe included the hippocampus, amygdala, posterior cingulate cortex and parahippocampal cortex. In total, we recorded 3,239 implanted contacts across all patients (2957 macro contacts).

**Table 2 | Summary of clinical profiles and seizure-related data**

| Dataset | ID | SOZ channels | SOZ structures | MRI Findings | Engel Score |
|---|---|---|---|---|---|
| BR | 2 | – | – | Normal | – |
| BR | 3 | B1–4, C1–3, B´1–5, C´1–3 | Hippocampus, Parahippocampal gyrus | Normal | – |
| BR | 5 | G´1–2, R´1–6, S´1–3 | Cingulate gyrus, Precuneus, Superior occipital gyrus | Normal | IVB |
| BR | 6 | G´1–4, H´7–11 | Middle temporal gyrus | Normal | IV |
| BR | 7 | Y´2–10, I´1–4 | Insula | Venous angioma, cavernoma parieto-occipital left | IV |
| BR | 8 | I´1–6 | Paracentral lobule, Postcentral gyrus | Polymicrogyria postcentral sulcus left | IIIA |
| BR | 9 | – | – | Normal | – |
| BR | 10 | B1–3, C1–3, B´1–3, C´1–5 | Hippocampus, Parahippocampal gyrus | Normal | – |
| BR | 11 | Y´1–4, I´1–5 | Insula | Normal | – |
| BR | 13 | B´1–4, C´1–4 | Hippocampus | Posttraumatic changes temporo-occipital left | – |
| BR | 14 | Y1–4, X1–3 | Insula | Normal | – |
| BR | 16 | B1–2 | Parahippocampal gyrus | Postsurgical changes after right anterior mesial temporal resection (AMTR) | – |
| WR | 1 | AMG1–2, TP1–2 | Parahippocampal gyrus | Cortico-subcortical blurring temporal pole, right | IB |
| WR | 2 | AMG1–2, HB1-2, TP1–2 | Amygdala, Hippocampus, Temporal Pole | Cortico-subcortical blurring temporal pole left | IIIB |
| WR | 3 | SMADL1–2 | Cingulate gyrus | Cortico-subcortical blurring cingulate gyrus, left | IA |
| WR | 4 | AMG1–3, TP1–2 | Amygdala, Hippocampus, Temporal Pole | Cortico-subcortical blurring temporal pole left | IB |
| WR | 5 | HB1–2, HH1–2, TP1–2 | Hippocampus, Temporal Pole | Cortico-subcortical blurring temporal left | – |

Clinical information for all participants, including dataset identifiers, seizure onset zone (SOZ) channels and corresponding anatomical structures, MRI findings, and postoperative seizure outcome when available (Engel classification).

The distribution of contacts participating in co-HFO bursting during word recall is shown per subject in Supplementary Fig. S8.

All participants recruited in this study were patients suffering drug-resistant epilepsy with various clinical backgrounds of the disease and anatomical localizations of the seizure onset zones (see Table 2). Hence, the intracranial recordings in these patients were inherently limited by the pathological discharges, including the interictal epileptiform spikes and HFOs, especially in the fast ripple frequency ranges. We cannot fully exclude a potential influence of these epilepsy-related activities on our methodology or the reported results. However, our design was based on detecting HFOs consistently around cognitive events associated with memory processing, which would be expected to average out any potential pathological activities. We previously reported an interaction between memory processing and incidence of epileptiform discharges in these patients, showing a lower probability of their occurrence during memory processing[80]. Also, most coincident discharges reported comprised HFOs detected at frequencies lower than 250 Hz, which is thought to be a lower threshold for pathological oscillations[61]. Overall, pathological discharges are not expected to be consistently induced by memory processing and analyses across large numbers of trials should effectively diminish any potential influence of their occurrence. Nevertheless, it can never be completely ruled out when studying any physiological function of the brain.

## Experimental design

Participants completed a battery of verbal memory tasks designed to examine neural dynamics underlying memory encoding, maintenance, and retrieval of words. The main free recall (FR) task involved presenting 12-word lists derived from common nouns in the participant's native language (Czech, Slovak, or Polish), sourced from a publicly available repository (http://memory.psych.upenn.edu/WordPools). In total, 180 words from the pool were selected based on their translatability to the three Slavic languages and randomly assigned to 15 lists before the start of a task run. Each run began with a 5 s countdown presentation of digits from 5 to 1, immediately followed by a presentation of a word list for encoding. Each word from a list was displayed for 1500 ms with 1000 ms inter-stimulus interval. Following the final word, participants performed an arithmetic distractor task (e.g., A + B + C =?, where A, B, and C were random integers between 1 and 9) to prevent rehearsing of the list. After 20 s of the distractor, participants had 30 s to verbally recall as many words as possible in any order. We recorded verbal responses using high-fidelity microphones and annotated them for accuracy with an automated transcription method, followed by supervision and post-hoc verification (see Supplementary Fig. S2). The transcription used a custom interface integrating a fine-tuned Whisper model. To achieve precise word alignment, we applied a timestamped correction using Dynamic Time Warping. This algorithm aligned predicted text with actual speech patterns, correcting timing discrepancies and improving synchronization.

The transcription method automatically marked the precise onset times of verbal responses, which we manually corrected when necessary (see Supplementary Fig. S2) to mark the very onset of the first sounds of word verbalization. All other task events were automatically time-stamped in the electrophysiological neural recordings via TTL pulse generation from the task computer.

In addition to the FR task, participants completed an analogous cued recall task, Paired-Associate Learning (PAL). Instead of the twelve-word lists, six-word pairs were sequentially presented with one pair after another. After the same arithmetic distractor task as in the FR task, we presented participants with a cue of one word from each pair, prompting them to recall its corresponding paired associate. The PAL task also comprised 15 lists of randomized word pairs drawn from the same word pool as the FR task. Some patients completed an additional audio version of the FR and PAL tasks with the presented words and the recall cue word played out loud with a fixation cross on the computer screen. The open-access datasets are continually updated with additional patient recordings for analysis of these tasks[81].

## Electrophysiological recordings

We acquired electrophysiological recordings using the ATLAS neurophysiological system (FHC-Neuralynx Inc.) from up to 128 channels recorded simultaneously at a 32,000 Hz sampling rate (Jan Mikulicz-Radecki University Hospital in Wroclaw), and the BrainScope BioSDA09 system (M&I Ltd.) from up to 192 channels at a 5000 Hz sampling rate (St. Anne's University Hospital in Brno). The signals were recorded from multiple standard depth electrodes (2.3 mm exposed contact surface, 5–10 mm inter-contact spacing) for a stereo EEG surgical procedure (AD-Tech Inc.) implanted for prolonged seizure monitoring. Additional subgaleal electrodes served as reference contacts. To enhance spatial specificity and minimize volume conduction artifacts, we re-referenced signals using a bipolar montage applied to adjacent contacts within each stereotactically implanted depth electrode. Another montage option for HFOs detected across widespread brain areas is a unipolar montage referenced to two nearby contacts in the white matter[82,83]. However, we did not use this approach because of some co-HFO bursting detected in the white matter contacts (Supplementary Fig. S9).

We stored all electrophysiological time-series data in multiscale electrophysiology format (MEF, version 3.6) and structured in accordance with the Brain Imaging Data Structure (BIDS) format, to ensure universal accessibility and reproducibility[40]. We anonymized all participant data with personally identifiable information removed in adherence to ethical guidelines. The complete dataset is publicly available on the EBRAINS portal[84].

## Detection of individual HFO bursts

We detected individual bursts of the high-frequency oscillations (HFOs) by decomposing the signal into 38 logarithmically spaced frequency bands (60–800 Hz) using zero-phase finite impulse response filters with 200 ms transition bands and a −6 dB cutoff[14,15]. We extracted the amplitude envelope for each band using the Hilbert transform and normalized via a sliding z-score transformation, with the mean and standard deviation computed over 10-second windows. We initially identified candidate HFOs when the z-scored amplitude exceeded a liberal threshold of 2 standard deviations (SD) for at least three consecutive oscillations. This initial threshold served only to identify candidate segments for further evaluation, primarily to localize the start and end boundaries of putative events. We did not directly include these initial events in subsequent analyses. We merged events spanning multiple frequency bands, and for each detected event, we determined the dominant frequency, defined as the band with the highest z-score. We also extracted the maximum amplitude within the event window to characterize peak activity.

To enhance detection specificity and remove artifacts, we implemented a dual-threshold detection approach. First, we required candidate events to exceed a peak z-score of 3 SD. Second, we retained only those lasting at least four complete oscillatory cycles at their dominant frequency. To further ensure that detected events represented true oscillatory phenomena rather than transient spectral fluctuations, we performed a final cycle-count verification, requiring at least one complete oscillation above the threshold for classification as an HFO burst. This approach reduced false positives in the human intracranial recordings of physiological HFOs. These false positives can pose challenges to the detection and differentiation from pathological epileptiform activities or non-oscillatory increases in the power or amplitude envelope due to filtering of sharp transitions in the signal and non-physiological artifacts[25]. Even the detection of the classic hippocampal sharp-wave ripple complexes in rodents prompts the development of more advanced classification criteria[60].

To assess event-level specificity, we additionally quantified spectral and temporal features (peak frequency, frequency span, duration, amplitude) of all detections and applied a frequency span–based k-means clustering (k = 2) to exclude spectrally diffuse events of larger frequency span (8–15% rejected; See Supplementary Fig. S1). Analyses restricted to the remaining high-specificity subset yielded results consistent with those reported (see Supplementary Fig. S1).

## Analysis of coincident HFO bursting

We represented HFOs as discrete points at the peak amplitude in frequency-time space[41], effectively quantizing each burst into discrete events with intrinsic features such as onset time, offset time, maximum frequency, and minimum frequency. We computed the occurrence rate for each phase (countdown, encode, distractor, recall) across three frequency ranges: 60–150 Hz, 150–250 Hz, and 250–500 Hz (see Supplementary Fig. S1). We constructed raster matrices to visualize the temporal distribution of the HFO detection events, using a 10-ms bin size. This bin width was selected based on both the typical HFO duration (10–50 ms) and empirical comparisons across multiple resolutions (5, 10, 50, and 100 ms). Binning at 5 ms produced overly sparse representations, whereas 50–100 ms bins excessively smoothed temporal patterns. The 10-ms resolution provided the clearest and most stable co-HFO structure across trials. We defined time zero for each task phase at digit presentation (countdown), word presentation (encode), arithmetic task onset (distractor), and onset of remembered word vocalization (recall). Rasters were aligned to these event times within a ±1.5 s window.

To assess spatiotemporal relationships, we binarized HFOs (1 for HFO presence, 0 for absence) and examined a ±50 ms window (±5 bins) across channels to identify coincident HFO bursts (co-HFOs), also called co-HFO bursting events. Although our primary approach used discrete event representations, future analyses using continuous, time-frequency methods such as wavelet-based decomposition may yield additional insight into finer-scale HFO dynamics. We defined a co-HFO as an HFO occurring within a ±50 ms window of another HFO on a different channel, ensuring that only temporally proximate events were considered functionally related. We computed Pearson correlation coefficients between channels for each raster, classifying channels as correlated ($r > 0.5$) or non-correlated ($r \leq 0.5$), and sorted channels into two groups based on the correlation values (see Fig. 2a). We selected this threshold to prioritize strong functional relationships while mitigating the impact of noise and weak correlations. We sorted channels based on the maximum value of pairwise Pearson correlation coefficients of any one channel. For each channel, we calculated the coefficient values for all possible pairwise combinations with the other channels. We then took the maximum values of each channel and sorted them in descending order. In Fig. 2a, the most correlated pair is followed by other channels with gradually smaller correlation values. The color axis in Fig. 2a denotes the anatomical lobe of each channel to show that the top channels come from all cortical lobes studied, not just the neighboring channels from one lobe. The correlation coefficients were not plotted at all in these plots—only used for sorting the order. The emergent pattern of co-HFO distribution across brain regions and over time was therefore not the effect of any 'temporal' sorting by the latency values, but the highest correlation only. One of the first conclusions is that the most correlated channels are coincidentally bursting at the same time, and this occurs consistently around memory recall and during memory encoding.

We evaluated all electrode pairs symmetrically based on their correlation values and co-HFO coincidence rates. No single channel was designated as a 'source'; instead, group-level comparisons were derived from subject-wise averaging of all channel pairs within each correlation class ($r > 0.5$ vs. $r \leq 0.5$). We calculated the correlation metric based on overall HFO binarized activity patterns, while co-HFOs were defined by finer-grained temporal overlap within a ±50 ms

window, ensuring that these two measures remained conceptually and computationally distinct. We repeated key analyses using alternative correlation thresholds ($r > 0.3$ and $r > 0.7$) to examine the effect of this parameter choice. We selected $r > 0.5$ threshold to prioritize moderately strong correlations that are more likely to reflect meaningful co-variation in HFO activity, without excluding a large proportion of relevant channel pairs. This criterion provided a practical balance between interpretability and signal reliability across subjects.

To quantify functional connectivity dynamics, we divided channels into positively correlated ($r > 0.5$) and non-correlated ($r \leq 0.5$) groups, and we computed the rate of co-HFOs per time bin separately for each group. This approach allowed for a direct comparison of co-activation patterns between correlated and non-correlated channels over time (see Fig. 2b). We averaged co-HFO rates across all detected events within each task phase. To account for inter-subject variability, we first computed the mean co-HFO rate for each subject at each time bin. To enable standardized comparisons across subjects, we then applied probability normalization, ensuring that the sum of all co-HFO rates across all subjects and time bins equaled 1. This transformation yielded a traditional probability distribution, where each value represented the probability of a co-HFO occurring at a given time bin relative to the entire dataset. For visualization purposes, we scaled the normalized values by a factor of 100 to express probability per second in figures, given the 10-ms bin size. Importantly, we applied the normalization after rescaling to a per-second probability, allowing for standardized comparisons across subjects and task phases. The resulting probability of co-HFO occurrence provided a standardized measure of how co-HFOs evolved over time relative to task events (see Fig. 2c). Importantly, we compared co-HFO dynamics between the paired-associate learning (PAL) and free recall tasks within the same subjects, using the same electrode contacts on the same recording day. This comparison was limited to the subset of participants who completed both tasks. This within-subject, within-session design ensures that observed differences in co-HFO timing reflect task-specific retrieval demands rather than inter-individual or session-related variability. We extended our coincidence analysis to compare the temporal dynamics of co-HFOs between subsequently recalled and forgotten words (see Fig. 3b) and across task phases in the paired-associate learning (PAL) task (see Fig. 3c). These comparisons allowed us to assess whether functional network engagement varied as a function of recall success and task structure. We further examined regional specificity by assessing co-HFO activity within individual cortical lobes to determine whether co-HFOs followed regionally distinct activation patterns (see Supplementary Fig. S5).

For each patient, we identified electrode contacts involved in correlated bursting using co-HFO matrices as in Fig. 2a. These rasters provided structured outputs from burst detection analyses, listing all implicated contacts and their corresponding cortical lobes. We extracted contacts from correlated HFO bursts and assigned lobe identities using the Automated Anatomical Labeling Atlas (AAL3)[79], mapping each contact to its respective cortical region. Thus, the impact of pathological epileptiform HFOs was minimized by including only channels that correlated their bursting in response to a cognitive task event.

## Anatomical distribution and stimulus specificity analyses

To assess the spatial distribution of the electrode contacts showing significantly correlated HFO bursting during memory processing, we mapped the brain coordinates of every contact and grouped into the five cortical lobes studied (see Fig. 4a). We then calculated Euclidean distances between any two correlated channels and summarized them as histograms to characterize the spatial extent of coincident HFO bursting (see Fig. 4c).

For each lobe, we calculated the proportion of correlated channels as the ratio of correlated channels to the total implanted channels

in that lobe. We derived the proportion of non-correlated channels by subtraction (1 − correlated proportion). We pooled these subject-level proportions across participants and averaged to obtain group-level estimates. To account for inter-subject variability in electrode contact coverage, we normalized the final proportions by expressing the number of correlated and non-correlated channels as a percentage of the total implanted channels in each lobe. This normalization ensured that their relative distributions were preserved for group-level comparisons (see Fig. 4b).

We anatomically normalized coordinates using the automated anatomical labeling atlas (AAL3)[79] to account for inter-subject cortical variability. We pooled the proportion of words in recall in which a given channel exhibited significant co-HFO bursting across subjects and mapped them spatially (see Fig. 4e). To quantify the engagement of each channel, we computed the frequency of significant co-HFO bursting occurrences across all trials in which a word was encoded or recalled. This provided a measure of how many words a given channel responded to with significant coincident bursting. We determined this frequency for each channel using a group-wise count transformation, which aggregates per-channel engagement occurrences across trials within each patient before pooling across the cohort. Specifically, for each channel, we first counted the number of trials where significant co-HFO bursting was observed. We then summed these counts separately for encoding and recall within each participant. Finally, to generate group-level engagement distributions, we pooled individual patient results while normalizing for inter-subject variability by expressing each subject's data as a proportion of their total implanted channels. This ensured comparability across participants with variability in total electrode contacts implanted.

We anatomically localized electrode contacts using co-registered post-implant CT and pre-implant MRI scans. We classified tissue type (gray or white matter) using standard cortical surface reconstructions and volumetric segmentation. Our primary analyses included all contacts, regardless of tissue type. To assess the contribution of white matter signals, we conducted control analyses restricted to gray matter contacts, and separately, using white matter contacts alone (see Supplementary Fig. S9). The gray-matter-only analysis reproduced the main temporal patterns, whereas the white-matter-only analysis revealed attenuated and more variable effects, indicating that observed co-HFO dynamics are predominantly driven by cortical activity.

To obtain cohort-wide estimates, we concatenated per-patient engagement frequencies into two aggregated datasets: one for encoding and one for recall. We then filtered these datasets into their respective lobes. We generated violin plots for each lobe to compare engagement frequency distributions between encoding and recall. Each plot displayed the density of co-HFO bursting occurrences per lobe, individual data points within each lobe group, and statistical significance markers for between-phase comparisons.

Our analysis was structured to illustrate how co-HFO bursting during memory processing is spatially distributed and functionally engaged across the cortex. We began with a single word recall event to demonstrate widespread, temporally coincident HFO activity, then mapped the anatomical distribution of significantly correlated channels (see Fig. 4a). This was followed by a cohort-level summary of the proportion of correlated contacts across cortical lobes (see Fig. 4b) and an assessment of the spatial range of co-HFO networks via inter-electrode Euclidean distances (see Fig. 4c). To capture functional differences between memory phases, we compared encoding and recall in terms of the lobar distribution of co-HFO engagement (see Fig. 4d) and then quantified how frequently each channel participated in co-HFO bursting across all words (see Fig. 4e). Finally, we aggregated these results and visualized them to highlight task-dependent and region-specific engagement patterns (see Fig. 4f, g). This sequence of analyses was designed not to introduce a new methodological framework, but to guide the reader from a single recall event to generalized patterns of anatomical and task-phase-specific co-HFO activity across participants.

**Hierarchical clustering analysis.** To identify patterns of co-occurrence across channels, we applied hierarchical clustering to the co-HFO matrices. The first step involved computing a correlation distance matrix, where the distance between two channels was defined as one minus the Pearson correlation coefficient. This metric ensured that channels with highly similar activity patterns (highly correlated signals) are considered closer, while those with dissimilar activity are treated as farther apart.

We then performed agglomerative hierarchical clustering using the average linkage method. In this approach, clusters were merged iteratively based on the average pairwise distance between all elements in the respective clusters. Average linkage (also known as UPGMA−Unweighted Pair Group Method with Arithmetic Mean) is particularly effective in preserving local structures while still capturing broader functional relationships between channels. Compared to other linkage methods, such as single linkage (which is sensitive to chaining effects) and complete linkage (which emphasizes compact clusters), average linkage provides a balance between local and global structure retention.

We visualized the hierarchical clustering results using a clustermap, where channels were reordered based on the clustering hierarchy. This reordering enhanced the interpretability of the co-HFO matrices by grouping functionally similar channels together, revealing spatially and temporally coherent network patterns. To test whether these structures depend on fine-scale timing, we applied a temporal jitter control in which HFO timestamps were randomly shifted within a ±50−100 ms window. We then recomputed clustering and synchrony metrics on the jittered data and compared them to the original to assess the role of precise temporal alignment. We defined synchrony as the mean pairwise Pearson correlation between binary HFO time series across channels for each trial, providing a global index of temporal coordination (range: 0−1). To assess clustering reproducibility, we used a bootstrap resampling approach, resampling channels independently per word. We quantified similarity between original and resampled clustering labels using the Adjusted Rand Index (ARI), which measures the agreement between two clustering solutions by comparing how often element pairs are grouped together or apart, adjusted for chance. ARI values range from −1 (complete disagreement) to 0 (random labeling) to 1 (perfect agreement).

All analyses were conducted in Python using NumPy, Pandas, SciPy, and Seaborn.

### Statistical analysis
We conducted all statistical analyses using Python to ensure rigorous and reproducible evaluation of the data. We first assessed the normality of the data distributions using the Shapiro−Wilk test ($\alpha = 0.05$). This test determined whether parametric (e.g., ANOVA, $t$-tests) or non-parametric (e.g., Kruskal−Wallis, Wilcoxon signed-rank) statistical approaches were appropriate for subsequent analyses.

To compare HFO bursting rates and coincident bursting probabilities across the four task phases (Countdown, Encoding, Distractor, Recall), we used the Kruskal−Wallis test, a non-parametric alternative to ANOVA that does not assume normally distributed data. We performed post-hoc comparisons between specific phases using Tukey's Honest Significant Difference (HSD) test, with a family-wise error rate (FWER) of 0.05, ensuring control over multiple comparisons. For the temporal dynamics of coincident bursting, we applied one-way ANOVA to detect significant differences in bursting probabilities over time. We then used Tukey's HSD post hoc test to identify specific time points that contributed to significant effects. We analyzed time-resolved comparisons of co-HFO bursting using a linear mixed-effects model

(LMM) implemented independently at each 10-ms bin. The dependent variable was the per-bin co-HFO probability, modeled as a function of task Condition (fixed effect) with a random intercept for Subject: Value ~ Condition + (1|Subject). We extracted omnibus p-values for the Condition term at each bin for significance testing. We applied this framework to all time-series probability analyses, including comparisons across task phases (see Fig. 2b–d) and contrasts between subsequently recalled vs. forgotten words and cued vs. free recall (see Fig. 3b, c). For all pairwise contrasts, we rotated the Condition reference level within the same LMM structure and computed two-sided p-values, applying Benjamini–Hochberg false discovery rate (FDR) correction across bins for each comparison. We plotted time points meeting the FDR threshold in Supplementary Fig. S4. This statistical framework accounts for subject-level heterogeneity in electrode sampling and improves inference reliability in small-sample intracranial datasets. We examined the spatial distribution of co-HFO bursting across different cortical lobes using Pearson's chi-squared test, applied separately to the encoding and recall phases. This test determined whether different cortical regions were equally involved in coordinated HFO bursting or whether specific lobes were preferentially engaged. To assess whether the same cortical regions were engaged during encoding and recall within individual subjects, we performed Wilcoxon signed-rank tests for non-normally distributed data or paired t-tests for normally distributed data. These tests determined if a shift in co-HFO involvement occurred between encoding and recall. We examined the subsequent memory effect, comparing HFO bursting rates between words that were later successfully recalled vs. forgotten using independent t-tests. This analysis evaluated whether memory-related differences in neural activity could be detected during initial encoding. We identified significantly co-activated channels using Pearson correlation analysis ($r > 0.5$). This method quantified the degree of synchronization between channels, revealing functional connectivity patterns during memory processing. We evaluated the spatial clustering of co-HFO bursting using permutation testing (10,000 iterations, FWER-corrected). This non-parametric approach tested whether observed clustering patterns exceeded those expected by chance, strengthening the reliability of spatial coordination findings. For the sequential co-HFO analyses, we compared clustering stability (Adjusted Rand Index, ARI) and synchrony differences after jittering between conditions using two-tailed paired t-tests for normally distributed data or Wilcoxon signed-rank tests otherwise. This analysis compared recall vs. encoding and recall vs. forgotten recall to determine whether the temporal organization of sequential co-HFO patterns differed across memory conditions. Finally, we quantified effect sizes for key comparisons using Cohen's d, with 95% confidence intervals (CIs) to provide standardized measures of observed differences. All statistical tests were two-tailed, and results were reported as mean ± standard error of the mean (SEM) unless stated otherwise.

### Reporting summary
Further information on research design is available in the Nature Portfolio Reporting Summary linked to this article.

## Data availability
The Human brain local field potential recordings during a battery of multilingual cognitive and eye-tracking tasks (v1) data used in this study are available in the EBRAINS database at https://doi.org/10.25493/4FZH-ZCG. All analyses were conducted on data from this openly accessible resource. The raw LFP data are protected and are not available in the public repository due to data privacy laws regarding human participants. The processed data used in this study are available at EBRAINS. Source data for all plots and figures generated in this study are provided as a Source Data file with this paper. Source data are provided with this paper.

## Code availability
The code for the analysis is available in the following GitLab repository: https://gitlab.com/brainandmindlab/coincident-hfo.

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

## Acknowledgements

This research was fully supported by the National Science Center, Poland, grant Opus LAP number: 2020/39/I/NZ4/02070 (received by M.T.K.); by the Czech Science Foundation (GAČR), project No. 21-44843 L (received by J.C.); and partially by the Gdansk University of Technology grant ARGENTUM number: 18/1/2022/IDUB/I3b/Ag (received by J.S.G.S.). The authors would like to thank Nastaran Hamedi and Luis Felipe Sarmiento of Gdansk University of Technology for insightful comments on the manuscript. This research would not be possible without the dedicated effort of the patients and their families.

## Author contributions

S.P. led the study, designed the analytical methodology and software, performed the investigation, curated the data, conducted all data processing, performed all formal analyses, and prepared the visualizations. J.C. contributed to data curation (BIDS formatting) and preprocessing. S.P. and M.T.K. wrote, reviewed, and edited the paper together. J.S.G.S., M.G., L.J., P.D., M.K., R.R., M.P., W.F., M.S.-N., P.T., A.C., and M.B. contributed to data curation and clinical resources. M.T.K. conceived the study, designed the experimental tasks, provided overall supervision, secured funding, and guided project administration. All authors approved the final version of the paper.

## Competing interests

The authors declare no competing interests.
