## [Transparent Peer Review file · Nature Communications]

Global coincident bursts of high frequency oscillations across the human cortex coordinate large-scale memory processing.

Corresponding Author: Dr Michal Kucewicz

Version 0:

Reviewer comments:

Reviewer #1

(Remarks to the Author)

The manuscript investigates the role of coincident high-frequency oscillations (co-HFOs) in memory processing using intracranial recordings from epilepsy patients performing verbal memory tasks. The authors report that co-HFOs are modulated during encoding and recall, engage widespread cortical regions, and exhibit sequential organization, suggesting a mechanism for large-scale neural integration. While the study addresses an important topic in cognitive neuroscience, several issues—ranging from methodological clarity to interpretational scope—should be addressed to strengthen the work. Below, I outline major and minor concerns, alongside possible suggestions for improvement.

MAJOR COMMENTS

1. General feedback

- I found the manuscript somewhat challenging to read. For instance, the terms "electrode channel" and "contacts" are seemingly used interchangeably. I suggest consistently using the same terminology throughout the text. Another example pertains to the figures: the color coding is not always clear or effective in conveying the intended message. In Figure 2, the trace colors in panel (b) refer to the legend in panel (c), but the colors are quite similar to those in panel (a). In Figure 4, the grayscale does not adequately represent the results you aim to communicate. These are just a few examples, but overall, I believe the manuscript, which is indeed an interesting work supported by valuable data, would benefit significantly from substantial reformatting to enhance readability for a broad audience.

- Thank you for sharing your data. I am deeply convinced that science can progress only through data sharing and ensuring replicability. Moreover, this practice further highlights the invaluable nature of such data and the time dedicated by these patients. I sincerely appreciate your efforts. At the EBRAINS link, the data are still listed as "under embargo" (is common with human data), and I understand that lifting this restriction takes time. In addition to EBRAINS, I recommend using other data-sharing platforms in parallel (e.g., OpenNeuro for non-sensitive derivatives like processed HFO timestamps)

2. Methodological comments

- Most neuroscientists working with intracerebral data examine physiological activity even though the data are collected from epileptic patients. This approach is feasible since, on average, 80% of the contacts are implanted in physiologically normal areas (see Parvizi, Nature Reviews Neuroscience, 2018). However, we must explicitly address this in the Methods and a dedicated Limitations section, where we must mention at least: (i) The seizure onset zone for each patient (this may already be in the dataset but should be reiterated here). (ii) Whether patients had malformations and, more broadly, their specific etiologies (e.g., high-frequency oscillations [HFOs] in cortical dysplasia may differ significantly from those in heterotopia). (iii) The number of excluded contacts per patient and their locations. (iv) The type of activity removed during preprocessing, with examples provided in the supplementary material. (v) Engel scores at one year post-surgery. Additionally, it should be clearly stated in the Limitations—especially since this study focuses on HFOs—that the results may be partially influenced

by epilepsy-related factors. I would highly recommend the authors to face these issues to guarantee reproducibility and rigour.

- The authors re-referenced the data using a bipolar montage. While bipolar referencing is optimal for localizing low-frequency activity (LFP), it is well-established that for gamma-frequency analysis, a monopolar reference (using nearby white matter or an acquisition-neutral white matter reference) is preferable (see Li, NeuroImage, 2018; Arnulfo, Nature Communications, 2020). I understand that re-running the entire analysis in a monopolar reference scheme would require significant effort, but I recommend either: repeating the full analysis with a monopolar reference (better choice), or at least testing whether key findings hold (or are better represented) when applying monopolar referencing in a subset of subjects.

- It is unclear whether the authors excluded white matter contacts from their analysis. Figure 4 suggests they did not. I recommend conducting separate analyses for white matter contacts to assess their potential impact on the results.

- I know that the data are shared, but a supplementary figure showing SEEG implantation of each subject could help (see also my comments on statistics)

3. Statistical Analysis

- Given the small sample size ($N=12$), the authors might consider using mixed-effects models statistics to better account for inter-subject variability. This is especially relevant in intracranial studies, where individual differences in anatomical implantations can heavily influence group-level results.

- Along the same line, since the findings are largely presented at the group level, it would be valuable to demonstrate that the effects are consistent across participants, particularly given the limited sample size. More broadly, wherever feasible, group-level results should be supplemented with subject-specific visualizations.

4. Interpretation and Claims

- I would set aside interpretations in the context of the Global Neuronal Workspace Theory (GNWT) and any potential implications for consciousness in general. Data reported here are rich and compelling without such considerations, and, most importantly, they were not recorded with the intention of studying consciousness.

- The finding that co-HFOs are not word-specific (Fig. 4) conflicts with engram literature (e.g., Vaz et al., 2020). Discuss whether this reflects macro-contact limitations or truly non-specific networks.

MINOR COMMENTS

- The audio version of the task is mentioned briefly. Clarify whether co-HFO patterns differed between modalities (could support sensory-independent effects).

- Temporal Resolution: The 10 ms bin size may obscure finer-scale HFO dynamics. Discuss whether higher temporal resolution (e.g., wavelet-based analysis) was explored.

- Some figures are cited out of order (e.g., Fig. 5 appears before Fig. 4 in the text). Ensure chronological flow.

- Define all abbreviations at first use (e.g., PAL in the Introduction).

- For hierarchical clustering (Fig. 5), add silhouette scores or bootstrapping to assess robustness.

CONCLUSIONS

The study provides valuable insights into the spatiotemporal organization of co-HFOs during memory tasks. Addressing the methodological and interpretational concerns above would, in my view, significantly elevate the manuscript's impact. With revisions, this work could become an important step forward in understanding large-scale neural coordination in memory. Finally, I commend again the authors for their comprehensive dataset and commitment to open science. The work is ambitious and timely, and I look forward to seeing the revised manuscript.

All the Best

Andrea Pigorini

(Remarks on code availability)

Reviewer #2

(Remarks to the Author)

This study by Prathapagiri et al. builds on rodent and human work showing that high-frequency oscillations (HFOs) co-occur across cortical regions up to ~20 mm apart (“co-HFOs”). Micro and macro-contact stereo-EEG in 17 epilepsy patients (electrode positions from co-registered CT/MRI), were used. Participants performed a Free Recall task (180 words, 15 lists) and a Paired-Associate Learning task (6 word-pairs, 15 lists). The authors applied a refined detection pipeline to extract HFO events, binarized these events into 10 ms time-bins, and computed Pearson correlations over these bins across channels. A threshold of $r > 0.5$ defined co-HFOs. They then averaged co-HFO rates across four task phases (preparation, encoding, distractor, recall), frequency bands, and cortical regions.

Main findings:

. Co-HFOs occur during all task phases but are most strongly modulated during memory encoding and recall across visual and higher-order cortical areas.

. For subsequently recalled words, co-HFO bursting is suppressed immediately before stimulus onset and enhanced during word presentation.

. Hierarchical clustering reveals that during recall co-HFOs form temporal sequences spanning multiple, spatially distributed networks.

Overall, the results are highly relevant, offering a new perspective on how HFOs and co-HFOs contribute to memory processes. However, I have some concerns regarding aspects of the data analysis, interpretation, and participant reporting, which I have detailed below.

Strengths:

1) The identification of “synfire chain”-like sequences of co-HFO activations across distributed cortical sites is particularly compelling. The authors relate this finding to broader frameworks such as “binding theory,” which is a useful conceptual anchor. However, since their paradigm is limited to verbal memory, further work is needed to explore whether these dynamics generalize to multisensory integration processes more typically associated with the binding problem. Nonetheless, these results pave the way for targeted experiments probing whether co-HFO dynamics might serve as a general neural substrate for perceptual binding. With additional analytical validation (e.g., using methods suggested below), this discovery will have far-reaching implications for Neuroscience.

2) A clear strength of this study is its cohort size: with 17 patients, it sits in the upper range of human intracranial HFO/memory investigations, providing greater statistical power and confidence in the generalizability of its findings.

3) By combining Free Recall and Paired-Associate tasks - and even audio versus visual versions - the study convincingly dissociates encoding, maintenance, distraction, and retrieval, while controlling for sensory and motor confounds.

4) By including native Czech, Slovak, and Polish speakers, the study broadens its applicability beyond the usual English-dominant cohorts, enhancing cross-linguistic generalizability.

Major comments:

1) I am skeptical of the arbitrary choice of an $r > 0.5$ threshold to define co-HFOs. This dichotomization introduces the risk of misclassifying marginal cases (e.g., $r = 0.499$ vs. 0.501) and may obscure meaningful variation in the data. Given that HFOs are already binarized, the stated rationale of ‘mitigating noise’ is unconvincing. For the core analysis presented in Figure 2, I recommend treating r as a continuous predictor. Alternatively, a sensitivity analysis across a range of thresholds (e.g., 0.3 to 0.7) would help demonstrate that the findings are not an artifact of a single, arbitrary cutoff.

2) The authors argue that their findings on co-HFO contributions to cognitive processes are generalizable beyond the specific patient cohort studied. I am nearly convinced, given the overlap with prior reports on co-HFOs. However, because their approach closely mirrors that of earlier studies involving similarly composed patient populations, there is a risk of establishing a self-reinforcing framework in which disease-related influences are systematically overlooked. To enable readers to critically assess the extent of generalizability, more detailed information about the clinical characteristics of the included patients is essential and would strengthen the scientific robustness of the work.

I recommend addressing this in a dedicated paragraph within the Methods section. For clarity, the current “Participants and Electrode Localization” section could be split - electrode localization details might be more appropriately placed in a new “Electrophysiological Recordings” subsection. The clinical description should include relevant information such as epilepsy etiology (e.g., hippocampal sclerosis, tumor, cortical malformation), disease severity (seizure frequency), duration and class of pharmacological treatment, and the laterality and localization of the epileptic focus (e.g., left vs. right hemisphere; temporal vs. extratemporal).

A related aspect that requires clarification is handedness. Was hemispheric dominance considered in the analysis?

Specifically, did HFO occurrence or co-HFO frequency differ between the dominant and non-dominant hemispheres? Addressing these questions could reveal potentially relevant lateralization effects. This aspect appears to be hinted at in Figure 4a, which shows bilateral HFO occurrence during recall of a single word, but it is not systematically explored in the current analysis.

3) How did task performance vary across participants, and was there a relationship between each patient's co-HFO abundance and their memory performance?

4) On the sequential occurrence of co-HFOs: In the second-to-last section of the Results, the authors introduce their analysis by asking whether global co-HFOs occur "during encoding or recalling a word." However, the results presented focus exclusively on sequential co-HFOs during recall. What were the corresponding findings for encoding? Does encoding similarly involve sequential, synfire chain-like activation patterns?

In addition, was the analysis of sequential co-HFOs extended to data from the PAL (paired-associate learning) task? And relatedly, do co-HFOs associated with failed recall ("forgotten" cases, as shown in Fig. 3b) also exhibit sequential activation?

Contrasting these different conditions – encoding, successful recall, and failed recall – could be highly informative and would help determine whether the observed sequential co-HFO patterns are specific to successful memory retrieval.

5) Given the potential impact of demonstrating sequential co-HFOs, the authors could reinforce their findings — beyond hierarchical clustering — by applying one or more of the following approaches:

A useful control could be the application of a surrogate jitter test: Randomly perturb each HFO timestamp by e.g. ± 50 -100 ms within trials, re-run the clustering, and assess whether the sequential modules persist. If they disappear or weaken, this would support the claim of true temporal order.

Another option would be cross-correlogram analysis: for each channel pair in a cluster, compute their HFO-train cross-correlation. Observing consistent peaks at fixed positive lags - aligned with the clustering sequence - would further corroborate a sequential propagation of HFOs.

Further comments:

- For this revision - and for future manuscripts authored by these colleagues - I strongly recommend enabling line and/or page numbering. This small adjustment requires minimal effort from the authors but significantly improves the efficiency of the review process, as it allows reviewers (and authors in their replies) to reference specific parts of the text with ease.

- At times, the narrative feels somewhat diffuse. I suggest tightening the focus throughout the manuscript — particularly in the Results and Discussion sections — to help ensure each part conveys a clear and targeted message.

- The term 'burst' needs clarification. Do you use it to mean a cluster of closely spaced HFO events, or something else? In electrophysiology, a 'burst' usually refers to a brief train of spikes (or bursts of neuronal firing), whereas here you're detecting oscillatory events. Please define what constitutes a 'burst' in your HFO analyses.

- Results, 3rd paragraph: "Pearson's co. relation" -> correlation.

- Methods: In the Detection of individual HFO bursts section, the authors state that "...greater artifact susceptibility necessitates more stringent detection criteria than those typically applied in rodent studies." However, artifacts also pose significant challenges in behaving-rodent recordings, where investigators routinely combine multiple criteria to exclude false-positive ripples. Defining robust HFO detection methods therefore remains an active area of debate (e.g., Navas-Olive et al., eLife 2022; Liu et al., Nat. Commun. 2022).

- Methods: Analysis of coincident HFO bursting, 1st paragraph: There appears to be a discrepancy between the frequency bands listed in the text ('60–150 Hz, 150–250 Hz, and 250–500 Hz') and shown in Figure 2 with those shown as illustration of band separation in Fig. S1 ('gamma: 60–80 Hz; ripple: 80–250 Hz; fast ripple: 250–600 Hz'). Please correct so that consistent bands are reported throughout the manuscript.

- Please check reference 57 – the authors are missing. Also, is this the appropriate citation for the fifth paragraph of the Discussion, sixth line from the bottom? The context refers to patients, whereas the study by Valero and colleagues was conducted in rats, so the reference seems potentially misplaced.

Figures:

- Figure labelings, general: Please increase the font size of all axis labels, as they are currently difficult to read.

- The color scheme in Figure 2 is confusing: in panel a and e, colors differentiate brain regions, whereas in panels b–d they mark experimental time phases. To resolve this, separate panel a and e from panels b–d. Please mention explicitly what is indicated as grey functions in panel 2b.

- The asterisks in Figure 2 are difficult to discern; please consider replacing them with more visually distinctive markers to clearly indicate statistical significance. Additionally, the authors could color-code the significance levels of these markers. This would not only make the figure more informative overall but also facilitate comparison of significance across the different experimental phases shown.

- Figure 4D: Please label the two grey shading areas in the panel. While the intended meaning (encoding vs. retrieval) is implicitly clear, explicitly indicating the corresponding phases would improve the clarity of the panel.

- In the legend to Fig. 5, Fig. 3c is mentioned as a reference, but this appears to be a mistake – likely, Fig. 2c was intended.

(Remarks on code availability)

Reviewer #3

(Remarks to the Author)

This study by Prathapagiri and colleagues examines human intracranial EEG recordings to investigate and characterize coordinated high frequency oscillations across multiple cortical sites during memory encoding and recall. The behavioral task is a free recall verbal memory task in which there is a preparatory period, and encoding list of words, a math distractor period, and a retrieval period. They examine co-occurring high frequency oscillatory events across all sites during all of these stages. As the authors note, recent evidence has suggested that ripples, a specific high frequency oscillatory event initially described in the rodent hippocampus, play a role in memory formation. Here the authors are careful to avoid the debate regarding how to name these high frequency events, and whether ripples should be considered a separate entity, and correctly just focus on the occurrence of these events and their distribution across brain regions. They find evidence of coordinated HFO events across brain regions during memory processing. They also find that this coordinated activity appears temporally organized, although the patterns of activity do not appear specific to the individual memoranda. Overall this is an interesting and timely study, as recent research has highlighted the role of HFO events, and possibly ripples, in human memory. The conclusions could potentially be valuable for the research community, but there are some suggestions that could improve the overall conclusions.

For the data presented in figure 1, it was a little difficult to follow how the data are being presented. For example, there are some rasters that show the HFO events across channels time locked to the events in a, but the mean distributions do not appear to show any meaningful trends (except perhaps during vocalization). Are there any statistics to support the conclusion that there are increased HFO detections during the memory stages?

The data in figure 1c and 2a in which they present co-incident detections were also confusing. It is not clear what is actually being plotted here. It looks like the y-axis are channels, and they sort channels passed on their pairwise correlation, but it is not clear what then the color axis represents and how this reflects pairwise correlations. Are these pairwise correlations with all other channels (which seems to be the case based on the methods)? However, they also look at the Pearson correlation between channels? Are these pairwise interactions consistent between channels? It is difficult to interpret these results. The presentation of the anatomic distribution of the data is also challenging.

This then extends to the time series comparing the rates of co-occurrence during different memory stages for the correlated channels. Presumably different channel pairs will have different correlations and involvement in different memory stages and even individual items. How are all these data collapsed here? Is every channel used as the 'source' from which the time series of the correlated pair is extracted? It also seems like this analysis may be a little circular. The authors are selecting channel pairs that exhibit high correlations in HFO activity, and then investigating the rates of co-occurrence of HFAs and comparing them to the electrode pairs that do not have correlated activity.

Its not clear that comparing the four different task phases as is done in Fig 2c is appropriate, since it is not clear that the time=0 point in each phase should be comparable.

The statistical analyses could also be improved. For example, the authors note that there is lower co-HFO activity during successful recall, but this is based on an incredibly small time period during which the two traces diverge. Is this corrected for multiple comparisons? The authors should use a clustering procedure for these time series analyses. More broadly for this point, it's not clear what these data represent. These represent the grand average from all lobes, but do any of the individual brain regions demonstrate significance. In addition, these effects appear to arise during encoding, yet the broad encoding analyses during free recall show almost no variability in co-HFO rates.

The authors' claim that the peak of co-HFO rates during retrieval occurs during word presentation in paired associates but at the time of vocalization during free recall is interesting. However, it does not appear that these two phenomena are directly compared in the same subjects in the same electrodes, which would strengthen this claim.

In general, the entire manuscript could be improved by more clearly presenting the data in an organized manner. For example, the authors should start presenting the results by first summarizing the number of subjects, demographics, behavioral performance, etc. Or, for example, figure 4 is difficult to parse in order to extract out significant conclusions. Some of the figures are not labeled (fig 4d). Others appear to simply show all electrodes without much in the way of an interpretable summary (4e). Or, for example, the participation of electrodes in either encoding or retrieval words should be directly determined by the involvement of co-HFO activity in each stage (4f). Or, for example, the presentation of hierarchical clustering and the sequence of HFO activity is also unclear, as well as whether this is a general phenomenon across all recall events or only particular to this one word example.

(Remarks on code availability)

Reviewer #4

(Remarks to the Author)

This paper by Prathapagiri et al. investigates the dynamics of high-frequency oscillations (HFOs) and their potential role in large-scale cortical coordination during human memory processing. The authors report that HFOs are not only locally generated but also exhibit global co-occurrence across widespread sensory and associative cortical regions. These coincident HFOs are shown to be modulated by different memory phases, with specific patterns of suppression and enhancement during encoding and recall. The dataset is substantial, and the analyses are carefully executed. While the study is based on a large intracranial human dataset, the manuscript does not offer a sufficiently novel mechanistic insight or theoretical advance beyond the existing literature. In addition, several methodological concerns require clarification and revision to ensure the robustness and interpretability of the findings.

Major comment:

1- Widespread cortical high-frequency oscillations (HFOs) during memory tasks have been previously documented, including by some from the authors themselves (e.g., Kucewicz et al., 2014; Mishra et al., 2025). The reported modulation of HFOs by memory phase (encoding, recall) has also been robustly reported (e.g., Burke et al., 2014; Staresina et al., 2015; Norman et al., 2019; Sakon and Kahana, 2022). Likewise, temporally structured cortical activity during retrieval, including replay-like sequences, has been described in human iEEG (e.g., Michelmann et al., 2016; Vaz et al., 2020; Xie et al., 2024). While the current work reinforces these findings, it does not provide sufficient new conceptual advances to meet the novelty threshold expected for Nature Communications.

2- The signal-to-noise ratio of selected HFO bursts is too low (2SD were used as a threshold). Consequently, many of these events appear to be small, narrow-band deviations that may not meet rigorous criteria for genuine HFOs or ripples (e.g., Fig S1). This raises the concern that the reported widespread co-activation patterns may, in part, result from spurious correlations or broad-band spectral fluctuations rather than true high-frequency events (see Leszczyński et al., 2020). In particular, some detected events may reflect gamma-band activity rather than discrete ripple-like transients. To address this, the authors should provide a detailed amplitude and frequency distribution of the detected events across regions and more rigorously assess their cross-regional relationships (surrogate or control analyses). This would help clarify whether the observed coordination reflects genuine oscillatory activity or broad-band non-specific co-activation.

3- Cortical HFOs are often associated with epileptogenic tissue, and the patients in this study were undergoing pre-surgical evaluation. Thus, some overlap between seizure onset zones and HFO-generating regions is expected. However, the manuscript provides no clinical information regarding seizure foci, medication status, or the extent of pathology, making it difficult to assess how much the reported widespread HFO propagation may be influenced by underlying epileptic networks. Without this context, it remains unclear whether the observed co-activation patterns reflect physiologically meaningful coordination or pathological spread. Moreover, no analysis is presented to examine directionality of propagation or whether there is evidence for structured transfer of activity beyond broadband associations. Such information would be crucial to dissociate large-scale functional coordination from confounds related to epileptiform dynamics.

4- The analysis presented in Fig. 5 raises several methodological concerns that challenge the claim of genuine large-scale coordination of HFOs. First, use of Pearson correlation ($r > 0.5$) without proper controls or surrogate testing may inflate apparent synchrony, particularly given the low SNR of detected events (see point 2). Second, the statistical rigor of the correlation-based network definition is unclear: the threshold appears arbitrary, and no correction for multiple comparisons is reported.

5- Finally, there are some issues with limited specificity of the reported changes and anatomical interpretability. Differences across cortical lobes may reflect uneven electrode coverage rather than true anatomical patterns. No null model is provided to account for implantation bias or the expected co-occurrence rate by chance. Moreover, the widespread and redundant involvement of electrodes across multiple recalled words (Fig. 5e) suggests low selectivity, weakening the argument for structured or memory-specific processing. Finally, the anatomical distinctions between encoding and recall phases are primarily descriptive; without measures of directed connectivity or temporal dynamics, these observations offer limited mechanistic insight.

(Remarks on code availability)

The code is not currently available for review. To ensure reproducibility and transparency, I strongly encourage the authors to share the full analysis code along with a representative subset of the dataset.

Version 1:

Reviewer comments:

Reviewer #1

(Remarks to the Author)

I would like to start by sincerely congratulating the authors for the incredible amount of work carried out in this revision. Preparing more than 60 pages of detailed responses to the reviewers is an impressive achievement. I also thank the authors

for carefully addressing not only my previous suggestions, but also those from reviewers 2 and 3. The manuscript has gained substantially in terms of readability, methodological rigor, and reproducibility (the latter already ensured by the public availability of the data).

I have only a minor comment regarding the Discussion. I fully understand the interpretative value of placing this study in the broader context of the neuroscience of consciousness, and as it stands the authors' treatment in the final part of the manuscript is appropriate at a speculative level. However, in the passage where the authors write:

"From the perspective of these studies, coordination of HFOs across the brain in coincident bursts could be linked to global integration of information and thus provide a plausible neural mechanism for concepts like 'ignition' or 'global broadcasting' proposed in the global neuronal workspace hypothesis 74"

I believe it might be useful to also acknowledge other theoretical frameworks that could potentially align with the reported findings. In particular:

- 1) Predictive processing & neurorepresentationalism
- 2) Recurrent processing theory
- 3) Information Integration Theory

For the authors' convenience, I would like to point out a recent review (Storm, Neuron 2024), where these different theories are compared from an empirical perspective, which might serve as a helpful reference.

Once again, congratulations to the authors for their outstanding work and good luck with the final steps of the publication process.

Andrea Pigorini

(Remarks on code availability)

Reviewer #2

(Remarks to the Author)

The authors have responded thoroughly and satisfactorily to the major comments I provided.

In my earlier comments, I raised concerns about the manuscript's overall readability, a point also made by other reviewers. In their revision, the authors added subheadings – they improve structure.

However, several passages remain difficult to follow, so I recommend another round focused on clarity and readability. The points below are intended to guide targeted revisions:

(I) Sentence length.

Several sentences run 40–70 words. Please split very long sentences and aim for an average of ~15–25 words (with occasional variation for emphasis).

(II) Redundancy / wordiness.

Reduce pleonasm and stacked phrases. Examples (illustrative, not exhaustive):

l.69–70: "networks of connected areas" → "networks."

l.68–69: "present a plausible activity for large-scale tracking of ... discharges" → recast; the construction is wordy and non-idiomatic.

l.71–72: "wide range ... spanning ... frequency ranges" → "range ..." (drop "spanning ... ranges").

(III) Results vs. Methods.

Please keep methodological detail out of the Results (move to Methods and retain brief reminders only where essential). For example:

l.132: inclusion/selection criterion ("same inclusion criterion to ensure enough trials...").

l.143–145: participant instructions ("patients were encouraged to visualize...").

l.147–149: event time-stamping rule (midpoint between onset and offset; Suppl. Fig. S1).

l.149–150: cross-channel analysis setup ("temporal correlations ... from the same and different regions") → define the pipeline in Methods.

l.187–190: LMM structure (Condition fixed; Subject random intercept), 10-ms bins, significance bars ($p < 0.05$).

I.191–196: post-hoc scheme (rotating reference level; six pairwise contrasts) and FDR correction across time bins.
I.233–236: repeated mention of the same LMM specifications.

(IV) Voice (active vs passive)

For readability, please favor active voice in the Results (e.g., “We observed/estimated/found ...”). In the Methods, active or passive is acceptable—choose one and apply it consistently within the section, avoiding agentless passives where the actor matters.

___ Further comments: ___

- Table S1. The column label “Rate of co-HFO bursts (Hz)” is unclear. Values up to ~2960 would imply ~2,960 events/s, which seems implausible for co-HFO detections. Please clarify or correct, if necessary.
- In I. 674ff and I. 699 text is struck through. Please check and correct.

(Remarks on code availability)

Reviewer #3

(Remarks to the Author)

The authors have provided a comprehensive revision of their manuscript in which they addressed and clarified many of the issues raised in the initial review. Overall, this is an interesting manuscript that will be of interest to researchers focused on human memory.

(Remarks on code availability)

Reviewer #4

(Remarks to the Author)

First, I would like to thank the authors for a thorough revision, which has significantly improved the manuscript. In particular, the clarification of the clinical data and the justification of analytical parameters have strengthened the paper. However, some of my major concerns remain and were not fully addressed with additional data or analyses. In particular:

- Event specificity and signal-to-noise ratio: While the authors clarified that candidate HFO events were subjected to additional amplitude and duration thresholds, I remain concerned that many detected events may still include low-SNR or borderline oscillatory activity. Without a detailed distribution of event amplitudes and frequencies across regions, it is difficult to confirm that the reported co-activation patterns reflect robust HFOs rather than broad-band fluctuations or residual gamma activity. In addition, while the surrogate and threshold tests strengthen the evidence that co-HFO patterns are not purely noise, they do not fully address my primary concern.
- Directionality and propagation of co-HFOs: The manuscript does not provide evidence for structured temporal propagation or causal relationships among co-HFO events across cortical regions. The authors’ explanation that directionality analysis is complex does not fully address my concern that the widespread near-simultaneous co-activations could reflect non-specific or epileptiform-driven correlations rather than physiologically meaningful HFO coordination.
- Interpretation of large-scale co-activation: Although surrogate jitter analyses and threshold robustness checks mitigate concerns about inflated correlations, the remaining uncertainty about the specificity of detected events limits confidence in the functional interpretation of widespread cortical co-HFOs. The manuscript does not convincingly dissociate genuine oscillatory coordination from possible confounds related to epileptiform activity.
- Novelty and scope: While the study provides interesting observations on large-scale co-HFO dynamics, the current manuscript does not provide sufficient evidence of specific, novel mechanistic insight that would justify publication in a high-impact journal such as Nature Communications. The remaining concerns regarding event specificity and network interpretation limit the conceptual advance

(Remarks on code availability)

I didn't test the code

Version 2:

Reviewer comments:

Reviewer #1

(Remarks to the Author)

the authors have addressed all my comments.

All the Best

AP

(Remarks on code availability)

Reviewer #2

(Remarks to the Author)

The authors have thoroughly revised their manuscript, addressing and clarifying the points I raised in the last round of review. I am pleased to recommend the manuscript for acceptance in its current form. Congratulations to the authors on this interesting and fundamental study!

(Remarks on code availability)

Reviewer #4

(Remarks to the Author)

I appreciate the authors' efforts in revising the manuscript. Most of my previous comments have been addressed through textual clarifications. However, as noted in my earlier report, I continue to have some concerns regarding the specificity of the findings, which I view as part of a broader conceptual discussion within the field.

Given that this is a matter of ongoing debate in the community, and considering the overall support from the other reviewers, I do not wish to delay publication further. I encourage the authors to remain mindful of these broader issues as the work is disseminated and discussed within the field.

(Remarks on code availability)

Dear Reviewers,

We would like to express our sincere gratitude for your careful and diligent review of our manuscript entitled 'Global coincident bursts of high frequency oscillations across the human cortex coordinate large-scale memory processing' (NCOMMS-25-28589). We appreciate the thoughtful and constructive comments provided by the reviewers, which have been invaluable in helping us significantly improve the quality and clarity of our work. Especially the final part of the results, showing temporal organization of coincident HFO bursting, has now been considerably strengthened with a jittering procedure suggested in your comments. We have also made the statistical analyses more rigorous, ensuring greater robustness and reliability of our findings.

Below, we provide a detailed point-by-point response to each comment. The comments are put in **bold** followed by our response and any edited text from the revised manuscript quoted in *italics* for your convenience. Any changes to the text are also **highlighted in yellow**, as they are in the new revised manuscript version. Revised versions of the figures/tables or new figures/tables prepared in response to the comments are pasted here in addition to the main manuscript and the supplementary materials. All revisions are indicated by references to particular pages and lines provided in the revised manuscript.

Reviewer #1

The manuscript investigates the role of coincident high-frequency oscillations (co-HFOs) in memory processing using intracranial recordings from epilepsy patients performing verbal memory tasks. The authors report that co-HFOs are modulated during encoding and recall, engage widespread cortical regions, and exhibit sequential organization, suggesting a mechanism for large-scale neural integration. While the study addresses an important topic in cognitive neuroscience, several issues—ranging from methodological clarity to interpretational scope—should be addressed to strengthen the work. Below, I outline major and minor concerns, alongside possible suggestions for improvement.

Thank you very much for recognizing the importance of this work in understanding the mechanisms of widespread cortical integration of memory processing. We particularly appreciate the suggestions to strengthen the statistical analysis and improve readability.

1. General feedback

I found the manuscript somewhat challenging to read. For instance, the terms "electrode channel" and "contacts" are seemingly used interchangeably. I suggest consistently using the same terminology throughout the text. Another example pertains to the figures: the color coding is not always clear or effective in conveying the intended message. In Figure 2, the trace colors in panel (b) refer to the legend in panel (c), but the colors are quite similar to those in panel (a). In Figure 4, the grayscale does not adequately represent the results you aim to communicate. These are just a few examples, but overall, I believe the manuscript, which is indeed an interesting work supported by valuable data, would benefit significantly from substantial reformatting to enhance readability for a broad audience.

The terminology used for the electrode contacts and channels has indeed been inconsistent, even right in the abstract. In the revised version of the manuscript, we define each electrode implanted in a patient to have multiple contacts (macro or micro) - each one of them recording a signal to one channel of the acquisition system. For example, one hybrid electrode lead typically had 8 macro- and 10 micro- contacts,

recording signals on 18 channels in total. HFO bursts were recorded from particular electrode contacts and then analyzed between particular channels, e.g., for pairwise correlation or hierarchical clustering.

We have now found every instance of using the terms ‘electrode’, ‘contact’, and ‘channel’ and revised these according this rule in the following lines of the revised manuscript:

Line 40,149,170, 306,309,311,559,567,572,574,640,642,643,790,797,801,802,804,920,922

To further improve clarity and accessibility of the figures, we have also added individual legends directly to the relevant plots, like in the example of the revised figure 2 pasted below. This ensures that the color coding and labeling are immediately interpretable without requiring the reader to cross-reference multiple panels.

Figure 2. Coincident bursting across sensory and associational cortical areas is modulated by memory processing. a) Matrices of coincident HFO detections from an example patient, plotted as in Fig. 1c, show all channels localized in the five cortical lobes studied in a descending order from the most correlated pairs on top (sorted by maximum pairwise correlation). Channel color indicates anatomical lobe. b) Probabilities of co-HFO bursting across the correlated channels from all patients reveal times of significantly increased or decreased co-activity relative to the non-correlated channels (shown in grey; asterisks denote a significant difference identified by the linear mixed-effects model(LMM)). c) Comparison between the four task phases indicates significantly decreased probability of coincident HFO bursting approx. one second before recall verbalization (blue) with subsequent gradual rise and significantly increased probability a couple of hundred milliseconds before and after the verbalization onset (asterisks denote a LMM-derived significant difference between the four task phases). d) A consistent temporal pattern of HFO co-bursting is present in the high gamma, ripple and fast ripple frequency ranges. e) Summary of cortical lobe engagement in HFO co-bursting shows equal contribution in all task phases. * indicates $p < 0.05$.

Thank you for sharing your data. I am deeply convinced that science can progress only through data sharing and ensuring replicability. Moreover, this practice further highlights the invaluable nature of such data and the time dedicated by these patients. I sincerely appreciate your efforts. At the EBRAINS link, the data are still listed as "under embargo" (is common with human data), and I understand that lifting this restriction takes time. In addition to EBRAINS, I recommend using other data-sharing platforms in parallel (e.g., OpenNeuro for non-sensitive derivatives like processed HFO timestamps)

Thank you for the kind words and suggesting other platforms for sharing the datasets. Working with EBRAINS actually turned out to be slow, resulting in the embargo delay. The dataset embargo has now

been lifted and we invite the reviewer to visit the link again. We actually have a long-term plan goal of storing all detections of HFO bursts and lower frequency discharges from our datasets into one large database for efficient queries and analysis. There already is a prototype database of pathological HFO detections in one of the partner sites in Brno, but implementing this across multiple sites in Poland and in the USA remains a challenge of our ongoing efforts. OpenNeuro provides a feasible alternative.

2. Methodological comments

Most neuroscientists working with intracerebral data examine physiological activity even though the data are collected from epileptic patients. This approach is feasible since, on average, 80% of the contacts are implanted in physiologically normal areas (see Parvizi, Nature Reviews Neuroscience, 2018). However, we must explicitly address this in the Methods and a dedicated Limitations section, where we must mention at least: (i) The seizure onset zone for each patient (this may already be in the dataset but should be reiterated here). (ii) Whether patients had malformations and, more broadly, their specific etiologies (e.g., high-frequency oscillations [HFOs] in cortical dysplasia may differ significantly from those in heterotopia). (iii) The number of excluded contacts per patient and their locations. (iv) The type of activity removed during preprocessing, with examples provided in the supplementary material. (v) Engel scores at one year post-surgery. Additionally, it should be clearly stated in the Limitations—especially since this study focuses on HFOs—that the results may be partially influenced by epilepsy-related factors. I would highly recommend the authors to face these issues to guarantee reproducibility and rigour.

We have now prepared a table with the requested clinical information and added a new paragraph in the first subsection of Methods to explicitly outline the limitations of this study associated with including potential pathological activities (inside and outside of the seizure onset zone) in the analysis of physiological activities. Below, we include the revised paragraph as well as the corresponding clinical data table (Table 2) for the reviewer's convenience.

Lines: 580 - 597

All participants recruited in this study were patients suffering drug-resistant epilepsy with various clinical backgrounds of the disease and anatomical localizations of the seizure onset zones (see Table 2). Hence, the intracranial recordings in these patients are inherently limited by the pathological discharges, including the interictal epileptiform spikes and HFOs, especially in the fast ripple frequency ranges. We cannot fully exclude a potential influence of these epilepsy-related activities on our methodology and the reported results, even though our design was based on detecting HFOs consistently around cognitive events associated with memory processing, which would be expected to average-out any potential pathological activities. We previously reported an interaction between memory processing and incidence of epileptiform discharges in these patients, showing a lower probability of their occurrence during memory processing (Matsumoto et al. 2013). Also, most of the coincident discharges reported here comprised HFOs detected at frequencies lower than 250 Hz, which is thought to be a lower threshold for pathological oscillations (Valero et al. 2017). Overall, pathological discharges are not expected to be consistently induced by memory processing and analyses across large numbers of trials should effectively diminish any potential influence of their occurrence, even though it can never be completely ruled out when studying any physiological function of the brain.

A new Table 2 has also been prepared specifying the contacts localized in the seizure onset zone, any structural MRI findings relevant for the pathophysiology of epilepsy, and the outcome of epilepsy surgery

expressed as the Engel score. The table has been added to the main manuscript and pasted below for reviewer's convenience.

Dataset	ID	SOZ channels	SOZ structures	MRI Findings	Engel Score
BR	2	-	-	Normal	-
BR	3	B1-4, C1-3, B'1-5, C'1-3	Hippocampus, Parahippocampal gyrus	Normal	-
BR	5	G'1-2, R'1-6, S'1-3	Cingulate gyrus, Precuneus, Superior occipital gyrus	Normal	IVB
BR	6	G'1-4, H'7-11	Middle temporal gyrus	Normal	IV
BR	7	Y'2-10, I'1-4	Insula	Venous angioma, cavernoma parieto-occipital left	IV
BR	8	I'1-6	Paracentral lobule, Postcentral gyrus	Polymicrogyria postcentral sulcus left	IIIA
BR	9	-	-	Normal	-
BR	10	B1-3, C1-3, B'1-3, C'1-5	Hippocampus, Parahippocampal gyrus	Normal	-
BR	11	Y'1-4, I'1-5	Insula	Normal	-

BR	13	B'1-4, C'1-4	Hippocampus	Posttraumatic changes temporo-occipital left	-
BR	14	Y1-4, X1-3	Insula	Normal	-
BR	16	B1-2	Parahippocampal gyrus	Postsurgical changes right anterior mesial temporal resection (AMTR)	-
WR	1	AMG1-2, TP1-2	Parahippocampal gyrus	Cortico-subcortical blurring temporal pole right	IB
WR	2	AMG1-2, HB1-2, TP1-2	Amygdala, Hippocampus, Temporal Pole	Cortico-subcortical blurring temporal pole left	IIIB
WR	3	SMADL1-2	Cingulate gyrus	Cortico-subcortical blurring cingulate gyrus left	IA
WR	4	AMG1-3, TP1-2	Amygdala, Hippocampus, Temporal Pole	Cortico-subcortical blurring temporal pole left	IB
WR	5	HB1-2, HH1-2, TP1-2	Hippocampus, Temporal Pole	Cortico-subcortical blurring temporal left	-

Table 2 Summary of clinical profiles and seizure-related data.

Clinical information for all participants, including dataset identifiers, seizure onset zone (SOZ) channels and corresponding anatomical structures, MRI findings, and postoperative seizure outcome when available (Engel classification).

The authors re-referenced the data using a bipolar montage. While bipolar referencing is optimal for localizing low-frequency activity (LFP), it is well-established that for gamma-frequency analysis, a monopolar reference (using nearby white matter or an acquisition-neutral white matter reference) is preferable (see Li, NeuroImage, 2018; Arnulfo, Nature Communications, 2020). I

understand that re-running the entire analysis in a monopolar reference scheme would require significant effort, but I recommend either: repeating the full analysis with a monopolar reference (better choice), or at least testing whether key findings hold (or are better represented) when applying monopolar referencing in a subset of subjects.

Choice of the optimal montage for detecting coordinated HFO bursting across widespread brain areas is very important and can heavily influence the analysis. In agreement with all of our previous studies of HFO detections, we decided not to use the raw signals with unipolar referencing to subgaleal electrodes because this method is sensitive to picking up non-local activities across the field from the common reference and resulting in spurious synchrony between detections from different contacts. Using common references from two contacts in acquisition-neutral white matter localization - suggested by the reviewer based on the two papers cited - is an interesting idea that is worth exploring. However, our new analysis in response to the next point below about the neutrality of the white matter contacts showed that some co-HFO bursting can still be recorded even from these contacts, suggesting that they are still picking up signals from the nearby gray matter. It would therefore be very challenging in this analysis to find a pair of white matter contacts that would be truly neutral.

In principle, we agree that it could potentially provide an optimal montage for this study. We have now proposed this option in the relevant Methods subsection about the electrophysiological recordings (line 645-648) for the readers' consideration. The revised text is pasted below:

Another montage option for HFOs detected across widespread brain areas is a unipolar montage referenced to two nearby contacts in the white matter (Arnulfo et al. 2020; Li et al. 2018), which was not used due to some co-HFO bursting detected in the white matter contacts (data not shown).

We have now also verified how much would applying the raw unipolar detections to the co-HFO bursting analysis change the main findings from Fig. 2C. Using these unipolar detections has not changed the main profile of significantly enhanced co-HFO bursting at the time of recall, even though the pattern for each task phase was not exactly the same as in our original bipolar montage. We are pasting below a figure from this additional analysis for reviewer's reference below. We conclude that the main effect of the peak co-HFO bursting at the time of recall persists even in this suboptimal raw signal analysis. With regard to the white matter referencing suggestion, please see the response below.

It is unclear whether the authors excluded white matter contacts from their analysis. Figure 4 suggests they did not. I recommend conducting separate analyses for white matter contacts to assess their potential impact on the results.

We thank the reviewer for this insightful suggestion. To clarify, our primary analyses included both gray and white matter contacts. In response to the reviewer’s recommendation, we conducted dedicated control analyses to assess the potential contribution of white matter signals.

First, we repeated the main analyses excluding all white matter contacts. This gray-matter-only analysis (plot ‘a’ below) replicated the key temporal dynamics observed in the original results—most notably, the prominent increase in co-HFO bursting around free recall onset and during word presentation in encoding. We also performed a complementary analysis restricted to white matter contacts. This white-matter-only analysis (plot ‘b’ below) still showed significant modulation of co-HFO bursting between the task phases with increased probability at the same times around vocalization in the recall phase and following word presentation in the encoding phase. The general pattern of differences between the four phases persisted in this white-matter-only analysis but the effects were substantially weaker and more variable than those seen in the gray-matter-only, partly due to the fact that there were less contacts localized in the white matter compared to the gray matter.

These results suggest that the observed co-HFO dynamics are primarily driven by coordinated activity in cortical (gray matter) regions, but these activities can still be detected from the nearby white matter contacts. Each white matter contact has thus been assigned to the anatomical localization of the nearby gray matter cortical region. We have included these additional analyses in the revised Supplementary Information as Supplementary Figure 9, which presents side-by-side comparisons of co-HFO burst rates for gray and white matter contacts. This new figure is now cited in the Methods section (line: 811 -820) to explicitly state that both gray and white matter contacts were included in the main analyses, and that these additional control analyses were conducted to assess their respective contributions. The revised text is pasted here for reviewer’s convenience:

Electrode contacts were anatomically localized using co-registered post-implant CT and pre-implant MRI scans. Tissue classification into gray and white matter was performed using standard cortical surface reconstructions and volumetric segmentation. The primary analyses included all contacts regardless of tissue type. To assess the contribution of white matter signals, we conducted control analyses restricted to gray matter contacts, and separately, using white matter contacts alone (see Suppl fig. 9). The gray-matter-only analysis reproduced the main temporal patterns, while the white-matter-only analysis revealed attenuated and more variable effects, suggesting that the observed co-HFO dynamics are predominantly driven by cortical activity.

We appreciate the reviewer's recommendation for this and the referencing analysis, which helped to strengthen the accuracy, interpretation and robustness of our findings in the anatomical space

Fig S9. Comparison of the global co-HFO bursting dynamics between electrode contacts localized in the gray and white matter. a) Mean co-HFO bursting across the significantly correlated channel pairs plotted as in Fig. 2c but restricted to gray matter contacts shows consistent increases around recall onset and during word presentation. b) The same analysis using only white matter contacts reveals analogous increases but less pronounced and more variable.

I know that the data are shared, but a supplementary figure showing SEEG implantation of each subject could help (see also my comments on statistics)

We agree that this supplementary figure would be helpful for the reader. As requested, we have now added a supplementary figure (Supplementary Fig. S8) that shows the SEEG implantation for each participant. This visualization displays the electrode contact locations and their percent involvement in co-HFO bursting events across the entire pool of the recalled words. By providing individual-level implantation maps, this figure enables a clear view of spatial coverage and variability across subjects, supplementing the statistical summaries presented in the main text. We hope that the reviewer will agree that the figure confirms widespread electrode contact coverage across all patients and similar proportions of electrode contribution to the global co-HFO bursting. The new figure (pasted in response to the comments below) is now cited in line 578-579 of the revised manuscript.

The distribution of contacts participating in co-HFO bursting during word recall is shown per subject in Suppl. Fig. 8.

3. Statistical Analysis

Given the small sample size (N=12), the authors might consider using mixed-effects models statistics to better account for inter-subject variability. This is especially relevant in intracranial studies, where individual differences in anatomical implantations can heavily influence group-level results.

To more accurately capture inter-subject variability inherent in intracranial recordings, we reanalyzed our data using a linear mixed-effects model (LMM) framework. This approach provides a more statistically principled test of condition effects, particularly well-suited for small-sample, high-resolution electrophysiological data like ours. For this analysis, we independently modeled each time bin using a mixed-effects design with Condition as a fixed effect and Subject as a random intercept (model specification: $\text{Value} \sim \text{condition} + (1|\text{Subject})$). This allows us to account for between-subject variability due to differences in electrode coverage and anatomical sampling, which are common sources of heterogeneity in iEEG datasets. The resulting omnibus p-values assess whether there is a significant effect of Condition at each time point, regardless of which conditions differ. Time bins that met the $p < 0.05$ threshold are now marked as black significance bars in the updated Figure 2 and 3, directly replacing the previous fixed-effects-based markers. For reference, the ANOVA (old) vs. FDR-corrected results for 2C are also provided below. These intervals largely recapitulate the timing of task effects identified in our original analysis, indicating that the observed dynamics are robust even when subject-level variance is explicitly modeled.

To directly address the reviewer's request for condition-specific comparisons, we extended this analysis by performing all six pairwise contrasts between task conditions (Recall, Countdown, Encode, Distractor). Using the same LMM structure, we rotated the reference level for Condition to extract each comparison while still controlling for the subject as a random effect. For each contrast, we then applied False Discovery Rate (FDR) correction across time to account for multiple comparisons. The full results of these pairwise comparisons are presented in Supplementary Figure 4, which displays FDR-corrected q-values for each condition pair across time bins.

This expanded analysis provides a detailed time-resolved map of task-specific differences. Notably, the pairwise q-value trajectories align with the timing and structure of effects seen in the omnibus test, while also revealing where specific condition pairs diverge. These findings offer stronger inferential resolution and further validate the task-related co-HFO dynamics reported in our main results.

Collectively, this reanalysis addresses the reviewer's concern with a more rigorous statistical approach. By incorporating subject-level random effects and controlling for multiple comparisons across time, we

demonstrate that the observed condition differences are not only qualitatively distinct but also statistically reliable. We believe this updated analysis strengthens the foundation of our claims and increases confidence in the generalizability of the findings. The changes are reflected Results section, Methods section, and in the corresponding figure caption. The revised text is pasted here for reviewer's convenience:

Results

Lines: 186-201

To account for inter-subject variability inherent in intracranial studies, we reanalyzed the data using a linear mixed-effects model (LMM) with Condition as a fixed effect and Subject as a random intercept. This approach tested for task-related effects in each 10 ms time bin, and significant intervals ($p < 0.05$) are indicated by black significance bars in Fig. 2b-d.

We further extended these models to all six pairwise comparisons between task conditions (Recall, Countdown, Encode, Distractor) by rotating the reference level for Condition within the same LMM structure. False Discovery Rate (FDR) correction was applied across time bins for each comparison. FDR-corrected trajectories are presented (Suppl. Fig. S4), providing a time-resolved map of condition-specific differences. These contrast effects aligned closely with those identified in the omnibus model, revealing condition-dependent divergence in co-HFO dynamics.

To assess the influence of the correlation threshold on the observed effects, additional analyses were performed using both more liberal ($r > 0.3$) and more conservative ($r > 0.7$) cutoffs. The temporal structure of co-HFO bursting across task phases was maintained across thresholds, consistent with the patterns described above.

Lines: 232-234

These time-resolved effects were tested using the previously described linear mixed-effects model (LMM) with Condition as a fixed effect and Subject as a random intercept.

Lines: 245-246

These differences were likewise assessed with the same LMM framework.

Lines: 894-908

Time-resolved comparisons of co-HFO bursting were analyzed using a linear mixed-effects model (LMM) implemented independently at each 10 ms bin. The dependent variable was the per-bin co-HFO probability, modeled as a function of task Condition (fixed effect) with a random intercept for Subject: $Value \sim Condition + (1|Subject)$. For significance testing, omnibus p -values for Condition were extracted at each bin. This framework was applied to all time-series probability analyses, including those comparing task phases (Fig. 2b–d) and those contrasting subsequently recalled vs. forgotten words and cued vs. free recall (Fig. 3b,c). For all pairwise contrasts, the Condition reference level was rotated within the same LMM structure and two-sided p -values were computed, with Benjamini–Hochberg false discovery rate (FDR) correction applied across bins for each comparison. Time points meeting the FDR threshold are plotted in Supplementary Fig. S4. This statistical approach accommodates subject-level heterogeneity in electrode sampling and improves inference reliability in small-sample intracranial datasets.

Fig. S4. Pairwise condition contrasts co-HFO bursting dynamics across time. Using linear mixed-effects modeling, all six comparisons between task conditions (Recall, Countdown, Encode, Distractor) were performed by rotating the reference level for Condition while controlling for subject as a random effect. FDR-corrected q-values are plotted across time bins for each condition pair.

- Along the same line, since the findings are largely presented at the group level, it would be valuable to demonstrate that the effects are consistent across participants, particularly given the limited sample size. More broadly, wherever feasible, group-level results should be supplemented with subject-specific visualizations.

We fully agree with the reviewer's point and have now included Supplementary Figure 8 mentioned in the response above, which presents per-subject visualizations of the percent involvement of each electrode contact in co-HFO bursting during word recall. The figure - pasted below for reviewer's convenience - reveals a consistent picture in each patient where most of the electrodes are involved in co-HFO bursting in response to similarly high proportion of words (dark dots). There are no individual cases that would deviate from this general picture either with a lower proportion of electrode contacts involved or words recalled from the pool. We believe that these results suggest similar temporal patterns of co-HFO bursting in time, even though the differences in electrode implantation coverage could result in some variance between individual patients, which would be confusing and difficult to read and interpret. Nevertheless, we can plot other results for individual patients if the reviewer found it necessary or helpful to show.

Fig. S8. Per-subject visualization of the involvement of each implanted electrode contact in co-HFO bursting during word recall. Each dot represents an electrode contact, color-coded by the proportion of recalled words in which it participated in a co-HFO burst.

4. Interpretation and Claims

I would set aside interpretations in the context of the **Global Neuronal Workspace Theory (GNWT)** and any potential implications for consciousness in general. Data reported here are rich and compelling without such considerations, and, most importantly, they were not recorded with the intention of studying consciousness.

Given that this interpretation may indeed seem quite speculative, we have discussed it only in the final paragraphs of the manuscript and noted in the text that we can only speculate about it without a more appropriate behavioral paradigm. The link between high frequency activities and conscious awareness of attended stimuli or experience in dreams from two example studies cited (Panagiotaropoulos et. and Siclari et al.) leading to this discussion point helped us to connect the global co-HFO bursting with the 'ignition' or 'global broadcasting' proposed in the global neuronal workspace hypothesis. This is also reminiscent of the global integration of information proposed for the co-HFO discharges in the recent papers from Eric Halgren's lab (Verzhbinsky et. al. 2024, Garrett et al. 2025). That is why we feel that it may be useful for the reader to at least consider this potential link.

At the same time, we fully understand how this may seem like a 'long-shot' for the reviewer, which is admittedly beyond the range and scope of this study. Hence, we have now revised the text in this paragraph to suggest a possible future direction for investigating the role of the global co-HFO discharges in the mechanisms of global broadcasting or integration of information for the contents of consciousness. Below we have pasted the revised paragraph below (lines: 526-546) for reviewer's consideration:

*There are other outstanding questions about the global HFO bursting that need to be addressed in future studies. First of all, can this widespread temporal coordination of local neural activities provide a mechanism to address the binding problem ^{1,52}? Recent studies suggest so ^{12,18,31} but testing it would require an appropriate behavioral paradigm probing conscious awareness of the perceived stimuli. **Neural activities in the high frequency ranges (above 50 Hz) were not only associated with encoding and recall of human memory ^{14,70,71} but also with explicit awareness of visual stimuli in monkeys ⁷² or conscious dream experiences in humans ⁷³. From the perspective of these studies, coordination of HFOs across the brain in coincident bursts could be linked to global integration of information and thus provide a plausible neural mechanism for concepts like 'ignition' or 'global broadcasting' proposed in the global neuronal workspace hypothesis ⁷⁴. Even though it is speculative at this point, one could nevertheless ask whether the sequences of bursting events correspond to a train of thought or a stream of consciousness. In the case of our paradigm, which is not designed to address these questions, we can only discuss potential future directions for testing the roles of co-HFO discharges in these functions; the sequences reported can, however, be explained more concretely in terms of a model for recalling engram representations of related word concepts from the memorized lists (see Fig. 5). Coincident HFO bursting presents a testable activity for these and other models of cognitive functions, including perceptual binding, conscious awareness, or engram representations of related concepts in the human mind.***

The finding that co-HFOs are not word-specific (Fig. 4) conflicts with engram literature (e.g., Vaz et al., 2020). Discuss whether this reflects macro-contact limitations or truly non-specific networks.

This is an excellent comment pointing to the crux of this work. We believe that it is indeed the limitation of using macro-contacts, which sample from large neural populations that participate in more than one engram. The study by Vaz et al. 2020 was using both macro- and micro-electrode recordings and showed word-specific firing patterns only for the latter (sequences of neuronal spiking to be more precise). This word-specific bursting of neuronal spiking was associated with ripple HFOs detected also on the adjacent macro-contact. The same macro-contact - localized together with the micro-electrode array in a word-processing area of the anterior temporal cortex - would also detect similar ripple HFOs in response to other words as well. The underlying neuronal spiking sequences would be different and specific to particular words but the emergent HFOs recorded on macro-contacts would be similar across multiple words.

We are currently preparing a follow-up study to confirm this hypothesis. Below is the main figure from that manuscript in preparation - shared here only for the purpose of this review - showing that the HFO detections on particular macro-contacts are actually specific to a subset of words, as assessed in another word screening (WS) task using the same pool of words as in the FR task. In the WS task, each word is pseudo-randomly presented five times in total to identify significantly more detections to particular words. We then tested during FR encoding and recall whether the same electrode contacts would show significantly more HFO detections in response to these 'preferred' words compared to all other words. The figure summarizes the time bins and anatomical structures, revealing significantly different HFO responses in the two tasks performed on the same day (same experimental session) and repeated on the following day. Our new results demonstrate persistent responses to specific subsets of words across different memory phases, tasks and days.

The degree of word specificity appears to be the lowest (i.e., a relatively greater proportion of words) in the sensory visual areas, and gradually increases (i.e., lower proportion of words) along the ventral visual stream in the temporal, limbic and prefrontal areas. As shown in another figure from that study pasted below, the sensory visual areas of the occipital cortex respond to 40-50% of all words in the pool. This proportion decreases (word specificity increases) in the more anterior, high-order processing areas down to approx. 10% in the limbic areas. The prefrontal

areas are also relatively specific (low proportion of words) - a breakdown into specific gyri shows significant HFO responses to less than 10% of words in the pool except for parts of the inferior frontal and precentral gyri, which were localized in the Broca's area responsible for speech. Hence, we believe that these macro-contacts detect HFOs generated by neuronal assemblies that are part of specific engrams, as proposed in our recent review (Kucewicz et al. 2024).

Both in this and the new follow-up study, we have also tested the HFO detections recorded on micro-contacts, expecting that they will show higher degree of word specificity. We have not been able to detect either the global co-HFO discharges in the current study or significant word responses in the follow-up study. This may be partly due to a much lower number of micro-contacts used in this study and effectively very sparse and inadequate sampling. Secondly, the design of both studies was to detect cortical areas that respond to word stimuli in general - very specific responses to single words would likely not be detected with the analysis tools that have been designed for these studies. Therefore, we do not exclude the possibility of capturing very specific engram responses on a micro-scale but it requires adjusting the analysis design in our current studies to further test our hypotheses. What we do know is that the single unit firing from selected patients from this cohort who had many micro-electrode contacts showed much higher degree of word specificity in the WS task (<5%) as expected (see the figure below from another manuscript in preparation).

A big picture from the current and the pending follow-up studies is that of gradually increasing word-specificity as we go down from the most macro-scale level of the global co-HFO bursting between multiple channels (50-60%), through HFOs detected on individual macro-contact channels in specific cortical areas (50-10%), all the way down to micro-scale level of neuronal spiking (<10%). This aspect of the current manuscript is not the main aim of the reviewed study and we are only now in the process of obtaining final results from the pending follow up studies.

In response to this comment, we have now revised the following paragraph in the Discussion (lines 484-504), refraining from mentioning any of the new results from our pending follow-up study.

To spot differences between such large-scale maps of two separate engrams for episodic memory traces would be possible on the level of single activated cells but challenging on the level of entire brain regions. That is likely why our macro-contact recordings resulted in responses to a large proportion of word items and relatively high overlap between them. Using micro- or meso-contact recordings and study designs presenting the same words multiple times would be expected to resolve more specific engram contributions. There are probably multiple distinct neuronal assemblies generating HFOs corresponding to different words that would be detected on the same macro-contact or distinct meso- or micro-contacts. A previous study by Vaz and colleagues confirmed that detecting single neuron firing on micro-contacts in parallel to HFO bursts on macro-contacts can be used to decode correct and incorrect recall of specific words from sequences of single cell firing¹⁰. The same group, more recently, showed that this temporal organization of unit firing carries specific information about the remembered stimuli³⁷. Our results suggest an analogous temporal organization of global co-HFO bursting on a macro-scale. In this study, we were limited to relatively low numbers of micro-electrode contacts, which prevented us from detecting micro-scale co-HFO discharges to specific words. Future studies employing combined macro- and micro-scale electrophysiological recordings^{24,65} and study designs adjusted to detect the hypothetical engram activities will not only track them with high spatial resolution but also in time to follow dynamically changing neuronal assemblies⁶⁶.

MINOR COMMENTS

- The audio version of the task is mentioned briefly. Clarify whether co-HFO patterns differed between modalities (could support sensory-independent effects).

Thank you for your interest and the prompt to demonstrate sensory-independent effects. Unfortunately, the audio versions of the tasks were run only with individual patients from this cohort and simply don't have enough N in either FR audio or PAL audio to compare with the effects described here. This goal is, nevertheless, on our list and will be possible to complete with more patients. The reason we have so few of these tasks now is that the battery of the tasks takes approx. 3h to run even without these additional audio. The audio versions are typically run toward the end of the recording sessions and by that time only the most dedicated patients agree to do more.

We would still like to mention these, anyway, to invite other researchers to investigate this aspect of the study. We have now added a reference in the relevant sentence (line 630-633) to the recently published open-access dataset paper, which will be constantly updated with new recordings. With time we hope to be able to answer these exciting questions about sensory independence and generalizability of our results to many multimodal memory tasks.

An additional audio version of the FR and PAL task was completed by some patients with the presented words and the recall cue word played out loud with a fixation cross on the computer

screen. **The open-access datasets are continually updated with additional patient recordings for analysis of these tasks (Cimbalnik et al. 2025).**

Table 1 was also updated to include information about the patients who completed the audio versions of the tasks.

Dataset	Language	Subject ID	Sex	Age	FR runs	PAL runs	Audio FR run	Audio PAL run	Micro contacts	Macro contacts
BR	CS	2	F	31	1	1	0	0	0	173
BR	CS	3	M	40	1	1	0	0	9	118
BR	CS	5	M	29	1	1	0	0	0	174
BR	SK	6	M	21	1	1	0	0	0	173
BR	CS	7	M	36	2	0	0	0	0	170
BR	SK	8	M	31	1	1	0	0	0	157
BR	SK	9	M	30	1	0	0	0	0	174
BR	CS	10	F	27	2	0	0	0	0	172
BR	SK	11	F	36	2	2	0	0	0	172
BR	CS	13	M	43	1	1	0	0	0	102
BR	CS	14	M	41	1	0	0	0	10	134
BR	CS	16	F	41	2	0	0	0	0	146
WR	PL	1	F	24	2	2	0	0	26	56
WR	PL	2	M	31	3	3	1	1	44	44
WR	PL	3	F	32	2	2	1	0	48	40
WR	PL	4	M	28	2	3	1	1	57	58
WR	PL	5	F	22	3	3	1	1	58	44
Total counts	CS - 8, SK - 4, PL - 5	17	10 M	31.9	34	24	4	3	282	2957

Temporal Resolution: The 10 ms bin size may obscure finer-scale HFO dynamics. Discuss whether higher temporal resolution (e.g., wavelet-based analysis) was explored.

We agree that the choice of bin size can influence the ability to capture fine-scale HFO dynamics. In our analysis, we represented HFOs as discrete events defined by their peak time and evaluated temporal patterns using raster matrices with a 10 ms bin size. This choice was guided initially by the typical duration of HFO bursts (10–50 ms) reported in the literature, as well as by our own empirical testing.

Specifically, we explored multiple bin sizes (5 ms, 10 ms, 50 m) in preliminary analyses. We found that:

- At 5 ms, the data became excessively sparse and noise-sensitive, reducing the interpretability of co-occurrence patterns.

- At 50 ms the binning window was too coarse, leading to over smoothing and loss of temporal specificity.
- The 10 ms bin size provided a balance between temporal precision and statistical stability, and produced the clearest and most interpretable structure in the co-HFO bursting patterns, particularly within the ± 50 ms coincidence window used to define co-activation events.

While we did not apply continuous time-frequency decomposition methods (e.g., wavelet analysis) in this study, we agree these may offer advantages for capturing transient or nested dynamics, and present an exciting avenue for future work. We now clarify the rationale behind our bin size selection in the revised Methods section.

Lines: 693 - 699

Raster matrices were constructed to visualize the temporal distribution of the HFO detection events, using a 10 ms bin size, this bin width was selected based on both the typical HFO duration (10–50 ms) and empirical comparisons across multiple resolutions (5, 10, 50, and 100 ms). Binning at 5 ms resulted in overly sparse representations, while 50–100 ms bins excessively smoothed temporal patterns. The 10 ms resolution provided the clearest and most stable co-HFO structure across trials. selected based on a typical HFO duration of 10–50 ms. Time zero for each task phase was defined at digit presentation (countdown), word presentation (encode), arithmetic task onset (distractor), and onset of remembered word vocalization (recall). Rasters were aligned to these event times within a ± 1.5 s window.

Lines: 703-708

To assess spatiotemporal relationships, HFOs were binarized (1 for HFO presence, 0 for absence), and a ± 50 ms window (± 5 bins) were examined across channels to identify coincident HFO bursts (co-HFOs) also called co-HFO bursting events. Although our primary approach used discrete event representations, future analyses using continuous, time-frequency methods such as wavelet-based decomposition may yield additional insight into finer-scale HFO dynamics.

Some figures are cited out of order (e.g., Fig. 5 appears before Fig. 4 in the text). Ensure chronological flow.

All the figures are now cited in the right order when reporting the results. Other references to the same figures beyond the Results section are preceded with 'see' (e.g., see Fig. 1c) to mark that it is not the original citation of the figure results.

Define all abbreviations at first use (e.g., PAL in the Introduction).

Both the FR and the PAL tasks are now introduced with their abbreviations defined in the final paragraph of the Introduction (lines 100-103). The revised sentence is pasted below for reviewer's convenience:

In this study, we investigate these coincident HFO discharges in response to four types of events related to preparation, encoding, distraction and recalling of stimuli in a Free Recall (FR) and also in a Paired-Associate Learning (PAL) verbal memory tasks to test their possible behavioral functions.

The other abbreviations used in the manuscript were checked for their first-use definition and found to be correct, apart from EEG - considered to be a standard abbreviation that does not require the definition.

For hierarchical clustering (Fig. 5), add silhouette scores or bootstrapping to assess robustness.

This is an excellent suggestion, which was also mentioned by the other reviewers to evaluate the quality and robustness of our hierarchical clustering. To this end, we computed silhouette scores across a range of cluster numbers ($n = 2-20$). The maximum silhouette score observed was 0.142. While this value is modest, such scores are common in high-dimensional, noisy, and temporally overlapping neural data such as high-frequency oscillations (HFOs), where cluster boundaries may be diffuse. This score suggests weak but non-random structure in the data, and highlights the importance of using complementary approaches to validate clustering results.

Accordingly, we conducted two additional assessments of robustness. First, a bootstrapping procedure ($n = 100$ resamples) yielded a mean Adjusted Rand Index (ARI) of 0.496, indicating moderate consistency in the derived module structure across resampled datasets. Bootstrapping was performed independently at the level of each recalled word: for each word, channels of the co-HFO raster were sampled with replacement, hierarchical clustering was re-applied using correlation distance, and cluster labels from the bootstrapped data were compared to the original labels using the Adjusted Rand Index. This yielded a distribution of ARI scores per word, and the final reported value reflects the mean ARI across all words and subjects.

Second, to test whether the observed sequential structure reflects true temporal organization rather than noise or spurious correlations, we implemented a surrogate jitter control suggested by another reviewer. HFO timestamps were randomly perturbed within a $\pm 50-100$ ms window and clustering was repeated. To quantify disruption, we computed a “synchrony” metric defined as the mean pairwise Pearson correlation across all channels in the co-HFO rasters. This value ranges from 0 (no synchrony) to 1 (perfect synchrony). Synchrony was computed independently for each recalled word in both the original and jittered conditions, and the reduction (~ 0.47) reflects the average difference across all words and patients. (suppl Fig S7) This substantial drop indicates that the original temporal modules are disrupted by jittering, supporting the interpretation that the observed sequential co-HFO patterns reflect meaningful temporal structures.

These additional analyses have now been incorporated into the Results and methods section of the main text. Specifically, we report the mean Adjusted Rand Index (ARI) from the bootstrapping procedure, and the results of the surrogate jitter control analysis, including the reduction in synchrony. While these results are not shown in an updated figure, they provide important complementary evidence supporting the robustness and non-randomness of the hierarchical clustering and have been described in detail to enhance transparency and interpretability. These changes have been made in the main manuscript and are pasted below for the reviewer’s reference.

Results

Lines: 318-341

To confirm that these sequential modules reflect genuine temporal structure rather than spurious correlations, we implemented a surrogate jitter control. HFO timestamps were randomly perturbed within a $\pm 50-100$ ms window, preserving overall burst rates but disrupting precise timing. This led to a marked reduction in co-HFO synchrony, measured as the mean pairwise Pearson correlation across channels, indicating that millisecond-scale precision is critical for the observed modular structure (Supplementary Fig. S7).

We further assessed the robustness of clustering using a bootstrap resampling procedure in which contact channels were resampled independently for each recalled word. The Adjusted Rand Index (ARI) between clustering solutions from original and resampled data yielded a mean ARI of 0.496 ± 0.021 , where 0 indicates chance-level similarity and 1 indicates identical cluster assignments, suggesting moderate clustering stability across trials.

Encoding trials, although generally exhibiting lower overall co-HFO bursting compared to recall (Fig. 2b), also showed evidence for non-random temporal organization: a surrogate jitter analysis confirmed that these sequences reflect structured timing at the millisecond scale. However, clustering stability during encoding was reduced relative to recall (mean ARI = 0.354 ± 0.026 vs. 0.496 ± 0.021 ; paired $t(17) = 7.21$, $p < 0.0001$) and synchrony differences after jitter were smaller (paired $t(17) = 4.83$, $p = 0.0002$), indicating weaker temporal coordination.

Forgotten recall trials displayed clustering stability (mean ARI = 0.472 ± 0.024) not significantly different from remembered trials (paired $t(17) = 1.32$, $p = 0.20$) and synchrony values broadly comparable across participants, without consistent differences.

Methods

Lines: 864-877

To test whether these structures depend on fine-scale timing, we applied a temporal jitter control in which HFO timestamps were randomly shifted within a ± 50 – 100 ms window. Clustering and synchrony metrics were recomputed on the jittered data and compared to the original to assess the role of precise temporal alignment. Synchrony was defined as the mean pairwise Pearson correlation between binary HFO time series across channels for each trial, providing a global index of temporal coordination (range: 0–1). To assess clustering reproducibility, we used a bootstrap resampling approach in which channels were resampled independently per word. Similarity between original and resampled clustering labels was quantified using the Adjusted Rand Index (ARI), which measures the agreement between two clustering solutions by comparing how many element pairs are grouped together or apart in both, adjusted for chance. ARI values range from -1 (complete disagreement) to 0 (random labeling) to 1 (perfect agreement).

Lines: 927-935

For the sequential co-HFO analyses, clustering stability (Adjusted Rand Index, ARI) and synchrony differences after jitter were compared between conditions using two-tailed paired t -tests for normally distributed data or Wilcoxon signed-rank tests otherwise. This analysis tested recall vs. encoding and recall vs. forgotten recall to determine whether the temporal organization of sequential co-HFO patterns differed across memory conditions.

CONCLUSIONS

The study provides valuable insights into the spatiotemporal organization of co-HFOs during memory tasks. Addressing the methodological and interpretational concerns above would, in my view, significantly elevate the manuscript's impact. With revisions, this work could become an

important step forward in understanding large-scale neural coordination in memory. Finally, I commend again the authors for their comprehensive dataset and commitment to open science. The work is ambitious and timely, and I look forward to seeing the revised manuscript.

Thank you, Andrea, for the kind words and your appreciation of our efforts toward an open science and understanding the large-scale coordination of memory processing. We hope the results and the data will be useful to the community for further exploration.

Reviewer #2:

This study by Prathapagiri et al. builds on rodent and human work showing that high-frequency oscillations (HFOs) co-occur across cortical regions up to ~20 mm apart (“co-HFOs”). Micro and macro-contact stereo-EEG in 17 epilepsy patients (electrode positions from co-registered CT/MRI), were used. Participants performed a Free Recall task (180 words, 15 lists) and a Paired-Associate Learning task (6 word-pairs, 15 lists). The authors applied a refined detection pipeline to extract HFO events, binarized these events into 10 ms time-bins, and computed Pearson correlations over these bins across channels. A threshold of $r > 0.5$ defined co-HFOs. They then averaged co-HFO rates across four task phases (preparation, encoding, distractor, recall), frequency bands, and cortical regions.

Main findings:

- . Co-HFOs occur during all task phases but are most strongly modulated during memory encoding and recall across visual and higher-order cortical areas.
- . For subsequently recalled words, co-HFO bursting is suppressed immediately before stimulus onset and enhanced during word presentation.
- . Hierarchical clustering reveals that during recall co-HFOs form temporal sequences spanning multiple, spatially distributed networks.

Overall, the results are highly relevant, offering a new perspective on how HFOs and co-HFOs contribute to memory processes. However, I have some concerns regarding aspects of the data analysis, interpretation, and participant reporting, which I have detailed below.

We would like to thank the reviewer for the time spent on carefully reading the manuscript and suggesting the changes to improve our analysis, interpretation and participant reporting. We are glad that the main findings and new contributions of this work to the field summarized by the reviewer were clearly communicated in the manuscript.

Strengths:

1) The identification of “synfire chain”-like sequences of co-HFO activations across distributed cortical sites is particularly compelling. The authors relate this finding to broader frameworks such as “binding theory,” which is a useful conceptual anchor. However, since their paradigm is limited to verbal memory, further work is needed to explore whether these dynamics generalize to multisensory integration processes more typically associated with the binding problem. Nonetheless, these results pave the way for targeted experiments probing whether co-HFO dynamics might serve as a general neural substrate for perceptual binding. With additional analytical validation (e.g., using methods suggested below), this discovery will have far-reaching implications for Neuroscience.

2) A clear strength of this study is its cohort size: with 17 patients, it sits in the upper range of human intracranial HFO/memory investigations, providing greater statistical power and confidence in the generalizability of its findings.

3) By combining Free Recall and Paired-Associate tasks - and even audio versus visual versions - the study convincingly dissociates encoding, maintenance, distraction, and retrieval, while controlling for sensory and motor confounds.

4) By including native Czech, Slovak, and Polish speakers, the study broadens its applicability beyond the usual English-dominant cohorts, enhancing cross-linguistic generalizability.

We appreciate that the reviewer recognized that the datasets comprised patients from multiple international centers performing the same battery of tasks in different languages. This was not an easy endeavor to collect these recordings, which took us several years. Not all patients completed all the tasks in the battery and that is why some of them, e.g., the audio version of the FR and PAL tasks, did not have enough patients recorded to complete all the analyses. Nevertheless, the main effects of the global temporal organization of HFO discharges across the entire cortex were found to persist in all patients and across different behavioral paradigms. In our responses to yours and the other reviewers' comments, you will find additional results from the analysis suggested in this review that strengthens the importance of this temporal organization, including jittering of the HFO detection times or changing the temporal window for coincidence detection.

Major comments:

1) I am skeptical of the arbitrary choice of an $r > 0.5$ threshold to define co-HFOs. This dichotomization introduces the risk of misclassifying marginal cases (e.g., $r = 0.499$ vs. 0.501) and may obscure meaningful variation in the data. Given that HFOs are already binarized, the stated rationale of 'mitigating noise' is unconvincing. For the core analysis presented in Figure 2, I recommend treating r as a continuous predictor. Alternatively, a sensitivity analysis across a range of thresholds (e.g., 0.3 to 0.7) would help demonstrate that the findings are not an artifact of a single, arbitrary cutoff.

This is a very pertinent point regarding the use of a fixed threshold ($r > 0.5$) to define co-HFOs. We have decided to use this fixed threshold to show that even this relatively conservative approach results with considerable high proportions of electrode contacts involved in response to more than half of the words in the pool on average. Lowering the threshold to more liberal levels like $r > 0.3$ resulted in even higher proportions, suggesting that most of the electrode contact pairs show considerable correlation values. Notice, for instance, that in Fig. 2b, the gray plot for the recall phase is not completely 'flat' suggesting that there is still some co-HFO bursting among these subthreshold electrode contacts. Even at the liberal 0.3 threshold, there could still be observed a slight increase at the time of recall. In the end, we also ran the key analyses in this study with all contacts (no thresholding), obtaining a similar though more 'noisy' pattern of co-HFO bursting. Hence, for the sake of clarity and interpretation we decided to use the conservative 0.5 threshold in the manuscript.

To address this concern in a more systematic fashion, we repeated the main analyses shown in Figure 2 across a range of correlation thresholds ($r = 0.3$ and $r = 0.7$). As detailed in the results pasted below, the core task-related modulation and temporal structure of co-HFO bursting remained robust across this range. While the absolute number of detected co-HFOs naturally varied depending on threshold stringency, the relative effects across task phases and the group-level patterns remained consistent. These findings indicate that the main results are not contingent on a specific correlation cutoff, but rather reflect stable and reproducible differences in co-HFO dynamics across task conditions. While modeling r as a continuous predictor is an interesting alternative, we prioritized a threshold-based approach to

maintain interpretability and to clearly distinguish strongly correlated channel pairs from weaker ones. These changes have been made in the main manuscript and are pasted below for the reviewer's reference. These revisions correspond to lines 198–201 in the Results section and lines 734–741 in the Methods section pasted below respectively.

To assess the influence of the correlation threshold on the observed effects, additional analyses were performed using both more liberal ($r > 0.3$) and more conservative ($r > 0.7$) cutoffs. The temporal structure of co-HFO bursting across task phases was maintained across thresholds, consistent with the patterns described above.

Key analyses were also repeated using alternative correlation thresholds ($r > 0.3$ and $r > 0.7$) to examine the effect of this parameter choice. The $r > 0.5$ threshold was selected to prioritize moderately strong correlations that are more likely to reflect meaningful co-variation in HFO activity, without excluding a large proportion of relevant channel pairs. This criterion provided a practical balance between interpretability and signal reliability across subjects.

2) The authors argue that their findings on co-HFO contributions to cognitive processes are generalizable beyond the specific patient cohort studied. I am nearly convinced, given the overlap with prior reports on co-HFOs. However, because their approach closely mirrors that of earlier studies involving similarly composed patient populations, there is a risk of establishing a self-reinforcing framework in which disease-related influences are systematically overlooked. To enable readers to critically assess the extent of generalizability, more detailed information about the clinical characteristics of the included patients is essential and would strengthen the scientific robustness of the work.

I recommend addressing this in a dedicated paragraph within the Methods section. For clarity, the current “Participants and Electrode Localization” section could be split - electrode localization details might be more appropriately placed in a new “Electrophysiological Recordings” subsection. The clinical description should include relevant information such as epilepsy etiology (e.g., hippocampal sclerosis, tumor, cortical malformation), disease severity (seizure frequency), duration and class of pharmacological treatment, and the laterality and localization of the epileptic focus (e.g., left vs. right hemisphere; temporal vs. extratemporal).

A related aspect that requires clarification is handedness. Was hemispheric dominance considered in the analysis? Specifically, did HFO occurrence or co-HFO frequency differ between the dominant and non-dominant hemispheres? Addressing these questions could reveal potentially relevant lateralization effects. This aspect appears to be hinted at in Figure 4a, which

shows bilateral HFO occurrence during recall of a single word, but it is not systematically explored in the current analysis.

We have now prepared a table with the requested clinical information and added a new paragraph in the first subsection of Methods to explicitly state the limitations of this study associated with including potential pathological activities (inside and outside of the seizure onset zone) in the analysis of physiological activities. Below we have pasted the new paragraph from the Methods subsection together with the table cited in the text (lines 580-597).

All participants recruited in this study were patients suffering drug-resistant epilepsy with various clinical backgrounds of the disease and anatomical localizations of the seizure onset zones (Table 2). Hence, the intracranial recordings in these patients are inherently limited by the pathological discharges, including the interictal epileptiform spikes and HFOs, especially in the fast ripple frequency ranges. We cannot fully exclude a potential influence of these epilepsy-related activities on our methodology and the reported results, even though our design was based on detecting HFOs consistently around cognitive events associated with memory processing, which would be expected to average-out any potential pathological activities. We previously reported an interaction between memory processing and incidence of epileptiform discharges in these patients, showing a lower probability of their occurrence during memory processing (Matsumoto et al. 2013). Also, most of the coincident discharges reported here comprised HFOs detected at frequencies lower than 250 Hz, which is thought to be a lower threshold for pathological oscillations (Valero et al. 2017). Overall, pathological discharges are not expected to be consistently induced by memory processing and analyses across large numbers of trials should effectively diminish any potential influence of their occurrence, even though it can never be completely ruled out when studying any physiological function of the brain.

With regard to the handedness issue, it is not part of the standard protocol to determine hemispheric language-dominance in our clinical centers. Patient report of handedness does not necessarily determine the left or right hemispheric dominance. Nevertheless, irrespective of the handedness, most patients would show language dominance in the left hemisphere based on the proportions in the general population. Hence, we interpret the laterality observed in Fig. 4a as confirming this general trend expected in case of the verbal tasks using words as the items to remember.

In addition to the cumulative plot of all electrodes implanted in Fig. 4a, we have now generated equivalent plots for each patient in the supplementary material (pasted below for reviewer's reference). In general, there are too few patients and too limited sampling of electrode implantation to properly address this issue of hemispheric laterality of co-HFO sampling. We do agree, however, that adding these plots provides the reader with a more complete picture of detecting coincident HFO bursting in any one patient.

Fig. S8. Per-subject visualization of the percent involvement of each implanted electrode contact in co-HFO bursting during word recall. Each dot represents an electrode contact, color-coded by the proportion of recalled words in which it participated in a co-HFO burst.

3) How did task performance vary across participants, and was there a relationship between each patient's co-HFO abundance and their memory performance?

Across participants, recall performance averaged $30.7\% \pm 8.0\%$ (mean \pm SD), with individuals recalling between 12.8% and 44.4% of words per list (see new Supplementary Table 1). To determine whether overall co-HFO activity was associated with memory performance, we examined the relationship between

the total number of co-HFOs per word per subject event during recall and correlated this with recall accuracy. No significant correlation was observed ($r = 0.24$, $p = 0.445$), indicating that the abundance of co-HFO burst events during recall was not predictive of task performance. These findings suggest that while co-HFO patterns reveal meaningful temporal structure, individual differences in recall accuracy were not explained by total co-HFO bursting rates. These changes have been made in the main manuscript and are pasted below for the reviewer's reference. The revisions can be found in lines 129-132 and 160-164

Recall performance averaged $30.7\% \pm 8.0\%$ (mean \pm s.d.), ranging from 12.8% to 44.4% across participants (Suppl. Table 1).

To test whether co-HFO abundance tracked behavioral performance, we correlated each participant's total number of co-HFO bursts during recall with their recall accuracy. No significant relationship was observed ($r = 0.24$, $p = 0.445$), suggesting that overall co-HFO activity was not predictive of memory performance (Suppl. Table 1).

Subject	Recalled Words (n)	Recall Rate (%)	Rates of Co-HFO bursts (Hz)
2	50	27.8	2934.920
3	68	37.8	2441.392
5	62	34.4	2608.656
6	42	23.3	2782.087
7	55	30.6	2466.364
8	55	30.6	2064.073
9	63	35.0	2618.757

10	47	26.1	2525.816
11	80	44.4	2956.562
13	48	26.7	1685.500
14	63	35.0	1802.169
16	23	12.8	2191.507
Average	54.3	30.7	2423.150

Table S1. Recall performance and co-HFO bursting rate abundance across participants.

This table summarizes the total number of words recalled, recall rate (as a percentage of studied words), and rates of co-HFO bursts during the recall period for each participant. These values were used to assess the relationship between task performance and co-HFO activity, as reported in the main text.

4) On the sequential occurrence of co-HFOs: In the second-to-last section of the Results, the authors introduce their analysis by asking whether global co-HFOs occur “during encoding or recalling a word.” However, the results presented focus exclusively on sequential co-HFOs during recall. What were the corresponding findings for encoding? Does encoding similarly involve sequential, synfire chain-like activation patterns?

In addition, was the analysis of sequential co-HFOs extended to data from the PAL (paired-associate learning) task? And relatedly, do co-HFOs associated with failed recall (“forgotten” cases, as shown in Fig. 3b) also exhibit sequential activation?

Contrasting these different conditions – encoding, successful recall, and failed recall – could be highly informative and would help determine whether the observed sequential co-HFO patterns are specific to successful memory retrieval.

In the original submission, we focused primarily on recall trials when investigating temporally ordered co-HFO patterns. To address the reviewer’s comment, we re-applied the hierarchical clustering and synchrony analyses to encoding trials. Although encoding trials generally exhibited lower overall co-HFO bursting compared to recall (as shown in Fig. 2b) a surrogate jitter analysis (± 50 – 100 ms) confirmed that these sequences during encoding reflect non-random temporal patterns. However, compared to recall (mean ARI = 0.496 ± 0.021), encoding-related sequences showed reduced clustering stability (mean ARI = 0.354 ± 0.026 ; paired $t(17) = 7.21$, $p < 0.0001$) and smaller mean synchrony differences after jitter (paired $t(17) = 4.83$, $p = 0.0002$). This suggests that while sequential co-HFOs occur during encoding, their organization is weaker than during successful memory retrieval. Forgotten recall trials showed

clustering stability (mean ARI = 0.472 ± 0.024) not significantly different from remembered trials (paired $t(17) = 1.32$, $p = 0.20$) and similar synchrony values. PAL task data, though available from fewer patients, yielded results consistent with the FR task. Overall, sequential co-HFO patterns were most prominent during successful recall, weaker during encoding, and reduced in failed recall, consistent with a role in engram reactivation rather than formation. This result is consistent with a functional role of these patterns in engram reactivation rather than formation. Together, these extended analyses indicate that sequential co-HFO patterns are most prominent during successful memory retrieval, with comparatively weaker organization during encoding and reduced expression in failed recall. While a full mechanistic exploration of these patterns lies beyond the scope of the present study, our findings establish a clear foundation for such future work, and we are actively investigating the broader phenomenon of global sequential bursting in ongoing experiments. The following changes have been made in the main manuscript and are pasted below for the reviewer's reference. These revisions correspond to lines 318–341 in the Results section and lines 864–877; 927–932 in the Methods section.

Results

To confirm that these sequential modules reflect genuine temporal structure rather than spurious correlations, we implemented a surrogate jitter control. HFO timestamps were randomly perturbed within a ± 50 – 100 ms window, preserving overall burst rates but disrupting precise timing. This led to a marked reduction in co-HFO synchrony, measured as the mean pairwise Pearson correlation across channels, indicating that millisecond-scale precision is critical for the observed modular structure (Suppl. Fig. S7).

We further assessed the robustness of clustering using a bootstrap resampling procedure in which contact channels were resampled independently for each recalled word. The Adjusted Rand Index (ARI) between clustering solutions from original and resampled data yielded a mean ARI of 0.496 ± 0.021 , where 0 indicates chance-level similarity and 1 indicates identical cluster assignments, suggesting moderate clustering stability across trials.

Encoding trials, although generally exhibiting lower overall co-HFO bursting compared to recall (Fig. 2b), also showed evidence for non-random temporal organization: a surrogate jitter analysis confirmed that these sequences reflect structured timing at the millisecond scale. However, clustering stability during encoding was reduced relative to recall (mean ARI = 0.354 ± 0.026 vs. 0.496 ± 0.021 ; paired $t(17) = 7.21$, $p < 0.0001$) and synchrony differences after jitter were smaller (paired $t(17) = 4.83$, $p = 0.0002$), indicating weaker temporal coordination.

Forgotten recall trials displayed clustering stability (mean ARI = 0.472 ± 0.024) not significantly different from remembered trials (paired $t(17) = 1.32$, $p = 0.20$) and synchrony values broadly comparable across participants, without consistent differences.

Methods

To test whether these structures depend on fine-scale timing, we applied a temporal jitter control in which HFO timestamps were randomly shifted within a ± 50 – 100 ms window. Clustering and synchrony metrics were recomputed on the jittered data and compared to the original to assess the

role of precise temporal alignment. Synchrony was defined as the mean pairwise Pearson correlation between binary HFO time series across channels for each trial, providing a global index of temporal coordination (range: 0–1). To assess clustering reproducibility, we used a bootstrap resampling approach in which channels were resampled independently per word. Similarity between original and resampled clustering labels was quantified using the Adjusted Rand Index (ARI), which measures the agreement between two clustering solutions by comparing how many element pairs are grouped together or apart in both, adjusted for chance. ARI values range from - 1 (complete disagreement) to 0 (random labeling) to 1 (perfect agreement).

For the sequential co-HFO analyses, clustering stability (Adjusted Rand Index, ARI) and synchrony differences after jitter were compared between conditions using two-tailed paired t-tests for normally distributed data or Wilcoxon signed-rank tests otherwise. This analysis tested recall vs. encoding and recall vs. forgotten recall to determine whether the temporal organization of sequential co-HFO patterns differed across memory conditions.

Fig. S7. Surrogate jitter analysis reveals temporally structured co-HFO bursting during encoding and recall. a) Word-level differences in mean synchrony (original minus jittered) across recalled words for a representative subject (subject 2). b) Distributions of mean pairwise synchrony values (Pearson correlation across co-HFO matrices) for original and jittered datasets during encoding and recall.

5) Given the potential impact of demonstrating sequential co-HFOs, the authors could reinforce their findings — beyond hierarchical clustering — by applying one or more of the following approaches:

A useful control could be the application of a surrogate jitter test: Randomly perturb each HFO timestamp by e.g. ± 50 -100 ms within trials, re-run the clustering, and assess whether the sequential modules persist. If they disappear or weaken, this would support the claim of true temporal order.

Another option would be cross-correlogram analysis: for each channel pair in a cluster, compute their HFO-train cross-correlation. Observing consistent peaks at fixed positive lags - aligned with the clustering sequence - would further corroborate a sequential propagation of HFOs.

This is one of the most important suggestions to strengthen the main findings of this work. Other reviewers have also proposed similar analyses to confirm the temporal organization of co-HFO bursting.

We have now run the jitter test to validate this temporal organization and are pleased to report that the recommended surrogate analysis supports criticality of precise global coincident bursting in the observed temporal patterns.

Following the reviewer's recommendations, we implemented a surrogate jitter test by randomly perturbing each HFO timestamp by ± 50 -100 ms within trials. This approach preserves the overall HFO firing rates while destroying precise temporal relationships, providing a robust control for testing whether our observed sequential modules represent genuine temporal structure rather than spurious correlations. The jitter test results demonstrate a clear separation between the original and the new jittered synchrony patterns, as shown in the new Supplementary figure 7, which is now cited in the revised manuscript (line 293). The new synchrony measure, defined as the mean pairwise Pearson correlation across all channels in the co-HFO matrix, was found to be very consistently reduced by the jittering procedure across the recalled and encoded words, and across subjects (Supple. Fig. 7). We have now summarized the distributions of synchrony values estimated from all encoded and recalled word trials in the final plot of the supplementary figure to show that:

- the original data showed clustering around specific synchrony values, indicating structured temporal relationships between HFO events across the electrode contact sites.
- the jittered data showed a broader distribution with significantly reduced peak values, consistent with elimination of a precise temporal coordination.

This clear separation between the original and the jittered data confirms a temporal structure in the reported coincident HFO bursting. Dispersion of sequential co-HFO bursting due to the jittering procedure demonstrates that our original hierarchical clustering was capturing actual propagation patterns rather than chance temporal coincidences, as shown in the recall of an example word shown in Fig. 5a and tested in the first plot of the new supplementary figure.

We conclude that: (1) the preservation of overall HFO rates while destroying temporal precision rules out potential artifacts from differential firing rates between brain regions; (2) the selective disruption of synchrony patterns confirms that our clustering depends on millisecond-scale temporal precision; (3) the clear separation between original and surrogate distributions provides statistical evidence against the null hypothesis of random temporal relationships.

While the jitter test provides strong validation of a general temporal structure, it does not address the issue of clear temporal sequences that could be assessed using cross-correlogram analysis as suggested by the reviewer. Although pairwise cross-correlations between HFO trains in each cluster have not revealed consistent lags, we believe that cross-correlations between pairs of channels from neighboring clusters would confirm the observed sequences in co-HFO bursting. We are now in the process of implementing this analysis in a follow-up study, which would go beyond the scope of this manuscript. Identification of distinct clusters and assigning intra- vs inter-cluster channel pairs from various cortical lobes is a major challenge that we are facing at the moment.

We have now cited the new supplementary figure (pasted below) and the new results (line: 185-293 and 730-738) in the revised manuscript.

Fig. S7. Surrogate jitter analysis reveals temporally structured co-HFO bursting during encoding and recall. a) Word-level differences in mean synchrony (original minus jittered) across recalled words for a representative subject (subject 2). b) Distributions of mean pairwise synchrony values (Pearson correlation across co-HFO matrices) for original and jittered datasets during encoding and recall.

Further comments:

For this revision - and for future manuscripts authored by these colleagues - I strongly recommend enabling line and/or page numbering. This small adjustment requires minimal effort from the authors but significantly improves the efficiency of the review process, as it allows reviewers (and authors in their replies) to reference specific parts of the text with ease.

We have now added line numbers in the revised manuscript and refer to changes in specific lines. Thank you for suggesting this easy change to facilitate the review process.

At times, the narrative feels somewhat diffuse. I suggest tightening the focus throughout the manuscript — particularly in the Results and Discussion sections — to help ensure each part conveys a clear and targeted message.

Nature Communications allows subsection headings, which we have now added in the Results and Discussion sections to tighten the focus by formulating clear messages of each heading. Thank you for this excellent suggestion.

The subsection headings were added in the following lines: 120,165,216,253,302,366,423,472,523

The term ‘burst’ needs clarification. Do you use it to mean a cluster of closely spaced HFO events, or something else? In electrophysiology, a ‘burst’ usually refers to a brief train of spikes (or bursts of neuronal firing), whereas here you’re detecting oscillatory events. Please define what constitutes a ‘burst’ in your HFO analyses.

This is an important point not only for this manuscript but also for the field. In all of our previous studies investigating individually detected HFOs we have used the term ‘burst’ to describe one distinct instance of

an oscillation crossing the detection thresholds in the frequency-time space. To our knowledge, this definition was first used in a study by Lachuaux et al. published in the European Journal of Neuroscience (doi.org/10.1046/j.1460-9568.2000.00163.x) to describe HFOs induced in response to visual stimuli in the gamma frequency ranges. The authors of that original paper called them: 'short oscillatory bursts' to point to their defined spread in time and frequency space. Since then, multiple micro-electrode studies have shown that these HFO bursts are generated by bursts of single unit firing of action potentials, including the recent papers cited in the manuscript like Vaz et al. 2020 or Tong et al. 2021. Therefore, the bursts of HFO can easily be confused with bursts of unit spiking but actually they are mechanistically related.

On the next level of organization, multiple distinct bursts of HFOs recorded from different sources could also be called a global 'burst' event, as the reviewer suggested. We were careful not to call this global event a burst (singular) but rather refer to it as coincident 'bursts' (plural) or 'bursting', pointing to the multiple sources across the brain. So the global bursting events are composed of multiple individual bursts of HFOs or, in other words, every co-HFO event is composed of multiple HFO bursts. We also occasionally used a more general term 'discharge', especially when discussing the detections in context of a wider frequency spectrum, including gamma and lower frequencies. Analogous discharges in the frequency-time space have also been reported in the theta frequency ranges and called 'bouts' (see Seeber et al. 2025). A discharge thus could mean a ripple or gamma HFO burst or a theta bout. The same rule would apply to describe coincident discharges in plural.

We have now carefully inspected each of 125 instances of using the term burst to verify if each one of them is according to the above-mentioned definition.

Results, 3rd paragraph: "Pearson's co. relation" -> correlation.

Corrected in line 171 of the revised manuscript.

Methods: In the Detection of individual HFO bursts section, the authors state that "...greater artifact susceptibility necessitates more stringent detection criteria than those typically applied in rodent studies." However, artifacts also pose significant challenges in behaving-rodent recordings, where investigators routinely combine multiple criteria to exclude false-positive ripples. Defining robust HFO detection methods therefore remains an active area of debate (e.g., Navas-Olive et al., eLife 2022; Liu et al., Nat. Commun. 2022).

We agree that even in the rodent studies with behaving animals it also remains a major challenge to detect true HFOs in the ripple frequency ranges. It is different in the human patient recordings since there are pathological activities in naturally occurring epilepsy compared to artificially-induced epileptiform activities in the rodent-models of pharmacologically (e.g., pilocarpine) or electrophysiologically (e.g., kindling) induced epilepsy. What we actually meant in the cited sentence was not differentiating true physiological or pathological HFOs but rather the detections, which comprise no actual oscillations but an elevated increase in the power or amplitude envelope due to filtering of sharp-transitions in the signal and non-physiological artifacts. By comparing the detection criteria in the rodent studies we specifically meant the hippocampal sharp-wave ripple complexes, which are very easily detected even manually in the raw signal with a naked eye. The human ripple HFO equivalents are not so clear, as described in the cited studies.

This sentence has now been clarified as follows in lines 680-686:

This approach was designed to reduce false positives in the human intracranial recordings of physiological HFOs, which can pose challenges to the detection and differentiation from pathological epileptiform activities or non-oscillatory increases in the power or amplitude envelope due to filtering of sharp-transitions in the signal and non-physiological artifacts (Liu et al. 2022). Even the detection of the classic hippocampal sharp-wave ripple complexes in rodents prompts development of more advanced classification criteria (Navas-Olive et al. 2022).

- **Methods:** Analysis of coincident HFO bursting, 1st paragraph: There appears to be a discrepancy between the frequency bands listed in the text ('60–150 Hz, 150–250 Hz, and 250–500 Hz') and shown in Figure 2 with those shown as illustration of band separation in Fig. S1 ('gamma: 60–80 Hz; ripple: 80–250 Hz; fast ripple: 250–600 Hz'). Please correct so that consistent bands are reported throughout the manuscript.

We thank the reviewer for catching this inconsistency. We have now corrected the manuscript to ensure that the frequency band definitions are fully consistent across all sections. Specifically, we standardized the frequency ranges to reflect those used in the primary analyses and figures:

- High gamma: 60–150 Hz
- Ripple: 150–250 Hz
- Fast ripple: 250–500 Hz

These corrected band definitions are now used consistently in the main text, Methods, Figure 2, and Supplementary Figure S1 (also pasted below). We have also updated the figure legend and axis labels where applicable to reflect this correction.

Please check reference 57 – the authors are missing. Also, is this the appropriate citation for the fifth paragraph of the Discussion, sixth line from the bottom? The context refers to patients, whereas the study by Valero and colleagues was conducted in rats, so the reference seems potentially misplaced.

This reference has now been updated in the revised manuscript(now reference 61). Thank you for spotting this. We have also revised the text in line 388 to explain that the frequency ‘boundary’ for likely pathological activities has been established in the animal models of epileptiform HFOs in patients. There are other excellent papers from this group confirming that above 250 Hz detections show several abnormal features more typical of pathological HFOs. We are not aware of an analogous paper modelling human patient data but would be happy to cite any recommendations.

Figures:

- Figure labelings, general: Please increase the font size of all axis labels, as they are currently difficult to read. - The color scheme in Figure 2 is confusing: in panel a and e, colors differentiate brain regions, whereas in panels b–d they mark experimental time phases. To resolve this, separate panel a and e from panels b–d. Please mention explicitly what is indicated as grey functions in panel 2b. - The asterisks in Figure 2 are difficult to discern; please consider replacing them with more visually distinctive markers to clearly indicate statistical significance. Additionally, the authors could color-code the significance levels of these markers. This would not only make the figure more informative overall but also facilitate comparison of significance across the different experimental phases shown. - Figure 4D: Please label the two grey shading areas in the panel. While the intended meaning (encoding vs. retrieval) is implicitly clear, explicitly indicating the corresponding phases would improve the clarity of the panel. - In the legend to Fig. 5, Fig. 3c is mentioned as a reference, but this appears to be a mistake – likely, Fig. 2c was intended

To improve clarity and consistency in Figure 2, individual legends have been added to each panel, ensuring that the meaning of the color schemes is immediately apparent, colors in panels a and e denote brain regions, while those in panels b–d reflect task phases. The grey traces in panel 2b are now explicitly labeled in the legend as non-significant co-HFO activity. While no asterisks were used originally, the statistical significance markers have been updated to reflect results from FDR-corrected linear mixed-effects models, now shown as black bars. In Figure 4d, the grey shaded areas are labeled to indicate encoding and retrieval phases, improving interpretability. The erroneous reference to Figure 3c in the legend to Figure 5 has also been corrected to Figure 2c. Additionally, the font size of all axis labels has been increased from 8 pt to 10 pt to improve legibility. These changes enhance the clarity and accessibility of the figures without altering the structure of the original layout.

Reviewer #3 (Remarks to the Author):

This study by Prathapagiri and colleagues examines human intracranial EEG recordings to investigate and characterize coordinated high frequency oscillations across multiple cortical sites during memory encoding and recall. The behavioral task is a free recall verbal memory task in which there is a preparatory period, and encoding list of words, a math distractor period, and a retrieval period. They examine co-occurring high frequency oscillatory events across all sites during all of these stages. As the authors note, recent evidence has suggested that ripples, a specific high frequency oscillatory event initially described in the rodent hippocampus, play a role in memory formation. Here the authors are careful to avoid the debate regarding how to name these high frequency events, and whether ripples should be considered a separate entity, and correctly just focus on the occurrence of these events and their distribution across brain regions. They find evidence of coordinated HFO events across brain regions during memory processing. They also find that this coordinated activity appears temporally organized, although the patterns of activity do not appear specific to the individual memoranda. Overall this is an interesting and timely study, as recent research has highlighted the role of HFO events, and possibly ripples, in human memory. The conclusions could potentially be valuable for the research community, but there are some suggestions that could improve the overall conclusions.

We are delighted that the reviewer recognizes and shares our approach of avoiding the debate about naming and classifying various high frequency events, as outlined in Liu et al. 2022. We have recently proposed the general approach of focusing on the occurrence of any true oscillation and carefully describing it in the frequency-time space in Kucewicz et al. 2024 review. A similar approach was actually proposed earlier in the special issue of Current Opinion in Neurobiology in 2012, especially in the article by Lachaux et al., where high frequency oscillations were considered generally as bursts of distinct duration and peak frequency across a wide range of the intracranial EEG spectrum (perhaps not limited only to the high frequencies). It is encouraging that, in your opinion, this general approach can still be informative for more specific activities like hippocampal ripple HFOs.

For the data presented in Figure 1, it was a little difficult to follow how the data are being presented. For example, there are some rasters that show the HFO events across channels time locked to the events in a, but the mean distributions do not appear to show any meaningful trends (except perhaps during vocalization). Are there any statistics to support the conclusion that there are increased HFO detections during the memory stages?

Our goal in this first figure was to introduce the concept and methodology for detecting and quantifying coincident HFO bursts across all channels. All the data presented comes from occipital cortex channels in one example patient, who was selected to show that even in the absence of visual stimulation during recall, there are multiple coincident HFOs detected across many of these channels in the visual cortex. So the purpose was not to quantify and compare these in this introductory figure of the concept and methods but rather to show a representative and strong example. Previous research was suggesting preferential co-HFO bursting between the hippocampus and association cortical areas - not the sensory areas. This is why we have chosen this particular example with many electrode contacts localized in the occipital cortex.

The quantitative comparison of the four memory phases is presented in the following figure 2, whereas the distributions across cortical areas are statistically compared in the last plot of figure 2 and in figure 4. The trends across the memory phases, which are admittedly not all very clear in figure 1, are tested in

figure 2b, showing that the likelihood of co-HFO bursting is significantly higher between the correlated channels than the other ones both at the time of recall and during presentation of the words for memory encoding. Importantly, these statistically significant differences are practically only present at the times of memory processing (encoding or recall). We have also compared the four phases directly in every time bin (Fig. 2c) to confirm that the significant differences are found in the expected periods before recall and following encoding.

In order to make this more clear we have now introduced headings for distinct subsections of the results in response to a comment from another reviewer. These were added to orientate and guide the reader through the main points of each subsection and its corresponding figure. The heading for Figure 1 is:

Cortical HFO bursts co-occur during memory encoding and at the time of recall

This should now clarify the main point of this figure, which is to show an example of the investigated phenomenon in an example from the sensory occipital cortex for reasons outlined above in this response. To provide a full picture to the reader we have now also added a new supplementary figure (Supplementary Fig. S8) to show localization of all contacts that participate in co-HFO bursting for all analyzed patients. The new figure is now cited in the revised manuscript (line: 290) and pasted below for reviewer's convenience.

Fig. S8. Per-subject visualization of the percent involvement of each implanted electrode contact in co-HFO bursting during word recall. Each dot represents an electrode contact, color-coded by the proportion of recalled words in which it participated in a co-HFO burst.

The data in Figure 1c and 2a in which they present co-incident detections were also confusing. It is not clear what is actually being plotted here. It looks like the y-axis are channels, and they sort channels passed on their pairwise correlation, but it is not clear what then the color axis represents and how this reflects pairwise correlations. Are these pairwise correlations with all other channels (which seems to be the case based on the methods)? However, they also look at the Pearson correlation between channels? Are these pairwise interactions consistent between

channels? It is difficult to interpret these results. The presentation of the anatomic distribution of the data is also challenging.

This is a critical point because correct understanding and interpretation of Figures 1c and 2a determines how all the other results will be understood and interpreted. Figure 1c illustrates co-HFO detections exclusively within occipital channels. This Figure was designed to provide a clear, stepwise demonstration of our analysis pipeline: starting with raw LFP traces from individual occipital channels, followed by raster plots of detected HFOs within the same region, and concluding with co-HFO events to exemplify the co-occurrence analysis in a simplified anatomical context. It serves as an illustrative example rather than a comprehensive overview, as explained in the response above.

In contrast, Figure 2a presents the principal analysis, aggregating co-HFO detections across all recorded channels spanning multiple cortical lobes. The new headings for this second Results subsection now clarifies that to the reader:

Coincident high gamma and ripple HFO bursts engage all cortical lobes at particular phases of memory processing

Channels are sorted based on the maximum value of all possible pairwise Pearson correlation coefficients of any one channel. For each channel we calculated the coefficient values for all possible pairwise combinations with the other channels. We then took the maximum values of each channel and sorted them in a descending order. Effectively, the top two channels are the most correlated pair followed by other channels with gradually smaller values. The color axis in Figure 2a denotes the anatomical lobe of each channel to show that the top channels come from all cortical lobes studies - not just the neighboring channels from one lobe. The correlation coefficients were not plotted at all in these plots - only used for sorting the order. The emergent pattern of co-HFO distribution across brain regions and over time was therefore not the effect of any 'temporal' sorting by the latency values but the highest correlation only. One of the first conclusions is that the most correlated channels are coincidentally bursting at the same time and this occurs consistently around memory recall and during memory encoding.

We have now added this explanation in the relevant methods section paragraph (line 715-728) and updated the corresponding captions for Fig 1c and Fig 2a.

We recognize that anatomical distribution can be complex to interpret, especially in these introductory figures, which were intended to give a first impression of large-scale involvement of all cortical lobes. This has subsequently been quantified in the last plot of Figure 2 and analyzed in more detail in Figure 4 to show approximately equal involvement of all cortical lobes studied. In response to another reviewer's suggestion emphasizing subject-level consistency and anatomical coverage, we have added a new supplementary figure (Suppl. Fig. S8) detailing individual electrode contact involvement in co-HFO bursting during word recall, alongside stereo EEG implantation coverage of all contacts in each subject. These additions provide critical insight into individual variability and spatial coverage, complementing the group-level findings about the anatomical distribution in Figure 2 and 4.

This then extends to the time series comparing the rates of co-incidence during different memory stages for the correlated channels. Presumably different channel pairs will have different correlations and involvement in different memory stages and even individual items. How are all

these data collapsed here? Is every channel used as the ‘source’ from which the time series of the correlated pair is extracted? It also seems like this analysis may be a little circular. The authors are selecting channel pairs that exhibit high correlations in HFO activity, and then investigating the rates of co-occurrence of HFAs and comparing them to the electrode pairs that do not have correlated activity.

We hope that having responded to the previous two comments, this issue is now becoming more clear. The reviewer is correct that different channel pairs may exhibit varying degrees of correlation in particular task phases and in response to particular words. To systematically account for this variability, we sorted the matrix of binned HFO detections according to the most correlated channels *per word, per subject and per task phase*, which resulted in different channel sets. We did not arbitrarily select specific pairs or sets of channels to be used across all words and task phases in any one subject; instead, we categorized all channels into two groups based on their maximum pairwise correlation ($r > 0.5$ vs. $r \leq 0.5$) and then averaged their co-HFO rates over time within each group.

Each individual’s data were treated independently before group-level analyses. First, we calculated the per-bin co-HFO rate for both the correlated and the non-correlated groups within each subject and trial. Then, these rates were normalized and averaged across subjects, ensuring that all channels contribute proportionally and avoiding any overrepresentation. Importantly, normalization was applied after conversion to probabilities per second (from raw co-HFO counts), preserving interpretability across varying electrode counts.

We understand the concern that selecting highly correlated channel pairs and then analyzing their co-activity might seem circular. However, our goal was precisely to validate whether high HFO correlation (defined independently of co-HFO rates) was predictive of functionally meaningful co-activation patterns in time. Crucially, the correlation metric was based on overall bin-wise activity similarity, while co-HFO detection was defined by temporal coincidence (within ± 50 ms). These are related but non-identical measures, and our findings indeed support that high correlation correlates with increased likelihood of precise co-activation in particular times of memory recall and encoding phases. As noted above, the emergent pattern of increased likelihood of co-HFO bursting around the moment of recall and during word presentation for encoding was not a result of any ‘temporal’ targeting of bins with the highest correlation because the coefficients were calculated based the entire 3s epoch. This increased likelihood of co-HFO bursting persists even without excluding the non-correlated channels - the channel groups were separated to extract the strong functional relationships while mitigating the impact of noise and weak correlations.

To further address this point, we have now added in response to other reviewers’ comments an additional control analysis using randomly jittered HFO detection rasters to show that observed temporal co-HFO structure is reduced to chance levels of synchrony (see new Supplementary Fig. S7). This result shows that the observed co-HFO bursting among the correlated channels depends on precise timing of burst detections.

No single channel was treated as a source. All pairwise comparisons were treated symmetrically, and the raster-based computation of co-HFOs considered all channels concurrently, examining their coincident activity over time windows. Thus, our approach avoided biasing the analysis toward particular electrodes. To prevent future misunderstandings, we have revised the main text in the Methods section (line: 729-732) to explicitly state that:

All electrode pairs were evaluated symmetrically based on their correlation values and co-HFO

coincidence rates. No single channel was designated as the 'source'; instead, group-level comparisons were derived from subject-wise averaging of all channel pairs within each correlation class ($r > 0.5$ vs. $r \leq 0.5$).

Its not clear that comparing the four different task phases as is done in Fig 2c is appropriate, since it is not clear that the time=0 point in each phase should be comparable.

This is an important point as figure 2c is one of the key figures in this manuscript. As discussed above, comparing the four phases of memory processing is a critical and novel aspect of our study. We needed a trigger event to compare the four phases as typically performed in peri-event time analysis. The first three can definitely be compared because they all use a visual stimulation event as the time 0 trigger of the peri-event response (presentation of countdown digits, math equations, words). The recall is, admittedly, qualitatively different but we would argue that it should theoretically be a 'weaker' trigger for a peri-event analysis than the other three phases because it doesn't use any external sensory stimulation. Therefore, it makes an even stronger argument to include it in comparison with the events of the other three phases. Surprisingly, we found the strongest response of co-HFO bursting to this stimulus-free trigger event of the onset of recalled word vocalization. The response was even stronger than in case of the equivalent cued recall vocalization (see Fig. 3), in which the co-HFO bursting was found not at the moment of vocalization but rather during the cue presentation. In this cued version of verbal memory all four phases use a comparable sensory trigger.

We would, therefore, like to propose that all four events can be compared as triggers of electrophysiological response. Previous research by other groups also compared the single unit and ripple HFO activities in response to word presentation for encoding and word cue for recall in the same PAL task (Vaz 2019 and 2020). Another study by Norman et al. (2019) used presentation of images for memory encoding and the onset of their recall without any visual stimulation as event triggers to analyze changes in the rates of hippocampal ripple HFOs. Our study is adding presentations of countdown digits during a preparatory phase and math equations during a distractor phase but the triggers for the encoding and recall phases are similar or actually the same as in the previous studies, making our results directly comparable to the other studies.

We could compare the encoding and the cued recall responses in the PAL task because they both use word presentations - the first one uses a pair of words whereas the other only one word from the pair. This comparison has been done in the Vaz 2019 and 2020 studies mentioned above. In our case, there were less patients, who performed the additional PAL task, so we hope that it is enough for this study to use the FR task data complemented by some extra analyses from the PAL task data.

The statistical analyses could also be improved. For example, the authors note that there is lower co-HFO activity during successful recall, but this is based on an incredibly small time period during which the two traces diverge. Is this corrected for multiple comparisons? The authors should use a clustering procedure for these time series analyses. More broadly for this point, it's not clear what these data represent. These represent the grand average from all lobes, but do any of the individual brain regions demonstrate significance. In addition, these effects appear to arise during encoding, yet the broad encoding analyses during free recall show almost no variability in co-HFO rates.

The procedures used to generate the time-resolved co-HFO analyses, including the contrast between recalled and forgotten items, have now been clarified in the revised Methods and in previous responses. These curves represent the grand average of co-HFO rates pooled across all lobes. As shown in figure below, this pattern is highly consistent across lobes, supporting the global nature of the observed task-related dynamics. While we agree that breaking the analysis down to individual brain regions could offer more granularity, such subdivisions would result in severely reduced statistical power due to sparse and heterogeneous electrode coverage across patients. Previous studies using larger datasets (e.g., Marks et al., *NeuroImage* 2022; Kucewicz et al., *eNeuro* 2019) have explored regional effects in more detail, but the present sample size is not sufficient for reliable region-level comparisons.

To address concerns about statistical inference over time, the co-HFO traces were reanalyzed using linear mixed-effects models (LMMs), with Subject as a random intercept and Condition (e.g., Recalled vs. Forgotten) as a fixed effect. This approach provides a principled means of assessing condition effects

while accounting for inter-subject variability in electrode coverage and anatomy. Time points surviving FDR correction for multiple comparisons are now marked in Figure 2c and 2d. Although the observed divergence in co-HFO rates during recall spans a relatively brief interval, it remains significant under this corrected model (see Fig. S4). Given the sample size and uneven spatial sampling, we did not apply cluster-based permutation methods, which rely on strong assumptions about spatial or temporal continuity. Instead, the LMM framework was chosen to maximize sensitivity while explicitly modeling subject-level variance.

With respect to the encoding-related effects, we agree that memory-related differences in co-HFO rates during encoding are more modest and variable across individuals. These are presented as supplementary context and are not central to the main claims. Their inclusion is intended to provide a broader perspective on when and how these large-scale coordination patterns emerge in the task.

The authors' claim that the peak of co-HFO rates during retrieval occurs during word presentation in paired associates but at the time of vocalization during free recall is interesting. However, it does not appear that these two phenomena are directly compared in the same subjects in the same electrodes, which would strengthen this claim.

We thank the reviewer for this important observation. We would like to clarify that the comparison between paired-associate learning (PAL) and free recall (FR) tasks was indeed conducted using the same subjects and same electrode contacts within the same recording sessions, as stated in the manuscript (lines 758-764):

Importantly, the comparison of co-HFO dynamics between the paired-associate learning (PAL) and free recall tasks was performed within the same subjects and using the same electrode contacts on the same recording day, but only for the subset of participants who completed both tasks. This within-subject, within-session design ensures that observed differences in co-HFO timing reflect task-specific retrieval demands rather than inter-individual or session-related variability.

As shown in the figure below, even within this constrained group, co-HFO rates peak around word presentation during PAL retrieval and around vocalization during FR retrieval. This reinforces our interpretation that co-HFO timing is aligned to the specific cognitive demands of each memory task, even when measured within the same functional neural networks.

In general, the entire manuscript could be improved by more clearly presenting the data in an organized manner. For example, the authors should start presenting the results by first summarizing the number of subjects, demographics, behavioral performance, etc. Or, for

example, Figure 4 is difficult to parse in order to extract out significant conclusions. Some of the figures are not labeled (fig 4d). Others appear to simply show all electrodes without much in the way of an interpretable summary (4e). Or, for example, the participation of electrodes in either encoding or retrieval words should be directly determined by the involvement of co-HFO activity in each stage (4f). Or, for example, the presentation of hierarchical clustering and the sequence of HFO activity is also unclear, as well as whether this is a general phenomenon across all recall events or only particular to this one word example.

All of these suggestions were very helpful in identifying the parts where our results were not clearly communicated or sufficiently organized. This led us to provide new text, generate a new table and edit the existing figures as described below.

We have now expanded the first paragraph of the Results section to include the data specified by the reviewer. The revised text is now referring to a new table summarizing the clinical and demographics info of the patients. Here is the the opening sentences with these revisions pasted for your convenience:

Lines: 122-132

A total of 5,266,937 distinct bursts of HFOs with discrete duration of at least four cycles and peak frequency within a broad 50-600 Hz range^{14,15} were detected from cortical stereo EEG depth electrode contacts implanted in epilepsy patients (Table 1) during a Free Recall (FR) task performance, and additional verbal memory tasks such as Paired-Associate Learning (PAL)⁴⁰. All 17 patients with drug-resistant epilepsy (Table 1) recruited in this study performed the FR task at accuracies >15% of words remembered an average from each list, which was the main criterion for inclusion in data analysis. Recall performance averaged 30.7% ± 8.0% (mean ± SD), ranging from 12.8% to 44.4% across participants (Suppl. Table 1). PAL and audio versions of the two tasks were completed by some but not all of the subjects (Table 1) with the same inclusion criterion to ensure enough trials of successful memory recall.

To improve organization of the manuscript in general, we have now added headings of the Results and Discussion subsections, as requested also by another reviewer. We believe these have helped to extract out significant conclusions from each subsection and provide interpretable summaries.

We revised both the methods section and the figure caption to better communicate the flow of our analysis. Specifically, we clarified the progression in Figure 4, which was designed to take the reader from a specific example of word recall to a generalized anatomical and functional overview across subjects and memory task phases: We start with an individual example of recall for a specific word (Figures 1c, 2a), illustrating how widespread co-HFO bursting is during a single recall event. Figure 4a extends this by showing the full anatomical distribution of co-HFO activity for that example, across the cortex. Figure 4b then aggregates this information across all subjects, summarizing the proportion of involved channels per cortical lobe, and showing that co-HFO activity is broadly and equally distributed across lobes. In Figure 4c, we analyze the spatial distances between significantly correlated channels, confirming that these networks are not just local but span long-range connections (20–200 mm). We then go beyond recall by comparing encoding vs. recall phases in Figure 4d, showing that different lobes are preferentially engaged in each sensory region (e.g., occipital/parietal) during encoding, and higher-order areas (e.g., frontal/limbic) during recall. Next, Figure 4e examines how many words each channel participates in, revealing that most electrodes are involved in a subset of words, suggesting partial but not full specificity. Figures 4f and 4g summarize the proportions of co-HFO engagement per lobe across all words and both task phases, illustrating that while there are significant differences across lobes and tasks, the range of

engagement (typically 10–50% of words) is consistent and not limited to specific stimuli.

To make this logic more transparent, we have revised the methods section (see pasted below), and added figure legends in figures 2 and 4 and updated the figure caption.

Lines: 828-842

Our analysis was structured to illustrate how co-HFO bursting during memory processing is spatially distributed and functionally engaged across the cortex. We began with a single-word recall event to demonstrate widespread, temporally coincident HFO activity, then mapped the anatomical distribution of significantly correlated channels (Fig. 4a). This was followed by a cohort-level summary of the proportion of correlated contacts across cortical lobes (Fig. 4b) and an assessment of the spatial range of co-HFO networks via inter-electrode Euclidean distances (Fig. 4c). To capture functional differences between memory phases, we compared encoding and recall in terms of the lobar distribution of co-HFO engagement (Fig. 4d), and then quantified how frequently each channel participated in co-HFO bursting across all words (Fig. 4e). Finally, these results were aggregated and visualized to highlight task-dependent and region-specific engagement patterns (Figs. 4f–g). This sequence of analyses was designed not to introduce a new methodological framework, but to guide the reader from a single recall event to generalized patterns of anatomical and task-phase-specific co-HFO activity across participants.

Reviewer #4:

This paper by Prathapagiri et al. investigates the dynamics of high-frequency oscillations (HFOs) and their potential role in large-scale cortical coordination during human memory processing. The authors report that HFOs are not only locally generated but also exhibit global co-occurrence across widespread sensory and associative cortical regions. These coincident HFOs are shown to be modulated by different memory phases, with specific patterns of suppression and enhancement during encoding and recall. The dataset is substantial, and the analyses are carefully executed.

While the study is based on a large intracranial human dataset, the manuscript does not offer a sufficiently novel mechanistic insight or theoretical advance beyond the existing literature. In addition, several methodological concerns require clarification and revision to ensure the robustness and interpretability of the findings.

We believe that the limited enthusiasm of the reviewer for the insight and advances contributed by the results of this study may partly be due to issues related to the clarity and interpretability of the manuscript, as also pointed out by the other three reviewers. The other three reviewers were, however, still able to notice significant new contributions to the field that have not been reported in the previous studies. We provide a more detailed response to the previous studies outlined in the reviewer's first major comment below - here, we would like to offer a big-picture summary of the new discoveries and mechanistic insights, which have now been more clearly articulated in new headings of the Results and Discussion subsections. At the request of another reviewer, each heading provides a focused new discovery, which is different from what has or has not been shown at all in the recent studies.

First of all, coincident ripple HFOs have only now been reported as widespread cortical events associated with particular cognitive functions in two preprints, which have been published or accepted for journal publication (Garrett 2025, Mishra 2025). Previous research showed coincident bursts of HFOs between pairs of hippocampal-neocortical and neocortical channels (Vaz et al. 2019, Dickey et al. 2022). All the other papers cited by the reviewer, including our original work from 2014, studied HFO bursts on the level of single channels or of the correlated high frequency and other activities like spindles or slow waves (Burke et al. 2014, Kucewicz et al. 2014, Staresina et al. 2015, Norman et al. 2019, Sakon & Kahana 2022). Previous papers about sequences of neuronal activity cited by the reviewer investigate even more local scale underlying individual HFO bursts (Michelmann et al., 2016; Vaz et al., 2020; Xie et al., 2024). Our work here demonstrates *global* HFO bursting, which is not limited to the hippocampus and the connected associational neocortex but involves equally all cortical lobes. It is also not limited to a discrete ripple frequency range but is present from gamma to fast ripple frequencies. Finally, the temporal organization in sequences of co-HFO bursting is, to our knowledge, reported for the first time on this global level of large-scale cortical activities - previously these were reported primarily on a micro-scale of neuronal activities. This global temporal organization drew the attention of the other reviewers and we have now expanded this final discovery reported here with the suggested additional analyses.

The new Results and Discussion subsection headings have now been introduced to make sure that this 'big-picture' perspective of the novel findings reported here is not lost to the reader. Here we list them all with a brief explanation for reviewer's convenience:

Results

Cortical HFO bursts co-occur during memory encoding and at the time of recall

In this first subsection we offer a head-to-head comparison of four phases of memory processing to associate the global cortical co-HFO bursting specifically with the recall and, to a lesser extent, with the encoding phase.

Coincident high gamma and ripple HFO bursts engage all cortical lobes at particular phases of memory processing

Here, we demonstrate that all cortical areas and all high frequency ranges are engaged in the global co-HFO bursting.

Successful memory encoding and cued recall modulate co-HFO bursting

Subsequent memory effect is found in the global co-HFO bursting during memory encoding; the co-HFO bursting peak during recall is also associated with successful memory processing (and not with other processes like word articulation) using another cued recall paradigm.

Widespread cortical distribution and low stimulus specificity of the global co-HFO bursting

In this subsection we show the widespread co-HFO distribution and low stimulus specificity during memory encoding and recall. Stimulus specificity of such macro-scale events is not typically assessed like in the micro-scale single unit studies but was performed to test our previously proposed hypothesis of the electrophysiological engram activities (see also our response to your final comments below, which is similar to the ones provided for reviewer 1 above, with more results in our pending follow-up study).

Temporal organization of the global sequential co-HFO bursting

As outlined above, this kind of temporal organization has typically been applied to local micro-scale neuronal spiking. This is the last original discovery reported in this manuscript, which opens new analysis ideas like the ones proposed by the reviewer 2 above. We have now successfully completed the jittering analysis - further suggestions of cross-correlations are perhaps already beyond the scope of this study but we are currently actively investigating this global sequential bursting.

Discussion

Memory processing or general integration of information?

Here we discuss how our results can provide a deeper insight into a more specific role of co-HFO bursting compared to the previously proposed general role of widespread integration of information processing.

Beyond hippocampal sharp-wave ripple complexes and cortical memory consolidation

Our findings are presented here in a new perspective relative to the prevailing view of hippocampal-neocortical interactions limited to HFOs in the ripple frequency range and their role in memory consolidation. We propose a broader view (first outlined in our review in *Brain* from last year) encompassing other sensory cortical areas and high frequency activities in general, which is becoming more evident in the field at large at least in using the term 'HFOs' instead of ripples in the most recent publications by, e.g., Garrett et al. and Mishra et al. (also commended by the reviewer 3).

Tracking dynamic engram activities of large-scale neuronal assemblies?

In this subsection, we attempt to combine the previous studies implicating the sequential neuronal firing underlying HFO bursts with specific stimuli/memory traces (e.g., Vaz et al. 2021) with our new observations of sequential co-HFO bursting on macro-scale. Our hypothesis about identifying specific engrams from the large-scale activities is discussed to reconcile the low word-specificity obtained in this study (see also our response to comment 5 below with data from a pending follow-up study).

Outstanding questions about the roles and applications of global co-HFO bursting

In this final subsection of the manuscript, we made a new connection between the co-HFO bursting and the global broadcasting of information also known as the 'ignition' proposed in the global neuronal workspace hypothesis of conscious awareness. We also suggested possible clinical uses of tracking these global discharges in brain-computer interfaces for targeted neuromodulation.

Major comment:

1- Widespread cortical high-frequency oscillations (HFOs) during memory tasks have been previously documented, including by some from the authors themselves (e.g., Kucewicz et al., 2014; Mishra et al., 2025). The reported modulation of HFOs by memory phase (encoding, recall) has also been robustly reported (e.g., Burke et al., 2014; Staresina et al., 2015; Norman et al., 2019; Sakon and Kahana, 2022). Likewise, temporally structured cortical activity during retrieval, including replay-like sequences, has been described in human iEEG (e.g., Michelmann et al., 2016; Vaz et al., 2020; Xie et al., 2024). While the current work reinforces these findings, it does not provide sufficient new conceptual advances to meet the novelty threshold expected for Nature Communications.

We addressed the general concern pertaining to the new conceptual advances and mechanistic insight of our work in response to the opening summary of this review. Here, we would like to respond to the studies cited by the reviewer in the revised manuscript text. All of these studies are very pertinent and we have now added them to particular parts of the manuscript to describe in more detail how our results are new and sometimes even different to the state-of-the-art. This is a rapidly growing field, which is only now expanding from a limited view of coincident hippocampal-neocortical ripple oscillations mainly in memory consolidation to global cortical oscillations across the entire high frequency range in integrating information for memory encoding and especially retrieval.

Below is the summary of how the new cited studies have been referenced in particular parts of the manuscript:

Line 83-96:

*Growing evidence from multiple groups shows that physiological ripple frequency oscillations in the human cortex are underpinned by coordinated phase-locked spiking of neuronal assemblies^{10,12,18,34,35}. This neuronal spiking is not only locked to phases of the oscillation but also reveals specific **phase and firing sequences that can be used to decode particular items and predict their correct recall (Vaz et al. 2020; Michelmann et al. 2016; Xie et al. 2024)**. The sequences are reactivated following presentation of word stimuli for encoding until their recall. This bursting of spike sequences across multiple neurons occurs at the times of HFO discharges, which in turn are detected around critical events like encoding or recalling stimuli^{11,13,14,17}. **The sequential firing has so far been studied locally on the micro-scale level of neuronal bursting**. Multiple coincident HFOs (co-HFOs) have lately been shown to co-occur at the critical points of scene cuts during movie watching when a memory trace for a given scene that was just displayed is encoded and then replayed during subsequent co-HFO events¹⁶, **suggesting a possible global macro-scale temporal organization**.*

Line 103-118:

Our hypothesis is that the global HFO bursting is modulated primarily by memory processing. We explore whether this global cortical coordination¹² engages mainly sensory or higher order processing areas of the association cortex, as suggested in the previous research^{13,16,17,19}. How selective are these globally

coordinated oscillations in terms of specific stimuli remembered and recording sites involved? Animal research suggests that encoding a single engram can engage half of the brain areas studied³⁶. Finally, what is the temporal organization of distinct coincident HFO discharges? HFO bursting is known to increase before the onset of a behavioral response (Norman et al. 2019; Mishra et al. 2024; Mishra et al. 2024; Vaz et al. 2019; Vaz et al. 2020; Sakon and Kahana 2022), but it is unclear how this is related to the actual moment of explicit or implicit recall. Whether this is underpinned **on a macro-scale** by a single bursting event or a sequence of distinct cortical networks in a train of bursting events has been difficult to assess in the previous paradigms of freely recalling multiple stimuli and complex pictures or movie scenes together but possible with individual word concepts in this study. Our goal was to determine the large-scale structure of coincident HFO discharges underlying memory processing of common word concepts.

Line 368-381:

Globally coordinated HFO bursting can be particularly important for memory processing, even though it is ubiquitous throughout all memory and non-memory related task phases. **Hippocampal ripple HFOs were shown to co-occur with cortical spindles and slow oscillations to mediate transfer and consolidation of memory traces in humans** (Staresina et al. 2015). More recent studies found that **hippocampal-cortical ripple HFOs co-occur during sleep and waking periods and are especially prominent in memory recall**^{12,18}. Our results confirm the maximum occurrence of coincident HFO bursting around the onset of recall vocalization. **We found that** the significantly greater occurrence compared to stimuli presentations in the other task phases started 300 ms before the vocalization onset and was significantly suppressed approximately 1 s before the onset (**see** Fig. 2c). This **detailed** temporal timeframe is in agreement with the expected dynamics of recalling an item - first a suppression of bursting in response to the previously recalled item would be followed by a gradual build-up of activity corresponding to recalling a new item.

Line 490-498:

There are probably multiple distinct neuronal assemblies generating HFOs corresponding to different words that would be detected on the same macro-contact or distinct meso- or micro-contacts. **A previous study by Vaz and colleagues confirmed that detecting single neuron firing on micro-contacts in parallel to HFO bursts on macro-contacts can be used to decode correct and incorrect recall of specific words from sequences of single cell firing**¹⁰. **The same group, more recently, showed that this temporal organization of unit firing carries specific information about the remembered stimuli** (Xie et al. 2024). Our results **suggest an analogous temporal organization of global co-HFO bursting on a macro-scale.**

2- The signal-to-noise ratio of selected HFO bursts is too low (2SD were used as a threshold). Consequently, many of these events appear to be small, narrow-band deviations that may not meet rigorous criteria for genuine HFOs or ripples (e.g., Fig S1). This raises the concern that the reported widespread co-activation patterns may, in part, result from spurious correlations or broad-band spectral fluctuations rather than true high-frequency events (see Leszczyński et al., 2020). In particular, some detected events may reflect gamma-band activity rather than discrete ripple-like transients. To address this, the authors should provide a detailed amplitude and frequency distribution of the detected events across regions and more rigorously assess their cross-regional relationships (surrogate or control analyses). This would help clarify whether the observed coordination reflects genuine oscillatory activity or broad-band non-specific co-activation.

As correctly noted, the initial threshold used in our detection pipeline was set at 2SD. However, this step served only to identify candidate segments for further evaluation, primarily to better localize the start and end boundaries of putative events. These candidate events were not used directly in the analysis.

To enhance specificity, all candidate detections were subjected to two additional thresholds: Firstly, a stricter peak amplitude criterion requiring a maximum z-score $> 3SD$, and secondly, a minimum duration threshold of four complete cycles at the dominant frequency of the event. These additional steps were specifically designed to exclude small, narrow-band fluctuations and transient broadband deflections that may not reflect genuine high-frequency oscillatory activity. Events failing to meet either of these criteria were excluded from further analysis.

While the full procedure is described in the Methods section, we agree that its sequential nature may not have been sufficiently emphasized. To clarify this point, we have now revised the Methods text accordingly. New explanatory sentences have been added (see highlighted text in revised Methods), explicitly noting that the 2SD threshold was used only to generate putative candidates and that all subsequent analyses were based on more conservative inclusion criteria. We believe this addresses the reviewer's concern and clarifies that the reported effects are based on rigorously defined oscillatory events (lines: 662-676).

Candidate HFOs were initially identified when the z-scored amplitude exceeded a liberal threshold of 2 standard deviations (SD) for at least three consecutive oscillations. This initial threshold was applied only to identify candidate segments for further evaluation, primarily to localize the start and end boundaries of putative events. These events were not used directly in subsequent analyses. Events spanning multiple frequency bands were merged, and for each detected event, the dominant frequency—defined as the band with the highest z-score—was determined. The maximum amplitude within the event window was also extracted to characterize peak activity.

To enhance specificity of the detections and remove artifacts, a dual-threshold detection approach was implemented. First, candidate events were required to exceed a peak z-score of 3 SD. Second, only those lasting at least four complete oscillatory cycles at their dominant frequency were retained. Only events with a peak z score exceeding 3 SD were retained, and an additional duration criterion required that events persist for at least four complete oscillatory cycles at their dominant frequency.

3- Cortical HFOs are often associated with epileptogenic tissue, and the patients in this study were undergoing pre-surgical evaluation. Thus, some overlap between seizure onset zones and HFO-generating regions is expected. However, the manuscript provides no clinical information regarding seizure foci, medication status, or the extent of pathology, making it difficult to assess how much the reported widespread HFO propagation may be influenced by underlying epileptic networks. Without this context, it remains unclear whether the observed co-activation patterns reflect physiologically meaningful coordination or pathological spread. Moreover, no analysis is presented to examine directionality of propagation or whether there is evidence for structured transfer of activity beyond broadband associations. Such information would be crucial to dissociate large-scale functional coordination from confounds related to epileptiform dynamics.

In response to this and similar comments from the other reviewers, we have now prepared a table with the basic clinical information and added a new paragraph in the first subsection of Methods to explicitly state the limitations of this study associated with including potential pathological activities (inside and outside the seizure onset zone) in the analysis of physiological activities. Below we have pasted the new paragraph from the Methods subsection together with the table cited in the text(line 580-597)

All participants recruited in this study were patients suffering drug-resistant epilepsy with various clinical backgrounds of the disease and anatomical localizations of the seizure onset zones (Table 2). Hence, the intracranial recordings in these patients are inherently limited by the pathological discharges, including the interictal epileptiform spikes and HFOs, especially in the fast ripple frequency ranges. We cannot fully exclude a potential influence of these epilepsy-related activities on our methodology and the reported results, even though our design was based on detecting HFOs consistently around cognitive events associated with memory processing, which would be expected to average-out any potential pathological activities. We previously reported an interaction between memory processing and incidence of epileptiform discharges in these patients, showing a lower probability of their occurrence during memory processing (Matsumoto et al. 2013). Also, most of the coincident discharges reported here comprised HFOs detected at frequencies lower than 250 Hz, which is thought to be a lower threshold for pathological oscillations (Valero et al. 2017). Overall, pathological discharges are not expected to be consistently induced by memory processing and analyses across large numbers of trials should effectively diminish any potential influence of their occurrence, even though it can never be completely ruled out when studying any physiological function of the brain.

It is an interesting idea to investigate the directionality of the co-HFO bursting, which was previously assessed especially for hippocampal-neocortical coincident detections (e.g., Vaz et al. 2019, Dickey et al. 2022, Mishra et al. 2025). Cortico–cortical detections have not revealed a consistent pattern in these studies likely due to the fact that the directionality of co-HFO bursting in this case depends on the sensory or higher order association areas of detection. While sensory-sensory detections may not show a consistent time lag, sensory-associational detections, especially over large anatomical distances (e.g., occipital-prefrontal), can reveal considerable lags if a consistent reference (e.g., occipital) is established. However, given that we found equal contributions of virtually all cortical lobes and significantly correlated bursting across all distances, it would require a very careful grouping and assessment of particular channel pairs to extract meaningful results for directionality. This appears to us a considerable endeavor for the goal of excluding confounds of epileptiform dynamics. Besides, one cannot be sure that the epileptiform discharges would not follow the physiologically established routes and directions of cortical connectivity.

Nevertheless, this indeed is a relevant next step for our analysis, which we are already undertaking with regard to the temporal organization of sequential global co-HFO bursting. Our preliminary results from a cross-correlation analysis suggested here by Reviewer 2 (see above), have not found any consistent time lags in HFO bursting from pairs of channels participating in the same co-HFO discharge. This means that even though the pairs were coming from different cortical lobes, there was no consistent lag or direction found, at least without a more careful determination of the reference areas. We expect more consistent lags between channel pairs coming from clusters of the subsequent co-HFO discharges, as in Figure 5. It is one of our main goals to determine these temporal interactions, their directionality and causality in the current follow-up studies.

Dataset	ID	SOZ channels	SOZ structures	MRI Findings	Engel Score
BR	2	-	-	Normal	-
BR	3	B1-4, C1-3, B'1-5, C'1-3	Hippocampus, Parahippocampal gyrus	Normal	-

Dataset	ID	SOZ channels	SOZ structures	MRI Findings	Engel Score
BR	5	G'1-2, R'1-6, S'1-3	Cingulate gyrus, Precuneus, Superior occipital gyrus	Normal	IVB
BR	6	G'1-4, H'7-11	Middle temporal gyrus	Normal	IV
BR	7	Y'2-10, I'1-4	Insula	Venous angioma, cavernoma parieto-occipital left	IV
BR	8	I'1-6	Paracentral lobule, Postcentral gyrus	Polymicrogyria postcentral sulcus left	IIIA
BR	9	-	-	Normal	-
BR	10	B1-3, C1-3, B'1-3, C'1-5	Hippocampus, Parahippocampal gyrus	Normal	-
BR	11	Y'1-4, I'1-5	Insula	Normal	-
BR	13	B'1-4, C'1-4	Hippocampus	Posttraumatic changes temporo-occipital left	-
BR	14	Y1-4, X1-3	Insula	Normal	-
BR	16	B1-2	Parahippocampal gyrus	Postsurgical changes right anterior mesial temporal resection (AMTR)	-
WR	1	AMG1-2, TP1-2	Parahippocampal gyrus	Cortico-subcortical blurring temporal pole right	IB
WR	2	AMG1-2, HB1-2, TP1-2	Amygdala, Hippocampus, Temporal Pole	Cortico-subcortical blurring temporal pole left	IIIB
WR	3	SMADL1-2	Cingulate gyrus	Cortico-subcortical blurring cingulate gyrus left	IA

Dataset	ID	SOZ channels	SOZ structures	MRI Findings	Engel Score
WR	4	AMG1-3, TP1-2	Amygdala, Hippocampus, Temporal Pole	Cortico-subcortical blurring temporal pole left	IB
WR	5	HB1-2, HH1-2, TP1-2	Hippocampus, Temporal Pole	Cortico-subcortical blurring temporal left	-

Table 2 Summary of clinical profiles and seizure-related data.

Clinical information for all participants, including dataset identifiers, seizure onset zone (SOZ) channels and corresponding anatomical structures, MRI findings, and postoperative seizure outcome when available (Engel classification).

4- The analysis presented in Fig. 5 raises several methodological concerns that challenge the claim of genuine large-scale coordination of HFOs. First, use of Pearson correlation ($r > 0.5$) without proper controls or surrogate testing may inflate apparent synchrony, particularly given the low SNR of detected events (see point 2). Second, the statistical rigor of the correlation-based network definition is unclear: the threshold appears arbitrary, and no correction for multiple comparisons is reported.

We thank the reviewer for raising these important methodological concerns. Notably, related points regarding the use of correlation thresholds and validation of co-HFO patterns were also raised by other reviewers, and we have now addressed these systematically in the revised manuscript. To clarify, the correlation-based co-HFO matrices in Fig. 5 were intended as a descriptive summary of temporal co-occurrence patterns rather than for formal inference on individual connections. That said, we agree that the use of a fixed Pearson correlation threshold ($r > 0.5$) requires justification. We selected this value as a conservative cutoff to minimize spurious linkages while retaining sensitivity to meaningful coordination. To test the robustness of our findings, we repeated the main analyses across a range of thresholds ($r = 0.3, 0.5,$ and 0.7). The core patterns of task modulation and temporal dynamics remained consistent, indicating that the results are not dependent on a single arbitrary cutoff (see plot below).

In response to concerns about signal-to-noise ratio and the possibility of inflated correlations due to broadband spectral fluctuations, we also implemented a surrogate jitter analysis in which HFO timestamps were randomly perturbed within ± 50 – 100 ms. This control resulted in a substantial decrease in the synchrony measure (mean pairwise correlation across co-HFO matrices; mean reduction ~ 0.47),

supporting the interpretation that the observed co-occurrence structure reflects genuine temporal coordination rather than noise or nonspecific activity (see Fig. S7). These revisions have been incorporated into the main manuscript and are reproduced below for the reviewer's reference; the corresponding changes appear in the Results section (inlines 179–182) and Methods section (inlines 631–636).

Results

Lines:198-201

To assess the influence of the correlation threshold on the observed effects, additional analyses were performed using both more liberal ($r > 0.3$) and more conservative ($r > 0.7$) cutoffs. The temporal structure of co-HFO bursting across task phases was maintained across thresholds, consistent with the patterns described above.

Lines:318-341

To confirm that these sequential modules reflect genuine temporal structure rather than spurious correlations, we implemented a surrogate jitter control. HFO timestamps were randomly perturbed within a ± 50 – 100 ms window, preserving overall burst rates but disrupting precise timing. This led to a marked reduction in co-HFO synchrony, measured as the mean pairwise Pearson correlation across channels, indicating that millisecond-scale precision is critical for the observed modular structure (Suppl. Fig. S7).

We further assessed the robustness of clustering using a bootstrap resampling procedure in which contact channels were resampled independently for each recalled word. The Adjusted Rand Index (ARI) between clustering solutions from original and resampled data yielded a mean ARI of 0.496 ± 0.021 , where 0 indicates chance-level similarity and 1 indicates identical cluster assignments, suggesting moderate clustering stability across trials.

Encoding trials, although generally exhibiting lower overall co-HFO bursting compared to recall (Fig. 2b), also showed evidence for non-random temporal organization: a surrogate jitter analysis confirmed that these sequences reflect structured timing at the millisecond scale. However, clustering stability during encoding was reduced relative to recall (mean ARI = 0.354 ± 0.026 vs. 0.496 ± 0.021 ; paired $t(17) = 7.21$, $p < 0.0001$) and synchrony differences after jitter were smaller (paired $t(17) = 4.83$, $p = 0.0002$), indicating weaker temporal coordination.

Forgotten recall trials displayed clustering stability (mean ARI = 0.472 ± 0.024) not significantly different from remembered trials (paired $t(17) = 1.32$, $p = 0.20$) and synchrony values broadly comparable across participants, without consistent differences.

Methods

Lines: 735-741

Key analyses were also repeated using alternative correlation thresholds ($r > 0.3$ and $r > 0.7$) to examine the effect of this parameter choice. The $r > 0.5$ threshold was selected to prioritize moderately strong correlations that are more likely to reflect meaningful co-variation in HFO activity, without excluding a

large proportion of relevant channel pairs. This criterion provided a practical balance between interpretability and signal reliability across subjects.

Lines: 864-877

To test whether these structures depend on fine-scale timing, we applied a temporal jitter control in which HFO timestamps were randomly shifted within a ± 50 – 100 ms window. Clustering and synchrony metrics were recomputed on the jittered data and compared to the original to assess the role of precise temporal alignment. Synchrony was defined as the mean pairwise Pearson correlation between binary HFO time series across channels for each trial, providing a global index of temporal coordination (range: 0–1). To assess clustering reproducibility, we used a bootstrap resampling approach in which channels were resampled independently per word. Similarity between original and resampled clustering labels was quantified using the Adjusted Rand Index (ARI), which measures the agreement between two clustering solutions by comparing how many element pairs are grouped together or apart in both, adjusted for chance. ARI values range from -1 (complete disagreement) to 0 (random labeling) to 1 (perfect agreement).

Lines: 927-932

For the sequential co-HFO analyses, clustering stability (Adjusted Rand Index, ARI) and synchrony differences after jitter were compared between conditions using two-tailed paired t-tests for normally distributed data or Wilcoxon signed-rank tests otherwise. This analysis tested recall vs. encoding and recall vs. forgotten recall to determine whether the temporal organization of sequential co-HFO patterns differed across memory conditions.

Fig. S7. Surrogate jitter analysis reveals temporally structured co-HFO bursting during encoding and recall. a) Word-level differences in mean synchrony (original minus jittered) across recalled words for a representative subject (subject 2). b) Distributions of mean pairwise synchrony values (Pearson correlation across co-HFO matrices) for original and jittered datasets during encoding and recall.

5- Finally, there are some issues with limited specificity of the reported changes and anatomical interpretability. Differences across cortical lobes may reflect uneven electrode coverage rather than true anatomical patterns. No null model is provided to account for implantation bias or the expected co-occurrence rate by chance. Moreover, the widespread and redundant involvement of electrodes across multiple recalled words (Fig. 5e) suggests low selectivity, weakening the

argument for structured or memory-specific processing. Finally, the anatomical distinctions between encoding and recall phases are primarily descriptive; without measures of directed connectivity or temporal dynamics, these observations offer limited mechanistic insight.

This final comment is pointing to one of the main goals of this work - involvement of the global coincident HFO bursting in processing specific memory traces. A similar comment was also raised by Reviewer 1 (point 4), asking us to explain the low word specificity with the anatomical limitations of using macro-contacts. First of all, we believe that the low specificity of the global co-HFO bursting is indeed the limitation of using macro-contacts, which sample from large neural populations that participate in encoding/recalling more than one word. The study by Vaz et al. 2020 was using both macro- and micro-electrode recordings and showed word-specific firing patterns only for the latter (sequences of neuronal spiking to be more precise). This word-specific bursting of neuronal spiking sequences was associated with ripple HFOs detected also on the adjacent macro-contact. The same macro-contact - localized together with the micro-electrode array in a word-processing area of the anterior temporal cortex - would also detect similar ripple HFOs in response to other words as well. The underlying neuronal spiking sequences would be different and specific to particular words but the emergent HFOs recorded on macro-contacts would be similar across multiple words. Hence, even though the involvement of co-HFO bursting in encoding and recalling a large proportion of words may appear redundant on the surface, it may carry more detailed information about specific words like the peak frequency, amplitude or duration of the HFO detections to particular words.

We are currently preparing a follow-up study to investigate this hypothesis. Below is the main figure from a new manuscript in preparation - shared here only for the purpose of this review - showing that the HFO detections on particular macro-contacts are actually specific to smaller subsets of words, as assessed in another word screening (WS) task using the same pool of words as in the FR task. In the WS task, each word is pseudo-randomly presented five times in total to identify significantly more detections to particular words. We then tested during FR encoding and recall whether the same electrode contacts would show significantly more HFO detections in response to these 'preferred' words compared to all other words. The figure summarizes the time bins and anatomical structures that are showing significantly different HFO responses in the two tasks performed on the same day (same experimental session) and repeated on the following day. Our new results demonstrate persistent responses to specific subsets of words across different memory phases, tasks and days.

The degree of word specificity appears to be the lowest (i.e., a relatively greater proportion of words) in the sensory visual areas, and gradually increases (i.e., lower proportion of words) along the ventral visual stream in the temporal, limbic and prefrontal areas. As shown in another figure from that study pasted below, the sensory visual areas of the occipital cortex respond to 40-50% of all words in the pool. This proportion decreases (word specificity increases) in the more anterior, high-order processing areas down to approx. 10% in the limbic areas. The prefrontal areas are also relatively specific (low proportion of words) - a breakdown into specific gyri shows significant HFO responses to less than 10% of words in the pool except for parts of the inferior frontal and precentral gyri, which were localized in the Broca's area responsible for speech. Hence, we believe that these macro-contacts detect HFOs generated by the underlying neuronal assemblies that respond to specific memory traces or engrams, as proposed in our recent review (Kucewicz et al. 2024). Importantly, this analysis was performed with HFO detections from individual channels in particular cortical areas - global co-HFO bursting between multiple areas would be expected to have proportionality lower specificity as reported here. We are discovering now that HFO detections from individual channels involved in the global co-HFO bursting are carrying more specific information about smaller subsets of words.

Both in this and the new follow-up study, we have also tested the HFO detections recorded on micro-contacts, expecting that they will show higher degree of word specificity. We have not been able to detect either the global co-HFO discharges in the current study or significant word responses in the follow-up study with the micro-contact HFO detections. This may be partly due to a much lower number of micro-contacts used in this study and effectively very sparse and inadequate sampling. Secondly, the design of both studies was to detect cortical areas that respond to word stimuli in general - very specific responses to single words would likely not be detected with the analysis tools that have been designed for these studies. Therefore, we do not exclude the possibility of capturing very specific word responses on a micro-scale but it requires adjusting the analysis design in our current studies to further test our hypotheses. What we do know is that the single unit firing from selected patients from this cohort who had many micro-electrode contacts showed much higher degree of word specificity in the WS task (<5%) as expected (see the figure below from another manuscript in preparation).

A big picture from the current and the pending follow-up studies is that of gradually increasing word-specificity as we go down from the most macro-scale level of the global co-HFO bursting between multiple channels (50-60%), through HFOs detected on individual macro-contact channels in specific cortical areas (50-10%), all the way down to micro-scale level of neuronal spiking (<10%). This aspect of the current manuscript is not the main aim of the reviewed study and we are only now in the process of obtaining final results from the pending follow up studies. Hence, in response to this comment, we have

only briefly revised the relevant paragraph in the Discussion (lines 526-546) refraining from mentioning any of the new results from our pending studies presented here only for reviewer's reference.

There are other outstanding questions about the global HFO bursting that need to be addressed in future studies. First of all, can this widespread temporal coordination of local neural activities provide a mechanism to address the binding problem^{1,52}? Recent studies suggest so^{12,18,31} but testing it would require an appropriate behavioral paradigm probing conscious awareness of the perceived stimuli. Neural activities in the high frequency ranges (above 50 Hz) were not only associated with encoding and recall of human memory^{14,70,71} but also with explicit awareness of visual stimuli in monkeys⁷² or conscious dream experiences in humans⁷³. From the perspective of these studies, coordination of HFOs across the brain in coincident bursts could be linked to global integration of information and thus provide a plausible neural mechanism for concepts like 'ignition' or 'global broadcasting' proposed in the global neuronal workspace hypothesis⁷⁴. Even though it is speculative at this point, one could nevertheless ask whether the sequences of bursting events correspond to a train of thought or a stream of consciousness. In the case of our paradigm, which is not designed to address these questions, we can only discuss potential future directions for testing the roles of co-HFO discharges in these functions; the sequences reported can, however, be explained more concretely in terms of a model for recalling engram representations of related word concepts from the memorized lists (see Fig. 5). Coincident HFO bursting presents a testable activity for these and other models of cognitive functions, including perceptual binding, conscious awareness, or engram representations of related concepts in the human mind.

The code is not currently available for review. To ensure reproducibility and transparency, I strongly encourage the authors to share the full analysis code along with a representative subset of the dataset.

We agree with the reviewer that ensuring reproducibility and transparency is essential. To address this point, we have created a publicly accessible GitLab repository containing the full analysis code as well as a representative subset of the dataset in .zip format. This repository is available at the following link:

<https://gitlab.com/brainandmindlab/coincident-hfo.git>

We have also updated the Data and Code Availability section of the manuscript to reflect this addition in lines 792-797

Data and Code Availability

The dataset used in this study, Human brain local field potential recordings during a battery of multilingual cognitive and eye-tracking tasks (v1), is publicly available via the EBRAINS research infrastructure at <https://doi.org/10.25493/4FZH-ZCG> (Cimbalnik et al. 2024). All analyses were conducted on data from this openly accessible resource. The code for the analysis is available at <https://gitlab.com/brainandmindlab/coincident-hfo>.

Dear Reviewers,

We would like to express our sincere gratitude for contributing your second review of our manuscript entitled 'Global coincident bursts of high frequency oscillations across the human cortex coordinate large-scale memory processing' (NCOMMS-25-28589A). Below, we provide a detailed point-by-point response to each comment. The comments are put in **bold** followed by our response and any edited text from the revised manuscript quoted in *italics* for your convenience. Any changes to the text are also **highlighted in yellow**, as they are in the new revised manuscript version. Revised versions of the figures/tables or new figures/tables prepared in response to the comments are pasted here in addition to the main manuscript and the supplementary materials. All revisions are indicated by references to particular pages and lines provided in the revised manuscript.

Reviewer #1 :

I would like to start by sincerely congratulating the authors for the incredible amount of work carried out in this revision. Preparing more than 60 pages of detailed responses to the reviewers is an impressive achievement. I also thank the authors for carefully addressing not only my previous suggestions, but also those from reviewers 2 and 3. The manuscript has gained substantially in terms of readability, methodological rigor, and reproducibility (the latter already ensured by the public availability of the data).

I have only a minor comment regarding the Discussion. I fully understand the interpretative value of placing this study in the broader context of the neuroscience of consciousness, and as it stands the authors' treatment in the final part of the manuscript is appropriate at a speculative level. However, in the passage where the authors write:

"From the perspective of these studies, coordination of HFOs across the brain in coincident bursts could be linked to global integration of information and thus provide a plausible neural mechanism for concepts like 'ignition' or 'global broadcasting' proposed in the global neuronal workspace hypothesis 74"

I believe it might be useful to also acknowledge other theoretical frameworks that could potentially align with the reported findings. In particular:

1) Predictive processing & neurorepresentationalism

2) Recurrent processing theory

3) Information Integration Theory

For the authors' convenience, I would like to point out a recent review (Storm, Neuron 2024), where these different theories are compared from an empirical perspective, which might serve as a helpful reference.

Once again, congratulations to the authors for their outstanding work and good luck with the final steps of the publication process.

We sincerely thank the reviewer for the thoughtful and encouraging comments, and for highlighting the relevance of the additional theoretical perspectives. The suggested review by Storm is indeed relevant to the point about the discussed theory of consciousness - among all the theories, it actually describes the Global Neuronal Workspace theory first. The schematics in Figure 1b, summarizing required global synchrony of the conscious state (versus local synchrony of the non-conscious state), closely resemble the final figure schematic in our manuscript, which similarly illustrates the global coincident HFOs. This review paper provides an excellent summary of the major theories of consciousness, and we will continue to use it as a reference to our work.

We have revised the Discussion according to the reviewer's suggestion (lines: 532-537). Specifically, we now acknowledge that, while our study does not adjudicate between the competing approaches, the present findings may inform multiple perspectives on the neural basis of consciousness, including predictive processing, recurrent processing theory, and information integration theory. In fact, we fully endorse the conclusion of the review article advocating for unifying elements from the various approaches, and believe that the global coincident bursts will be helpful not only in the example GNWT used in the manuscript but also across broader theoretical frameworks.

From the perspective of these studies, coordination of HFOs across the brain in coincident bursts could be linked to global integration of information and thus provide a plausible neural mechanism for concepts like 'ignition' or 'global broadcasting' proposed in the global neuronal workspace hypothesis ⁷⁴At the same time, we acknowledge that such dynamics may also be compatible with other theoretical frameworks, including predictive processing and neurorepresentationalism, recurrent processing theory, and information integration theory; while our study does not adjudicate between these accounts, the present findings may inform multiple perspectives on the neural basis of consciousness (Storm, Neuron 2024)

Reviewer #2 :

The authors have responded thoroughly and satisfactorily to the major comments I provided.

In my earlier comments, I raised concerns about the manuscript's overall readability, a point also made by other reviewers. In their revision, the authors added subheadings – they improve structure.

However, several passages remain difficult to follow, so I recommend another round focused on clarity and readability. The points below are intended to guide targeted revisions:

(I) Sentence length.

Several sentences run 40–70 words. Please split very long sentences and aim for an average of ~15–25 words (with occasional variation for emphasis).

(II) Redundancy / wordiness.

Reduce pleonasms and stacked phrases. Examples (illustrative, not exhaustive):

I.69–70: “networks of connected areas” → “networks.”

I.68–69: “present a plausible activity for large-scale tracking of ... discharges” → recast; the construction is wordy and non-idiomatic.

I.71–72: “wide range ... spanning ... frequency ranges” → “range ...” (drop “spanning ... ranges”).

(III) Results vs. Methods.

Please keep methodological detail out of the Results (move to Methods and retain brief reminders only where essential). For example:

I.132: inclusion/selection criterion (“same inclusion criterion to ensure enough trials...”).

I.143–145: participant instructions (“patients were encouraged to visualize...”).

I.147–149: event time-stamping rule (midpoint between onset and offset; Suppl. Fig. S1).

I.149–150: cross-channel analysis setup (“temporal correlations ... from the same and different regions”) → define the pipeline in Methods.

I.187–190: LMM structure (Condition fixed; Subject random intercept), 10-ms bins, significance bars ($p < 0.05$).

I.191–196: post-hoc scheme (rotating reference level; six pairwise contrasts) and FDR correction across time bins.

I.233–236: repeated mention of the same LMM specifications.

(IV) Voice (active vs passive)

For readability, please favor active voice in the Results (e.g., “We observed/estimated/found ...”). In the Methods, active or passive is acceptable—choose one and apply it consistently within the section, avoiding agentless passives where the actor matters.

Further comments:

- Table S1. The column label “Rate of co-HFO bursts (Hz)” is unclear. Values up to ~2960 would imply ~2,960 events/s, which seems implausible for co-HFO detections. Please clarify or correct, if necessary.

- In I. 674ff and I. 699 text is struck through. Please check and correct.

We thank the reviewer for the thoughtful and constructive feedback, as well as for the detailed guidance aimed at improving the clarity and readability of the manuscript. We have carefully revised the text throughout to address all the points raised.

The manuscript has been comprehensively edited for clarity, readability, and conciseness. Sentences have been shortened and simplified to achieve a more balanced rhythm and improved flow. Redundant or wordy expressions have been removed or rephrased for precision. Methodological details that previously appeared in the Results have been relocated to the Methods section, with concise reminders retained where necessary to maintain readability. The Results and Methods sections now consistently use active voice to enhance engagement and clarity.

Table S1 has been corrected to accurately represent the rates of co-HFO bursts as normalized values per recalled word across all channels and participants(pasted below). The instances of struck-through text have also been corrected.

All these revisions were applied across the entire manuscript to ensure improved structure, coherence, and readability.

Table S1. Recall performance and co-HFO bursting rate abundance across participants.

This table summarizes the total number of words recalled, recall rate (as a percentage of studied words), and rates of co-HFO bursts *(in Hz per recalled word) across all recording channels for each participant during the recall period.* These values were used to assess the relationship between task performance and co-HFO activity, as reported in the main text.

Subject	Recalled Words (n)	Recall Rate (%)	Rates of Co-HFO bursts (Hz / word)
2	50	27.8	105.57
3	68	37.8	64.59
5	62	34.4	75.83
6	42	23.3	119.40
7	55	30.6	80.60
8	55	30.6	67.45
9	63	35	74.82
10	47	26.1	96.77
11	80	44.4	66.59

13	48	26.7	35.11
14	63	35	28.63
16	23	12.8	95.28
Average	54.3	30.7	48.55

Reviewer #3 (Remarks to the Author):

The authors have provided a comprehensive revision of their manuscript in which they addressed and clarified many of the issues raised in the initial review. Overall, this is an interesting manuscript that will be of interest to researchers focused on human memory.

We thank the reviewer for the positive evaluation and for recognizing the revisions made to improve the manuscript. We are pleased that the study is considered of interest to researchers working on human memory.

Reviewer #4 (Remarks to the Author):

First, I would like to thank the authors for a thorough revision, which has significantly improved the manuscript. In particular, the clarification of the clinical data and the justification of analytical parameters have strengthened the paper. However, some of my major concerns remain and were not fully addressed with additional data or analyses. In particular:

- **Event specificity and signal-to-noise ratio:** While the authors clarified that candidate HFO events were subjected to additional amplitude and duration thresholds, I remain concerned that many detected events may still include low-SNR or borderline oscillatory activity. Without a detailed distribution of event amplitudes and frequencies across regions, it is difficult to confirm that the reported co-activation patterns reflect robust HFOs rather than broad-band fluctuations or residual gamma activity. In addition, while the surrogate and threshold tests strengthen the evidence that co-HFO patterns are not purely noise, they do not fully address my primary concern.

We appreciate the reviewer's continued attention to the issue of event specificity and signal-to-noise ratio (SNR). We agree that this is a critical point, as ensuring that detected events represent genuine oscillatory bursts rather than broadband fluctuations or asynchronous transients in spectral power is essential for the validity of our conclusions.

In fact, this critical point of differentiating true oscillations from other non-oscillatory activities in the signal and during its processing has been central to our research for more than a decade. Over these years, we developed a methodology for detecting high frequency oscillations associated with physiological processes of memory and cognitive functions, which has been rigorously validated in a series of our previous publications (Matsumoto et al. 2013; Kucewicz et al. 2014; Cimbalnik et al. 2016; Kucewicz et al. 2017; Website ; Worrell et al. 2012). These studies demonstrated that the amplitude and duration-based thresholding methods, as applied in the currently revised manuscript, can reliably isolate events within discrete spectrogram ranges of the frequency and time space (Kucewicz et al. 2014; Kucewicz et al. 2017), which are qualitatively and quantitatively distinct from broadband gamma or other non-oscillatory activities (Cimbalnik et al. 2016; Kucewicz et al. 2017; Website ; Worrell et al. 2012). We also showed in these studies that the detected HFO bursts exhibit consistent amplitude, duration and frequency-span properties, as well as reproducible temporal dynamics during memory encoding and retrieval across cortical and hippocampal regions (Kucewicz et al. 2014; Kucewicz et al. 2017). The spectral and temporal parameters of the HFO detections described in these previous studies distinguish them from spectrally diffuse broadband activity. These results and methodology for detecting and carefully characterizing every candidate HFO detection has been further summarized in previous and more recent review articles from our group (Cimbalnik et al. 2016; Website ; Worrell et al. 2012) and others (Liu et al. 2022; Lachaux et al. 2012).

To directly address the reviewer's concern in the present dataset, we have now analyzed the spectral and temporal properties of each burst from 5,266,937 distinct detections used in the present study. For each detection, we determined its frequency at peak amplitude, frequency span (bandwidth), duration, and normalized amplitude, as in the previous study (Kucewicz et al. 2017). The resulting distributions (now presented in Supplementary Fig. S1) reveal clear relationships among these parameters across gamma, ripple, and fast ripple frequency ranges. Consistent with our previous findings (Kucewicz et al., 2017), the higher the frequency, the greater the frequency span, and the shorter the duration. In other words, detections at higher frequencies had proportionally greater frequency span (it is more difficult to accurately estimate the exact frequency in the higher ranges) and shorter duration (oscillations of the same number of cycles necessarily have a shorter duration at higher frequencies). Although our methodology used in the present study was even more careful to detect only oscillations (compared to the previous studies) by using the double-thresholding approach, there was still a subpopulation of detections with a greater frequency span - possibly reflecting broadband activities.

Therefore, like in the previous studies (Kucewicz et al. 2017), we applied an unsupervised k-means clustering ($k = 2$) procedure using the frequency span feature. The cluster of detections with the broader frequency span was excluded, as these events were more likely to represent spectrally diffuse potential broadband activities. This filtering step rejected approx. 8–15% of all detections. Importantly, repeating all main analyses using this more conservative, high-specificity subpopulation of all detections produced results that were consistent with those originally reported, confirming that the findings are robust to stricter detection criteria (see example of the main finding from Fig. 2c below).

To illustrate this entire methodological procedure in a more transparent and intuitive way, we now added panel b to the Supplementary Fig. S1, which shows an artificial example signal demonstrating how the detection algorithm isolates high-frequency oscillatory bursts, while excluding broadband activities due to sharp transients in both the raw and Hilbert-transformed domains. We have also added the estimated ranges of the frequency span and duration parameters for the example detections presented in the Supplementary Fig. S1a. Distributions of the frequency span from an example recall trial were also added in the Supplementary Fig. S1c to show the subpopulation of detections with more broadband frequency span. Updated supplementary Fig. S1 is pasted below for reviewer’s convenience.

Given that excluding these more broadband detections from the analysis did not change the main results, we chose not to incorporate this additional clustering filter step into the main analysis pipeline for the sake of simplicity and reproducibility in the future studies. The detections of higher frequency span form a minority of all detections and cannot be definitely classified into broadband activities as in the previous studies (Kucewicz et al., 2014; 2017; 2024). Instead, we report these validation procedures and their results in the manuscript as an additional test of our main results.

Methods (lines: 685-690):

To assess event-level specificity, we additionally quantified spectral and temporal features (peak frequency, frequency span, duration, amplitude) of all detections and applied a frequency span–based k-means clustering ($k = 2$) to exclude spectrally diffuse events of larger frequency span (8–15% rejected). Analyses restricted to the remaining high-specificity subset yielded results consistent with those reported (See Suppl. Fig. S1).

Figure S1. Detection and validation of high-frequency oscillations (HFOs).

a) Example raw signal with its decomposition into the studied ranges of high gamma (60–150 Hz), ripple (150–250 Hz), and fast ripple (250–600 Hz) frequencies. The amplitude envelope was extracted using the Hilbert transform and z-scored. Detected HFO bursts are highlighted by red boxes. The corresponding Hilbert matrix illustrates spectral power dynamics across the frequency scale and over time with the vertical white bars indicating the frequency span and the horizontal white bars the duration of each detection, respectively. b) Comparison of a synthetic broadband transient introduced into the signal, which was not detected by the algorithm, with a selectively detected oscillatory burst. Notice that the artifactual broadband event was not detected despite a relatively large amplitude and duration. c) Distribution of all HFO detections from an example recall trial shows a subpopulation of events with a higher frequency span (red points) relative to the majority of the detected events with a narrow frequency span (black points) identified by unsupervised clustering (k-means, $k = 2$, x mark the centroid of each cluster). Notice that excluding the subpopulation of higher frequency span did not affect the main results reported in the study.

- **Directionality and propagation of co-HFOs:** The manuscript does not provide evidence for structured temporal propagation or causal relationships among co-HFO events across cortical regions. The authors' explanation that directionality analysis is complex does not fully address

my concern that the widespread near-simultaneous co-activations could reflect non-specific or epileptiform-driven correlations rather than physiologically meaningful HFO coordination.

We thank the reviewer for emphasizing the analysis of directionality and causal propagation among co-HFO events as a potential tool for excluding the effects of non-specific or epileptiform-driven correlations. Our methodology was specifically developed to reassure excluding synchronous activities detected across multiple channels at the same time, which is typical of interictal epileptiform discharges, movement or muscle artifacts. Our detection methods have been validated in multiple studies for differentiating between the pathological and physiological activities, and for excluding artifacts (Cimbalnik et al. 2016; Matsumoto et al. 2013; Kucewicz et al. 2017; Website ; Worrell et al. 2012). In our previous response to this comment, we also pointed out the issue of temporal consistency of task-induced responses as another means for excluding the effects of pathological or artifactual activities, which are less likely to be triggered at a consistent time by cognitive tasks. Cognitive processing in the tasks can actually suppress the likelihood of epileptiform activities (Matsumoto et al. 2013).

We agree that assessing the temporal interactions across cortical regions would be highly informative, as in the case of our previous study (Kucewicz et al. 2014), where we showed different latencies of the HFO detections in particular critical areas (sensory earlier than higher order) and frequency ranges (fast ripple earlier than gamma). Our preliminary analysis described in the previous response to this comment has not found a temporal relationship within the co-HFO events that would be as clear as in this previous study. The primary aim of this study was focused on investigating the co-occurrence of high-frequency bursts at global anatomical (cortical lobes - not fine grain areas), frequency (ranges beyond the typical high gamma/ripple range) and temporal (phases of memory processing rather than phases of particular oscillatory events) scales. Nevertheless, the analyses already presented in the manuscript support a global temporal structure of the co-HFO discharges.

Firstly, the global clustering analysis (Results, Fig. 5) revealed sequential networks of co-HFO bursting that followed one another within single recall episodes. This observation indicates that the global co-activations are not simultaneous or random across all channels, but rather reveal a structured temporal organization of distributed cortical networks. Even within any one co-HFO global bursting, channels show increased detections not in the very same time bin but scattered within the co-HFO detection window. Our previous response showed that the millisecond-scale surrogate jitter procedure (± 50 – 100 ms) abolished this temporal organization beyond the coincidence detection window but did not investigate more precise micro-organization within that window. Here, we focused on the global ‘macro’ coordination of the HFO detections - our pending studies investigate more detailed temporal interactions between high and low frequency discharges detected on the macro- and micro-contacts in particular cortical areas.

We now clarify this point further in the revised Discussion (lines 512- 516).

Notably, these co-HFO bursting networks did not activate simultaneously but sequentially within recall episodes, indicating a structured temporal organization rather than random or purely synchronous co-activation. This pattern argues against non-specific epileptiform synchronization and suggests temporally coordinated cortical activity.

- Interpretation of large-scale co-activation: Although surrogate jitter analyses and threshold robustness checks mitigate concerns about inflated correlations, the remaining uncertainty about the specificity of detected events limits confidence in the functional interpretation of widespread

cortical co-HFOs. The manuscript does not convincingly dissociate genuine oscillatory coordination from possible confounds related to epileptiform activity.

We appreciate the reviewer's concern regarding the interpretation of widespread co-HFO activation and its implications for novelty in light of possible confounds related to epileptiform activities. We hope that our responses to the previous two comments have already addressed the issue of excluding the pathological and non-specific activities from the detection analysis, as we have now clarified in the Methods (lines: 685-690), Supplementary material and the Discussion (lines 512- 516).

The reported global co-HFO discharges are proposed to reflect coordinated activation of distributed neuronal assemblies rather than another non-specific uniform activity in the cortex. The observed systematic pattern of suppression followed by enhancement in the likelihood of coincident detection before and during successful encoding or recall, respectively (Figs. 2–3), argue against non-specific epileptiform or artifactual activity. In recall, we observed this suppression of co-HFO bursting approx. 1s before the onset of vocalization (Fig. 5a - notice the diagonal pattern described in the caption) - it would be hard to imagine an epileptiform or other activity that would consistently be suppressed like that without any external stimuli or behavioral triggers. Moreover, the consistent modulation by task phase and memory outcome across subjects and cortical lobes supports the conclusion that these events reflect organized, behaviorally relevant coordination rather than more random in time artifactual or pathological synchronization.

- Novelty and scope: While the study provides interesting observations on large-scale co-HFO dynamics, the current manuscript does not provide sufficient evidence of specific, novel mechanistic insight that would justify publication in a high-impact journal such as Nature Communications. The remaining concerns regarding event specificity and network interpretation limit the conceptual advance

Regarding the concern for conceptual advancements, our work put the previous observations of hippocampo-cortical and cortico-cortical ripples in a new *global* perspective of the frequency, anatomical and temporal scales. First of all, the detections are not limited to ripples in the 70-150 Hz range but extend from high gamma to fast ripple frequencies. Secondly, coincident bursting involves equally the sensory and associational cortical areas beyond the framework of limbic networks centered on the hippocampus. Finally, we demonstrate for the first time that co-HFO bursting reveals sequences of global cortical networks during memory processing. This higher-order level of organization presents testable mechanisms for large-scale processing of information proposed recently in our review article (Kucewicz, et al. 2024) — from local bursts of neuronal firing to global neural networks — providing a new framework for linking micro-scale neuronal assemblies with macro-scale cortical networks. While we refrain from over-interpreting the mechanistic insights from our obtained results, this study offers a novel large-scale perspective, bridging previous investigations of hippocampal and cortical HFOs, global cortical networks, and memory and cognitive functions.

We have revised the first paragraph of the Discussion to better articulate the novel aspects of this work compared to the previous studies. The revised paragraph is pasted below for the Reviewer's convenience (lines 338- 357).

In this study, we took a global perspective beyond the more focal framework of hippocampo-cortical ripples to explore wider scales of anatomical, temporal and spectral organization of coincident HFO

bursting. Our results show that bursts of high frequency oscillations co-occur over a large anatomical scale of multiple sensory and associational cortical areas. These coincident bursting happens across all phases of performing a task, but their occurrence is strongly modulated by memory recalling and encoding. This co-HFO bursting is suppressed before and enhanced during presentation of the stimuli that will subsequently be recalled compared to those that will be forgotten, and peaks approximately 300 ms before the recall vocalization. Around half of the recorded sites from all cortical lobes are engaged in this global coordination of HFO bursting underlying encoding and recalling the remembered stimuli, which was not specific to particular areas or words. Multiple distinct cortical networks were coordinated together in a sequence of bursting when recalling a single word. Instead of one bursting network, our results suggest a sequence of temporally coordinated networks. These findings suggest possible roles of globally coordinated discharges of high frequency oscillations in organizing communication and integration of information for memory processing. Our study offers a new large-scale perspective on the mechanism and functions of coincident HFO bursts, bridging previous investigations of hippocampal and cortical ripple HFOs, global cortical networks, and their role in human memory and cognition.